



**Defining BGC-Argo-based metrics of ocean health and biogeochemical**
**functioning for the evaluation of global ocean models**
Alexandre Mignot[1], Hervé Claustre[2,3], Gianpiero Cossarini[4], Fabrizio D'Ortenzio[2,3], Elodie
Gutknecht[1], Julien Lamouroux[1], Paolo Lazzari[4], Coralie Perruche[1], Stefano Salon[4], Raphaelle
Sauzède[3], Vincent Taillandier[2,3], Anna Teruzzi[4]
[1]Mercator Océan International, Ramonville-Saint-Agne, France
[2]Laboratoire d'Océanographie de Villefranche-sur-Mer, Villefranche-sur-Mer, CNRS and
Sorbonne Université, 06230 Villefranche-sur-Mer, France
[3]Institut de la Mer de Villefranche, CNRS and Sorbonne Université, 06230 Villefranche-sur-
Mer, France
[4]National Institute of Oceanography and Applied Science-OGS, Trieste, Italy

17        Numerical models of ocean biogeochemistry are becoming a major tool to detect and

predict the impact of climate change on marine resources and ocean health. Classically, the
validation of such models relies on comparison with surface quantities from satellite (such as
chlorophyll-*a* concentrations), climatologies, or sparse *in situ* data (such as cruises
observations, and permanent fixed oceanic stations). However, these datasets are not fully
suitable to assess how models represent many climate-relevant biogeochemical
processes.  These limitations now begin to be overcome with the availability of a large
number of vertical profiles of light, pH, oxygen, nitrate, chlorophyll-*a* concentrations and
particulate backscattering acquired by the Biogeochemical-Argo (BGC-Argo) floats network.
Additionally, other key biogeochemical variables such as dissolved inorganic carbon and
alkalinity, not measured by floats, can be predicted by machine learning-based methods
applied to float oxygen concentrations.  Here, we demonstrate the use of the global array of
BGC-Argo floats for the validation of biogeochemical models at the global scale. We first
present 18 key metrics of ocean health and biogeochemical functioning to quantify the
success of BGC model simulations. These metrics are associated with the air-sea $CO_2$ flux,
the biological carbon pump, oceanic pH, oxygen levels and Oxygen Minimum Zones

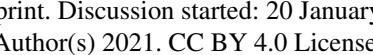


(OMZs). The metrics are either a depth-averaged quantity or correspond to the depth of a
particular feature. We also suggest four diagnostic plots for displaying such metrics.
## 1. Introduction
Since pre-industrial times, the ocean had taken up ~36 % of the $CO_2$ emitted by the
combustion of fossil fuel (Friedlingstein et al., 2019) leading to dramatic change in the
ocean's biogeochemical (BGC) cycles, such as ocean acidification (Iida et al., 2020).
Moreover, deoxygenation (Breitburg et al., 2018) and change in the biological carbon pump
are now manifesting on a global scale (Capuzzo et al., 2018; Osman et al., 2019; Roxy et al.,
2016). Together with plastic pollution (Eriksen et al., 2014) and an increase in fisheries
pressure (Crowder et al., 2008), major changes are therefore occurring in marine ecosystems
at the global scale. In order to monitor these ongoing changes, derive climate projections and
develop better mitigation strategies, realistic numerical simulations of the oceans' BGC state
are required.
Numerical models of ocean biogeochemistry represent a prime tool to address these issues
because they produce three dimensional estimates of a large number of chemical and
biological variables that are dynamically consistent with the ocean circulation (Fennel et al.,
2019). They can assess past and current states of the biogeochemical ocean, produce short-
term to seasonal forecasts as well as climate projections.  However, these models are far from
being flawless, mostly because there are still huge knowledge gaps in the understanding of
key biogeochemical processes and, as a result,  the mathematical functions that describe BGC
fluxes and ecosystems dynamics are too simplistic (Schartau et al., 2017). For instance, most
models do not include a radiative component for the penetration of solar radiation in the
ocean. It has been nevertheless shown that coupling such a component with a BGC model
improves the representation of the dynamics of phytoplankton in the lower euphotic zone
(Dutkiewicz et al., 2015). Additionally, the parameterisation of the mathematical functions
generally result from laboratory experiments on few a priori expected representative species
and may not be suitable for extrapolation to ocean simulations that need to represent the large
range of organisms present in oceanic ecosystems (Schartau et al., 2017; Ward et al., 2010).
Furthermore, the assimilation of physical data in coupled physical-BGC models that improves
the physical ocean state can paradoxically degrade the simulation of the BGC state of the



ocean (Fennel et al., 2019; Park et al., 2018). A rigorous validation of BGC models is thus
essential to test their predictive skills, their ability to reproduce BGC processes and estimate
confidence intervals on model predictions (Doney et al., 2009; Stow et al., 2009).
However, the validation of BGC models is presently limited by the availability of data. It
relies principally on comparison with surface quantities from satellite (such as chlorophyll-$a$
concentrations), cruises observations, and few permanent oceanic stations (e.g., Doney et al.,
2009; Dutkiewicz et al., 2015; Lazzari et al., 2012, 2016; Lynch et al., 2009; Séférian et al.,
2013; Stow et al., 2009). All these datasets neither have a sufficient vertical or temporal
resolution, nor a synoptic view nor can provide all variables necessary to evaluate how
models represent climate-relevant processes such as the air-sea $CO_2$ fluxes, the biological
carbon pump, ocean acidification or deoxygenation.
In 2016, the Biogeochemical-Argo (BGC-Argo) program was launched with the goal
to operate a global array of 1000 BGC-Argo floats equipped with oxygen ($O_2$), chlorophyll $a$
(Chl$a$) and nitrate ($NO_3$) concentrations, particulate backscattering ($b_{bp}$), pH and downwelling
irradiance sensors (Biogeochemical-Argo Planning Group, 2016; Claustre et al., 2020).
Although the planned number of 1000 floats has not been reached yet, the BGC-Argo
program has already provided a large number of quality-controlled vertical profiles of $O_2$,
Chl$a$, $NO_3$, $b_{bp}$, and pH (Fig. 1). With respect to $O_2$, Chl$a$, $NO_3$, and $b_{bp}$; the North Atlantic
and the Southern Ocean are reasonably well sampled whereas pH is so far essentially sampled
in the Southern Ocean. At regional scale, the Mediterranean Sea is also fairly well sampled by
BGC-Argo floats (Salon et al., 2019; Terzić et al., 2019).  However, there are still, large
under-sampled areas, like the subtropical gyres or the sub-polar North Pacific. Nevertheless,
the number of quality-controlled observations collected by the BGC-Argo fleet is already
greater than any other data set (Claustre et al., 2020). The BGC-Argo data have also an
unprecedented temporal and vertical resolution of key variables acquired simultaneously as
well as a satisfactory level of accuracy and stability over time (Johnson et al., 2017; Mignot et
al., 2019).  Thanks to machine learning based methods (Bittig et al., 2018; Sauzède et al.,
2017), floats equipped with $O_2$ sensors can be additionally used to derive, vertical profiles of
$NO_3$, phosphate ($PO_4$), silicate (Si), alkalinity (Alk), dissolved inorganic carbon (DIC), pH
and $pCO_2$. All these specificities overcome the limitations of previous data sets  from now and
open new perspectives for the validation of BGC models (Gutknecht et al., 2019; Salon et al.,
2019; Terzić et al., 2019).



We aim to demonstrate the use of the BGC-Argo global array for the validation of
BGC models at the global scale. In regional seas or enclosed basins, where a limited number
of floats have been so far deployed, point-by-point model-observation comparison is possible
(Gutknecht et al., 2019; Salon et al., 2019). However, at the global scale, the BGC-Argo
dataset provides a massive and ever-growing amount of data, and it can be difficult to
manipulate this large data set, especially when it comes to evaluate a 3-D time-varying model
simulation for about ten variables. In such cases, it is useful to define observationally-based
metrics that are able to quantify the skill of a model to represent key oceanic processes
(Russell et al., 2018). These metrics are quantities that summarize a particular process into a
single number [e.g., the amplitude or the depth of an Oxygen Minimum Zone (OMZ)]. In this
study, we present 18 metrics of ocean health and biogeochemical functioning for the
assessment of a BGC model simulation. The metrics are either a depth-averaged quantity (e.g,
nutrients concentration, Chl$a$, …) or correspond to the depth of a particular feature (e.g.,
nitracline). These metrics are associated with the air-sea $CO_2$ flux, the biological carbon
pump, oceanic pH, oxygen levels and Oxygen Minimum Zones (OMZs).
The paper is organised as follow: section 2 presents the data sets used in the study. In
section 3, we define the metrics necessary to compare the model to floats' observations. In
section 4, we show examples of diagnostic plots for displaying the metrics. In section 5, we
discuss metrics relative to optical properties in the water column. Finally, section 6
summarizes and concludes the study.
**2. Data**
**a. BGC-Argo floats observations**
The float data were downloaded from the Argo Coriolis Global Data Assembly Centre
in France (ftp://ftp.ifremer.fr/argo). The CTD and trajectory data were quality controlled
using the standard Argo protocol (Wong et al., 2015). The raw BGC signals were transformed
to biogeochemical variables  and quality-controlled according to international BGC-Argo
protocols (Johnson et al., 2018b, 2018a; Schmechtig et al., 2015, 2018; Thierry et al., 2018;
Thierry and Bittig, 2018).

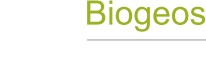 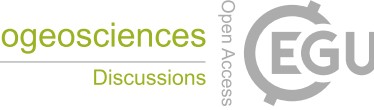

In the Argo data-system, the data are available in three data modes, "Real-Time",
"Adjusted" and "Delayed" (Bittig et al., 2019). In the "Real-time" mode, the raw data are
converted into state variable and an automatic quality-control has been applied to "flag" gross
outliers. In the "Adjusted" mode, the "Real-time" data receive a calibration adjustment in an
automated manner. In the "Delayed" mode, the "Adjusted" data are adjusted and validated by
a scientific expert.  While the "Real-Time" and "Adjusted" data are considered acceptable for
operational application (data assimilation), the "Delayed" mode" is designed for scientific
exploitation and represent the highest quality of data with the ultimate goal, when time-series
with sufficient duration will have been acquired, to possibly extract climate-related trend.
However, for some parameters, only a limited fraction of data is accessible in "Delayed-
Mode". Consequently, for each parameter, we selected the highest quality of data that did not
compromise too much the number of observations available (see Table 1). We removed data
with missing location or time information and flagged as "Bad data" (flag =4). Depending on
the parameter and the associated data mode, we also excluded data flagged as "potentially bad
data" (flag=3) (see Table 1).
Particulate Organic Carbon (POC) concentrations were derived from $b_{bp}$ observations.
First, three consecutive low-pass filters were applied on the vertical profiles of  $b_{bp}$  to remove
spikes (Briggs et al., 2011): a 2-points running median followed by a 5-points running
minimum and 5-points running maximum. Then, the filtered $b_{bp}$ profiles were converted into
POC using the relationship proposed by Cetinic et al. (2012), i.e,  $POC=35422* b_{bp}-14.4$.
Negative values resulting from this transformation were set to 0.
Finally, we complemented the existing BGC-Argo dataset with pseudo-observations
of $NO_3$, $PO_4$ , Si, and DIC concentrations as well pH and $pCO_2$ using the CANYON-B neural
network (Bittig et al., 2018). CANYON-B estimates vertical profiles of nutrients as well as
the carbonate system variables from concomitant measurements of floats pressure,
temperature, salinity and $O_2$ qualified in "Delayed "mode together with the associated
geolocation and date of sampling.
**b.  CMEMS global BGC Model**





2       The global model simulation used in this study (see Appendix A.1) originates from the

Global Ocean hydrodynamic-biogeochemical model, implemented and operated by the Global
Monitoring and Forecasting Center of the EU, the Copernicus Marine Environment
Monitoring Service (CMEMS). It is based on the coupled NEMO–PISCES model and it is
constrained by the assimilation of satellite Chl*a* concentrations. The BGC model is forced
offline by daily fields of ocean, sea ice and atmosphere. The ocean and sea ice forcing come
from Mercator Ocean global high-resolution ocean model (Lellouche et al., 2018) that
assimilates along-track altimeter data, satellite Sea Surface Temperature and Sea-Ice
Concentration, and *in situ* temperature and salinity vertical profiles. The BGC model has a
1/4° horizontal resolution, 50 vertical levels (with 22 levels in the upper 100 m, the vertical
resolution is 1m near the surface and decreases to 450m resolution near the bottom). It
produces daily outputs of Chl*a*, $NO_3$, $PO_4$, Si, $O_2$, pH, DIC and Alk, and weekly outputs of
POC (resampled offline from weekly to daily frequency through linear interpolation) from
2009 to 2017. The POC model used in this study corresponds to the sum two size classes of
particulate organic matter modelled by PISCES (Aumont et al., 2015). Partial pressures of
$CO_2$ values are calculated offline from the modelled DIC, Alk, temperature and salinity data
using the seacarb program for R (https://CRAN.R-project.org/package=seacarb). The Black
Sea was not taken into account in the present analysis because the model solutions are of very
poor qualities. Finally, the daily model outputs were collocated in time and the closest to the
BGC-Argo floats positions, and they were interpolated to the sampling depth of the float
observations. The characteristics of the model are further detailed in the appendix.
**3. Metrics**

26       In this section, we present 18 key metrics of ocean health and biogeochemical

functioning. The metrics are associated with the air-sea $CO_2$ flux, the biological carbon pump,
oceanic pH, oxygen levels and Oxygen minimum zones (OMZs). The metrics are described
below and summarized in Table 2.

31       **a.  Air-sea $CO_2$ flux**



1       The air-sea $CO_2$ flux is generally calculated following a bulk formulation

2       (Wanninkhof, 2014), $F_{CO2}=k\alpha(pCO_{2atm} - spCO_2)$, where $F_{CO2}$ is the air-sea $CO_2$ flux, $\alpha$ is the

3       $CO_2$ solubility in seawater, k is a gas transfer coefficient that depends on wind speed, $spCO_2$

4       is the partial pressure of $CO_2$ at the ocean's surface, and $pCO_{2atm}$ is the partial pressure of

5       $CO_2$ in the atmosphere. Among the uncertainties affecting the different components of the

6       model $CO_2$ flux, BGC-Argo data can contribute to estimate that on $spCO_2$. Thus, the

7       validation of $pCO_2$ plays a critical role to assess the skill of a BGC model in representing

8       correctly the air-sea $CO_2$ flux.

Here, $spCO_2$ is defined as the average of $pCO_2$ profile between the surface and the
mixed layer depth (MLD). Following De Boyer et al. (2004), the MLD is computed as the
depth at which the change in potential density from its value at 10 m exceeded 0.03 kg m$^{-3}$.
**b. Oceanic pH**
Ocean acidification is the decrease in oceanic pH due to the absorption of
anthropogenic $CO_2$. The acidification of the ocean is expected to impact primarily the surface
oceanic waters as well as the 200-400 m layer (Kwiatkowski et al., 2020). Assessing how
models correctly represent oceanic pH at the surface is therefore critical if we aim to derive
accurate climate projections on acidification. The surface ocean pH (spH) is defined as the
average of pH profile between the surface and the base of the mixed layer and the pH in the
200-400 m layer ($pH_{200-400}$) as the average of pH profile in this layer.
**c. Biological carbon pump**
The biological carbon pump is the transformation of nutrients and dissolved inorganic
carbon into organic carbon in the upper part of the ocean through phytoplankton
photosynthesis and its subsequent transfer of this organic material into the deep ocean.
A useful way to investigate the biological carbon pump is to look at the depth-
averaged concentrations in nutrients ($NO_3$, $PO_4$, and Si), DIC, Chl*a* and POC computed from
the surface down to the MLD, hereinafter denoted $sNO_3$, $sPO_4$, sSi, sDIC, sChl and sPOC. To
assess the quantity of POC that is exported to the deep ocean, we compute the mesopelagic




POC concentration (POC$_{meso}$), which correspond to the depth-averaged POC concentrations
between the base of the mixed layer down to 1000 m (Dall'Olmo and Mork, 2014).
At the base of the euphotic layer of stratified systems, a Chl$a$ maximum (hereinafter
denoted Deep Chlorophyll Maximum, DCM) develops that generally escapes detection by
remote sensing (Barbieux et al., 2019; Cullen, 2015; Letelier et al., 2004; Mignot et al., 2014,
2011). It has been suggested that the DCM plays an important role in the synthesis of organic
carbon by phytoplankton (Macías et al., 2014). The DCM is therefore an important feature to
be assessed in BGC models with respect to the production of organic carbon and more
generally to the biological carbon pump. The depth and magnitude of DCM (H$_{dcm}$ and Chl$_{dcm}$)
are helpful metrics for the assessment of DCM dynamics. The depth of the DCM is calculated
as the depth where the maximum of Chl$a$ occurs in the profile with the criterion that H$_{dcm}$
should be deeper than H. The magnitude of the DCM is computed at the value at H$_{dcm}$.
Finally, the depth of nitracline (H$_{nit}$) is also evaluated as it is an important driver for H$_{dcm}$ and
Chl$_{dcm}$ (Barbieux et al., 2019; Herbland and Voituriez, 1979). Following Richardson and
Bendtsen (2019), H$_{nit}$ was computed at the depth at which NO$_3$ = 1 µmol kg$^{-1}$.
**d. Oxygen levels and oxygen minimum zones**
Oxygens levels in the global and coastal waters have declined over the whole water
column over the past decades (Schmidtko et al., 2017) and OMZs are expanding (Stramma et
al., 2008). Assessing how models correctly represent ocean oxygen levels as well as the
OMZs is therefore critical. We evaluate oxygen levels in 3 layers, at the surface, at 300 m and
at 1000 m. The surface O$_2$ (sO$_2$), important for the air-sea O$_2$ flux, is defined as the average
of O$_2$ profile in the mixed layer. The oxygen at 300 m (O$_{2\,300}$), a depth where large areas of
the global ocean have very low O$_2$ (Breitburg et al., 2018), is defined as the average of O$_2$
profile between 250 and 300 m. The deep oxygen content, (O$_{2\,1000}$), is defined as the average
of O$_2$ profile between 950 and 1000 m. Finally, to characterize the OMZs, we evaluate the
depth (H$_{O2min}$) and concentration (O$_{2min}$) of O$_2$ minimums. O$_2$ level lower than 80 µmol kg$^{-1}$
are used to characterize OMZs (Schmidtko et al., 2017).
**4. Diagnostic plots to display the BGC-Argo based metrics**



Based upon the existing literature (e.g., Aumont et al., 2015; Cossarini et al., 2019; Doney
et al., 2009; Dutkiewicz et al., 2015; Gutknecht et al., 2019; Salon et al., 2019; Séférian et al.,
2013; Terzić et al., 2019), we propose 4 graphical representations that can be used to display
the novel validation metrics and to assess the skill of a model in reproducing a particular
process or variable: Taylor diagrams, scatterplots, spatial maps, and time series.

8        **a.  Taylor diagram**

Taylor diagrams are useful to display simultaneously information on model-data skill
for a suite of metrics (Taylor, 2001). These diagrams combine the Pearson correlation
coefficient (r), root-mean-square difference (RMSD) and the model standard deviation (SD).
In order to represent all metrics with different units into a single diagram, we use a
normalized Taylor diagram (RMSD and the model SD are divided by the SD of the
observations). In the diagram, the Pearson correlation coefficient between the model and the
observations is related to the azimuthal angle. The normalized SDs are proportional to the
radial distances from the origin. The observational reference is indicated along the x-axis and
corresponds to the normalized SD and r =1. Finally, the normalized RMSD is proportional to
the distance from the observational difference.

21        **b.  Scatter/Density plots**

In validation exercises, scatter plots are useful to identify relationships between the
predicted and observed values. It is common to add a least squares regression line to quantify
the strength of the linear relationship between the observed and predicted values. Scatter plots
are also helpful to show other patterns in data, such as non-linear relationships, clusters of
points and outliers. In those cases, when a large amount of data points has to be plotted (like
in our study), the points overlap to a degree where it can be difficult to distinguish the
relationship between the variables. To overcome this, scatter plots are displayed as density
plots, where each axis is divided in a number of bins while the colour within each bin
indicates the number of points.

33        **c.  Spatial maps**





Spatial maps draw attention to the spatial distribution of a given metric. The maps are
handy to determine if the model is skilled in reproducing global patterns, spatial gradients,
and basins inter-difference. It is also helpful to display the BIAS and RMSD between
predicted and observed values on a spatial map to quickly determine regions where the model
uncertainty is the highest.  Depending on the context, the comparison between the model and
the observation can be performed either on a climatological level, or for a specific period
(year, month, etc ..). In our case, the scarcity of observations imposes us to display all data
(from 2009 to 2017; the period of analysis of the model simulation) in a climatological way if
we want to highlight large scale patterns. To do so, the metrics from 2009 to 2017 are
averaged in 4°x4° bins, bins with less than 4 points being not included. We also computed the
BIAS and RMSD within each bin.
**d.  Seasonal time-series**
Taylor diagrams, scatter plots and spatial maps are powerful diagnostics plots to
evaluate the global skills of a model but understanding the causes of difference remains
somewhat limited with these diagrams.  Rather, the comparative analysis of seasonal time-
series of multiple metrics and their inter-relationships is a powerful tool to highlight and to
understand BGC processes. This is especially true for the biological carbon pump that has a
strong seasonal variability due to the seasonal variation in sunlight, surface heating and
surface wind (Williams and Follows, 2011).  As a matter of fact, the analysis of seasonal
dynamics in nutrients as well as in phyto- and zoo- plankton has a rich history for the
development of BGC model (Evans and Parslow, 1985; Riley, 1946).
**5.  Results: Application to CMEMS global model**
Examples of the diagnostic plots described in section 4 in combination with the metrics
defined in Section 3 are shown. The objective of this section is to illustrate the opportunities
offered by the BGC-Argo-based metrics for evaluating global BGC model solutions, rather
than to provide a full evaluation of the CMEMS global model. Consequently, for each
diagnostic plot, we only present one detailed example. The density plots and spatial maps for
all metrics are displayed in the Appendix section (Fig. A1-A36).



### a. Taylor diagram

The CMEMS global model skill is summarized in the normalized Taylor diagram
(Fig. 2). The oxygen levels metrics ($sO_2$, $O_{2\ 300}$, $O_{2\ 1000}$), $pH_{200-400}$, the average nutrients and
DIC concentrations in the mixed layer are particularly well represented in the model. The
correlation coefficients are greater than 0.95, the predicted SDs are close the observed SDs
and the normalized RMSDs are lower than 0.4. The OMZs as well as the depths of DCM and
nitracline are reasonably well represented in the model, with r > 0.9 (OMZs) and r > 0.8 (for
$H_{nit}$ and $H_{dcm}$) and normalized RMSDs <0.6. The variability in the predicted $O_{2min}$ is however
larger than the observed ones. Finally, the POC concentrations, the Chl*a* in the mixed layer
and at the DCM as well as $spCO_2$ and spH are the worst predicted metrics. The normalised
RMSD is greater than 0.7-0.8, r is between 0.4 and 0.6, and the amplitude of model variations
is lower than the BGC-Argo observations.
The representation of all metrics into a single Taylor diagram allows to rapidly
evaluate the strengths and the weaknesses of a model simulation. For instance, the CMEMS
global model is skilled in reproducing oxygen levels and the cycling of nutrients and DIC in
the mixed layer, but the representation of Chl*a* and POC needs to be improved.

### b. Scatter/Density plots

The density plots for all metrics are displayed in the Appendix section (Fig. A1-A18).
Here, we detail only the density plot for $O_{2min}$ to illustrate the potential of such representations.
Figure 3 shows the comparison between the observed and predicted $O_{2min}$ values. The
regression line, the slope, and the intercept as well the coefficient of determination ($R^2$) are
indicated. Overall, the model and the float $O_{2min}$ are in good agreement with a slope close to 1
and $R^2$ close to 0.8. There is however a positive offset of ~11 µmol kg$^{-1}$ across all $O_{2min}$ values
suggesting that the modelled OMZs are on average too much oxygenated by a constant value.
It is worth noting that the scatter around the regression line is larger for $O_{2min}$ > 50 µmol kg$^{-1}$,
which corresponds to the Atlantic OMZ around the Cap Verde Archipelago (Fig. A35). This
suggests that the uncertainty in this OMZ is particularly high, as confirmed in Fig. A35.





2       **c.  Spatial maps**

4       The spatial maps for all metrics are displayed in the Appendix section (Fig. A19-A36),

while we detail hereafter the spatial distribution of sChl.

7       Figure 4 shows the spatial distribution of sChl estimated from the BGC-Argo floats

(Fig. 4a), the model (Fig. 4b), the BIAS (Fig. 4c) and the RMSD (Fig. 4d). As already noticed
in Fig. 1, the density of sChl observations is satisfactory for high latitude regions (latitudes >
50° N and S) whereas it is poor in subtropical gyres and the Equatorial band. Nevertheless,
large scale patterns in sChl are still distinguishable in Fig. 1a, especially the juxtaposition of
the high-latitudes-high- sChl regions with the low-latitudes-low- sChl regions. The model
(Fig. 4b) exhibits large-scale, coherent patterns. However, the model tends to be lower than
the BGC-Argo observations in the high-latitudes region and higher in the subtropical gyres
(Fig. 4c). The RMS difference between the predicted and the observed values seems to be
quite uniform, suggesting the uncertainty in model sChl is fairly constant in all oceanic
basins.

19       **d.  Seasonal time-series**

21       An example of a BGC-Argo float seasonal time-series compared to a simulation of the

same time-series along the float trajectory is presented in Fig. 5 for a case study in the North
Atlantic during the "spring bloom" .

25       Figure 5 compares the seasonal time series of MLD, sChl, $sNO_3$, sSi and $sPO_4$

derived from the BGC-Argo floats observations (blue) and from the model simulation
(yellow). The seasonal cycle of MLD, sChl and nutrients is typical of the North Atlantic
bloom dynamics (Dale et al., 1999; Mignot et al., 2018). In spring, phytoplankton
concentration, as measured by sChl increases dramatically and it is accompanied by a
consumption of inorganic nutrients in the mixed layer. The increase in sChl stops when one or
several nutrients become exhausted and the nutrients-Chl*a* system remains in an equilibrium
phase. In fall, as the mixed layer starts deepening, deep nutrients and inorganic carbon are
entrained in the surface layer driving an increase in surface concentrations. However, the



decrease in sea surface light and the increase in upper ocean mixing drive phytoplankton cells
away from the well-lit surface inducing a decrease in phytoplankton abundance and thus sChl.
The seasonal cycle of sChl and nutrients is well approximated by the model with the
timings of minima, maxima and the onset of the bloom being correctly represented. The
winter- sChl -minimum and winter-nutrients-maxima are also properly estimated by the
model. However, the summer- sChl -maximum is underestimated and the summer- $sNO_3$ -
minimum and summer- $sPO_4$ -minimum are overestimated while the summer- sSi -minimum
is correctly represented.  This explain the negative BIASs observed in the spatial map of sChl
in the North Atlantic (Fig. 4) and the positive BIAS in the spatial map of $sNO_3$ and $sPO_4$  in
the North Atlantic (Figs. A23 and A24).
The conjoint analysis of the seasonal times-series of Chl*a* and nutrients strongly
suggest that modelled rates of primary production are too weak in summer so that $sNO_3$  and
$sPO_4$ are not consumed fast enough by phytoplankton.  The summer sSi being correctly
estimated, we can also hypothesized that the main phytoplankton class in the model
consuming Si, i.e; the diatoms (Aumont et al., 2015), are well represented whereas the other
phytoplankton class in the model , i.e.,  nanophytoplankton, are misrepresented during
summer. The reasons for this could be that nanophytoplankton growth rates are too weak or
that grazing on nanophytoplankton is too strong.
The underestimation in the rates of primary production has a direct impact on the
oceanic carbon cycle in the North Atlantic (Fig. 6). The summer sDIC are higher in the model
compared to the BGC-Argo estimates. Similarly, the summer sPOC concentrations are too
low, suggesting that the uptake of atmospheric $CO_2$ and the transformation of dissolved
inorganic carbon into organic carbon are too weak in the model during summer. However,
this seems to have a limited effect on the export of POC to the deep ocean as the modelled
POC concentrations in the mesopelagic layer are consistent with the BGC-Argo observations
during summer.
**6.  Perspectives: metrics relative to ocean optical properties**



BGC-Argo floats equipped with sensors measuring the downward planar irradiance are
essential observations to evaluate the performance of recently-developed BGC models that
resolve the spectral and directional properties of the underwater light field. For several years,
the number of BGC models coupled with a multispectral light module has been steadily
increasing (Baird et al., 2016; Dutkiewicz et al., 2015; Gregg and Rousseaux, 2016; Lazzari et
al., 2020; Skákala et al., 2020). Such models require dedicated observations and metrics to
evaluate their skill in representing the ocean's optical properties of the ocean. Diffuse
attenuation coefficient for downwelling irradiance ($K_d$) is one of the most common properties
to characterise the optical state of the ocean (Sosik, 2008). Values of $K_d$ can be derived at
three different wavelengths (380, 412, 490 nm) from the BGC-Ago floats observations.  This
metric also provides information about the constituents of seawater (Organelli 2017)
(phytoplankton for $K_d$ at 490 nm and coloured dissolved organic carbon for $K_d$ at 380 nm and
412 nm) and is complementary to Chl*a* measurements for the assessment of the modelled
phytoplankton dynamics.

16       As an example of the potentiality of such comparison, spatial distribution of $K_d$ at 490

nm in the first optical depth estimated from the BGC-Argo floats and from a model of the
Mediterranean Sea equipped with a multispectral light module (Lazzari et al., 2020)
(Appendix A.2) are shown in Fig. 7.  The BGC-Argo estimated $K_d$ at 490 nm exhibits a basin-
scale pattern, with high values in the North-Western Mediterranean Sea and lower values in
the Eastern Mediterranean Sea, consistent with the spatial distribution of surface Chl*a* in the
Mediterranean Sea (Bosc et al., 2004).  The model is able to reproduce the large-scale pattern
of $K_d$ at 490 nm, but it tends to underestimate $K_d$ at 490 nm in the North-Western
Mediterranean Sea; area where the RMSD is also the highest. The annual cycle of
phytoplankton being largely influenced by a spring bloom in this region (Bosc et al., 2004;
D'Ortenzio et al., 2014), we can speculate that the underestimation of $K_d$ at 490 nm highlights
a possible misrepresentation of the spring bloom in the model that yields to lower
phytoplankton and Chl*a* concentrations.
**7. Conclusion**

32       Biogeochemical ocean models are powerful tools to monitor changes in marine

ecosystems and ecosystem health due to human activities, make climate projections and help



developing better strategies for mitigation. However, these models are subject to flaws and
require rigorous validation processes to test their predictive skills. The model's evaluations
have long been damped by the lack of *in situ* observations, which has certainly slowed the
development and the improvement of BGC models. The amount of observations collected by
the BGC-Argo program is now greater than any other *in situ* data set (Claustre et al., 2020)
and thus offers new opportunities for the validation of BGC models.

8        In this study, we use the global data set of BGC-Argo observations to validate a state-of-

the-art BGC model simulation. Our aim was to demonstrate the invaluable opportunities
offered by the BGC-Argo observations for evaluating global BGC model solutions. To ease
the comparison between model and observations at global scale, we proposed 18 key metrics
of ocean health and biogeochemical functioning. These metrics are either a depth-averaged
quantity or correspond to the depth of a particular feature. We did not propose BGC-Argo-
based phenology metrics (Gittings et al., 2019), because the numbers of observation per
month and per bin is still presently too low, to derive such robust metrics. We suggested 4
diagnostic plots, which we believe are particularly suitable for displaying the metrics in
support of identification of model-data difference and subsequent analysis of model
representativity. We also discuss the promising avenue of BGC-Argo-based metrics relative
to optical properties in the ocean for the validation of the new generation of BGC model
equipped with a multispectral light module.

22       We assumed that the differences between the observed and predicted BGC values were

only attributable to the BGC model, PISCES. However, BGC models are coupled to ocean
general circulation systems and the quality of the BGC predictions strongly depends on the
accuracy of the physical properties that control the BGC state variables. In our case, the
dynamical component has been extensively validated (Lellouche et al., 2018, 2013), and
correctly represented variables that are constrained by observations (e. g., temperature and
salinity). However, unconstrained variables in the physical system (e.g., vertical velocities)
can generate unrealistic biases in various biogeochemical variables, especially in the
Equatorial Belt area (Fennel et al., 2019; Park et al., 2018).

32       In addition, BGC-Argo floats are not flawless (Roesler et al., 2017), and in some cases,

the discrepancies observed between the floats and model data do not result from the model
estimations alone. This is particularly true for the BGC-Argo estimates of Chl*a* in the mixed




layer that can be significantly biased due to non-photochemical chlorophyll fluorescence
quenching (Xing et al., 2012) or regional variations in fluorescence of Chl*a* vs Chl*a*
relationship (Roesler et al., 2017).

5        We have restricted the number of diagnostic plots as well the statistical indices to the ones

that are most commonly used in the modelling community. More complex statistical
indicators (Stow et al., 2009) can be computed with the proposed metrics, depending on the
context and the skill level necessary. Likewise, similar or more elaborate diagrams can also be
used, such as Target diagram (Salon et al., 2019), zonal mean diagrams (Doney et al., 2009),
or interannual time series (Doney et al., 2009).

12        The comparison between BGC-Argo data and model simulations is not only beneficial

for the modelling community but also for the BGC-Argo community. Observation System
Simulation Experiments (OSSEs) are generally used to inform, *a priori*, observing network
design (Ford, 2020). Here, we showed that model-observations comparison is, also
informative, *a posteriori,* with respect to the network design, as it highlights sensitive areas
where BGC-Argo observations are critical and where sustained BGC-Argo observations are
required to better constrain the model. It corresponds to the regions where the model
uncertainty (see RMSD spatial maps in Figs. A19-A36) is the highest, i.e., the Equatorial
band with respect to the carbonate system variables, the Southern Ocean with respect to the
nutrients and the DCM variables and the western boundary currents and OMZs with respect to
oxygen.





# 1 Tables

3 **Table 1.** Data mode and QC flags of the BGC-Argo observations used in this study.

| Parameter | Data mode | Date mode of associated pressure, temperature and salinity profiles | QC flags |
|---|---|---|---|
| Chl$a$ | Adjusted and Delayed | Real time, Adjusted and Delayed | • Real time: All flags except 4 <br> • Adjusted or Delayed: All flags except 3 and 4 |
| $O_2$ | Delayed | Delayed | • All flags except 3 and 4 |
| $NO_3$ | Adjusted and Delayed | Real time, Adjusted and Delayed | • Real time: All flags except 4 <br> • Adjusted or Delayed: All flags except 3 and 4 |
| pH | Adjusted and Delayed | Real time, Adjusted and Delayed | • Real time: All flags except 4 <br> • Adjusted or Delayed: All flags except 3 and 4 |
| $b_{bp}$ | Real time and Delayed | Real time, Adjusted and Delayed | • Real time: All flags except 4 <br> • Adjusted or Delayed (P,T,S): All flags except 3 and 4 <br> • Adjusted or Delayed ($b_{bp}$): All flags 4 |





2 **Table 2.** BGC-Argo metrics used to assess the model simulation

| Process | Metric | Definition | units |
|---|---|---|---|
| Air-sea $CO_2$ flux | $spCO_2$ | Depth-averaged $pCO_2$ in the mixed layer | µatm |
| Oceanic pH | $spH$ | Depth-averaged pH in the mixed layer | total |
| | $pH_{200-400}$ | Depth-averaged pH in the 200-400 m layer | total |
| Biological carbon pump | $sChl$ | Depth-averaged Chl$a$ in the mixed layer | mg m$^{-3}$ |
| | $sNO_3$ | Depth-averaged $NO_3$ in the mixed layer | µmol kg$^{-1}$ |
| | $sPO_4$ | Depth-averaged $PO_4$ in the mixed layer | µmol kg$^{-1}$ |
| | $sSi$ | Depth-averaged Si in the mixed layer | µmol kg$^{-1}$ |
| | $sDIC$ | Depth-averaged DIC in the mixed layer | µmol kg$^{-1}$ |
| | $sPOC$ | Depth-averaged POC in the mixed layer | mg m$^{-3}$ |
| | $POC_{meso}$ | Depth-averaged POC in the mesopelagic layer | mg m$^{-3}$ |
| | $Chl_{DCM}$ | Magnitude of DCM | mg m$^{-3}$ |
| | $H_{DCM}$ | Depth of DCM | m |
| | $H_{nit}$ | Depth of nitracline | m |
| Oxygen levels and OMZs | $sO_2$ | Depth-averaged $O_2$ in the lixed layer | µmol kg$^{-1}$ |
| | $O_{2\ 300}$ | $O_2$ at 300 m | µmol kg$^{-1}$ |
| | $O_{2\ 1000}$ | $O_2$ at 1000 m | µmol kg$^{-1}$ |
| | $O_{2min}$ | value of $O_2$ minimum | µmol kg$^{-1}$ |
| | $H_{O2min}$ | Depth of $O_2$ minimum | m |





**Figures**

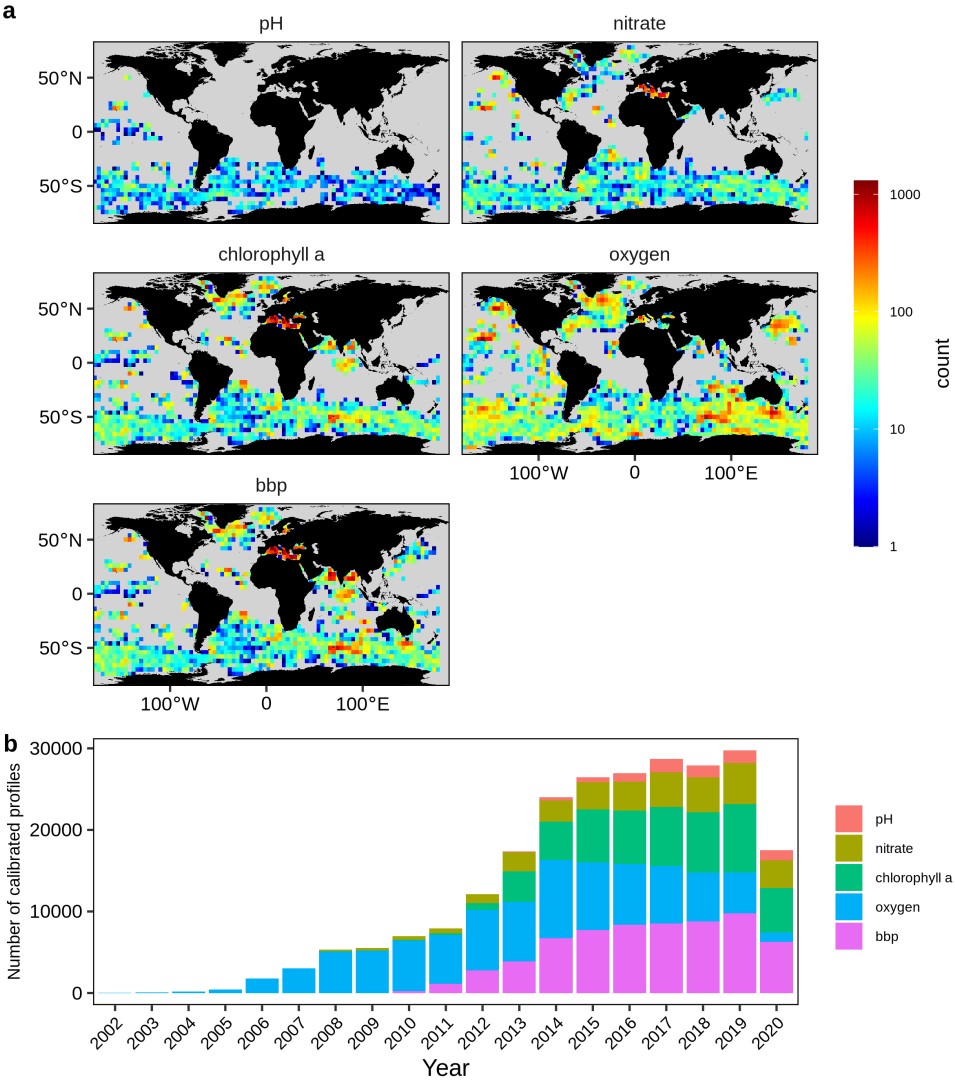

**Figure 1.** Spatial and temporal coverage of quality-controlled BGC-Argo pH, $NO_3^-$, Chl*a*, $O_2$,
and $b_{bp}$ profiles. **(a)** Number of quality-controlled profiles for the entire period per 4°x4° bin.
**(b)** Number of quality-controlled profiles per year.





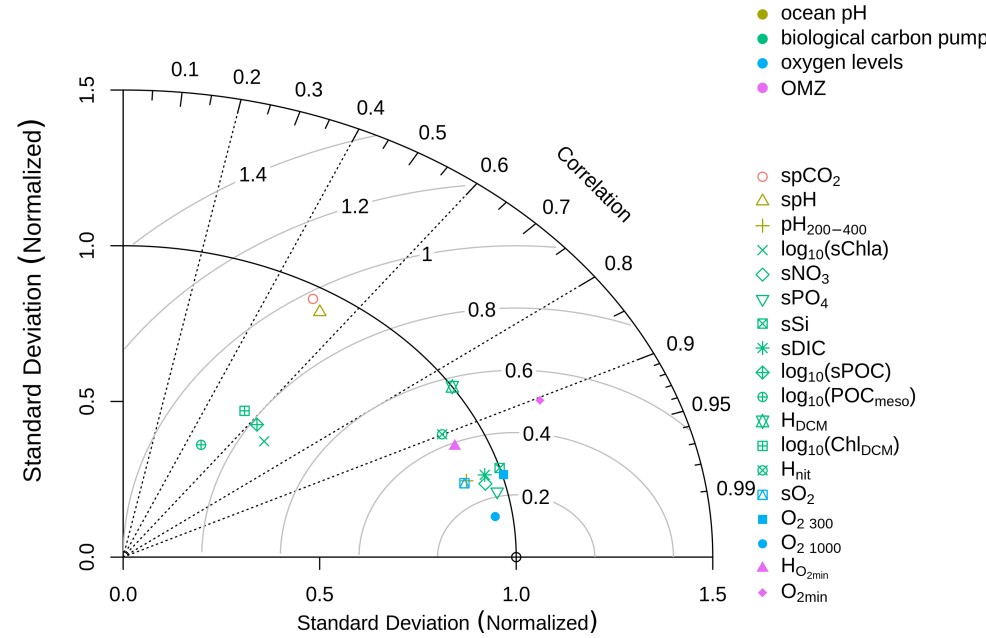

**Figure 2.** Comparison of BGC-Argo floats' observations and model values for all metrics
using Taylor diagram. The symbols correspond to the metrics and the colours represent the
BGC processes with which they are associated. Note that the metrics calculated from the float
pH and $NO_3$ used both the direct observations of the floats and as well as the estimations from
CANYON-B. The metrics related to Chl*a* and POC, namely sChl, $Chl_{DCM}$, sPOC, $POC_{meso}$
were $log_{10}$-transformed because they cover several orders of magnitude and they are
lognormally distributed. Observed DCMs and nitracline deeper than 250 m are not included.





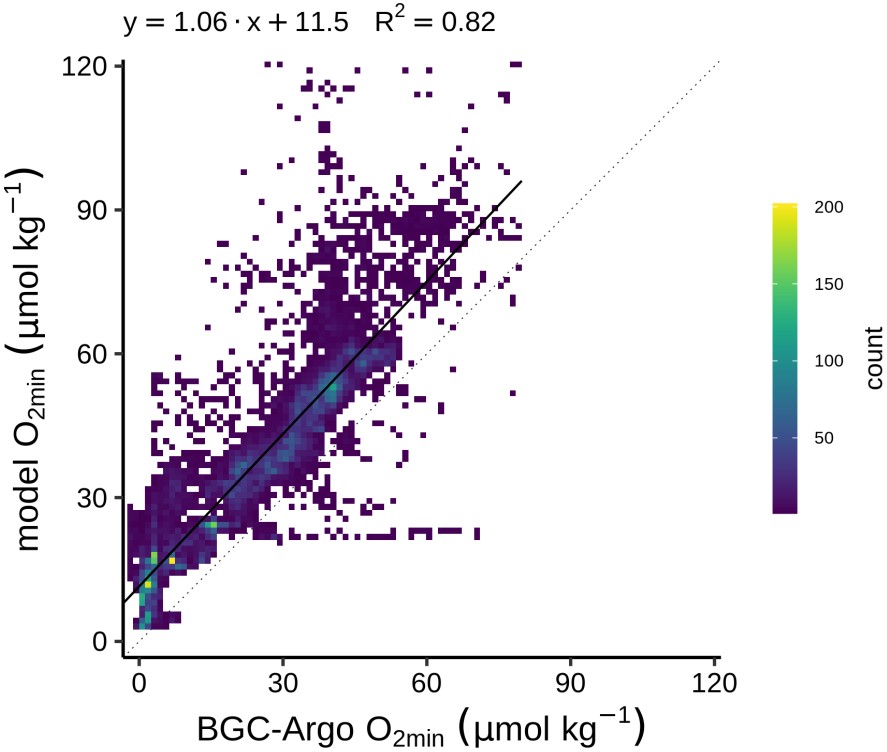

**Figure 3.** Density plots of BGC-Argo floats' observations and model $O_{2min}$ . Each axis is

divided in 100 bins and the colour represents the number of points in each bin. The dashed

line represents the 1:1 line. The plain line represents the linear regression line between the

two data sets**.** The coefficients of the linear regression line (gain and offset) as well the

coefficient of determination ($R^2$) are indicated on the top of the plot.


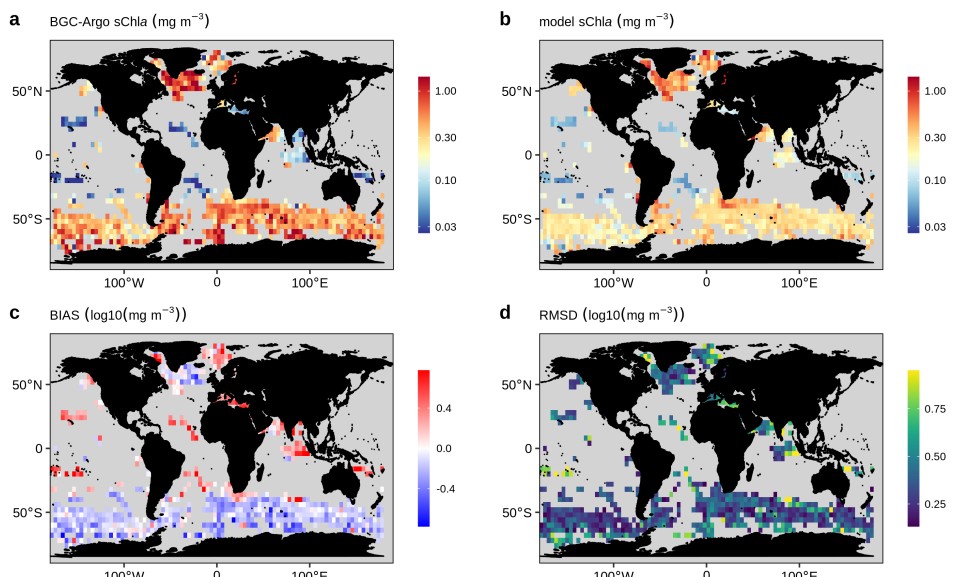

**Figure 4.** Spatial distribution maps of BGC-Argo floats' observations of sChl **(a),** model sChl

**(b)**, the BIAS **(c)** and the RMSD **(d)**. The data are averaged in 4°x4° bins. Bins containing

less than 4 points are excluded. The BIAS and RMSD are computed on the $\log_{10}$-transformed

data to account that sChl covers several orders of magnitude and is lognormally distributed

(Campbell, 1995).





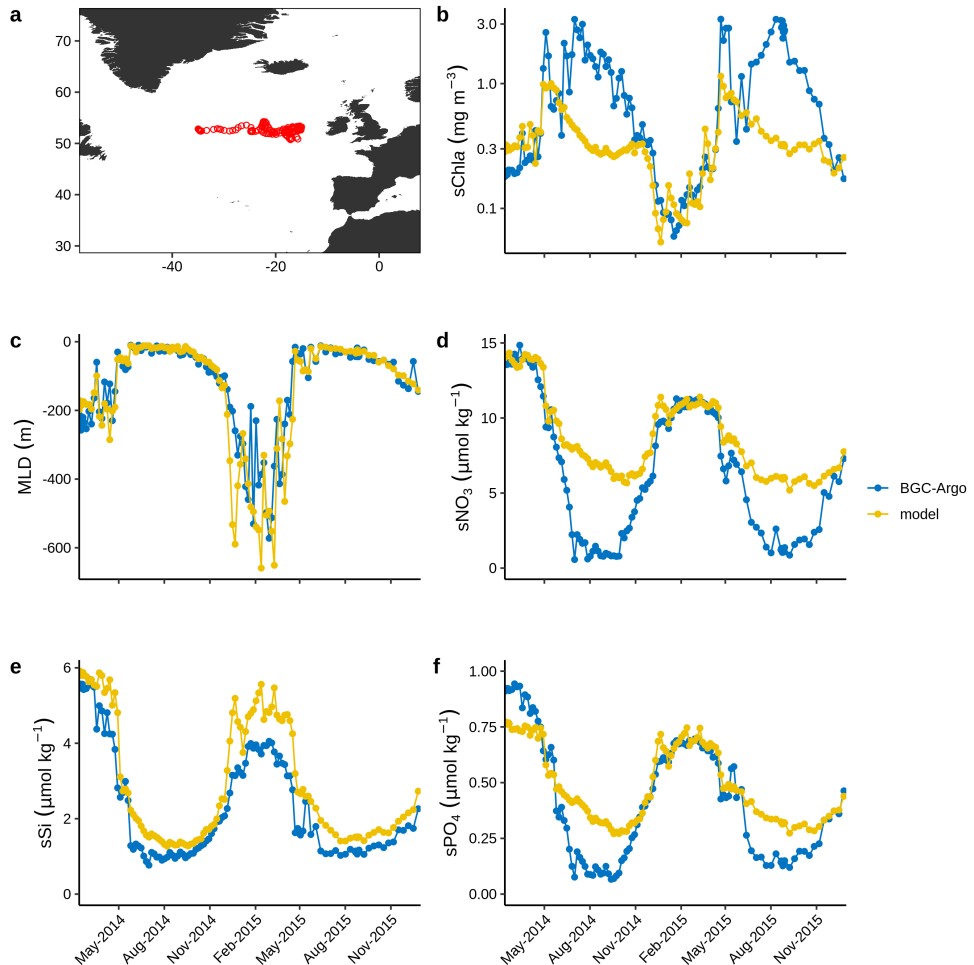

**Figure 5.** (**a**) Float trajectory of the BGC-Argo float (WMO number: 5904479). 2014-2015
time series of (**b**), mixed layer depth, (**c**), sChl, (**d**), $sNO_3$, (**c**), sSi , (**f**), $sPO_4$ , derived from
the BGC-Argo floats observations (blue) and from the model simulation (yellow). The float
sChl and $sNO_3$ are calculated from the direct observations of the floats, whereas the float sSi
and $sPO_4$ result from CANYON-B predictions.



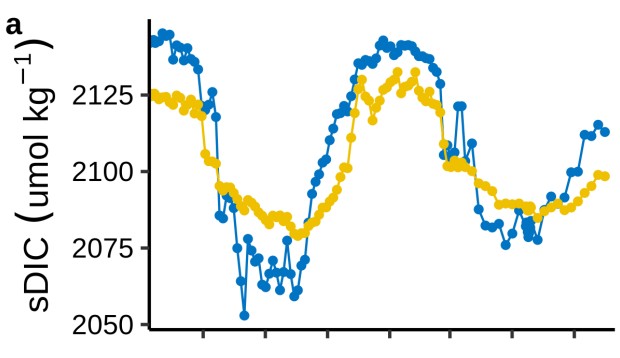

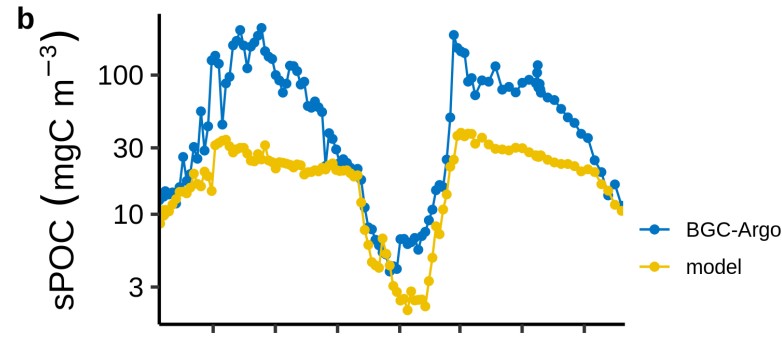

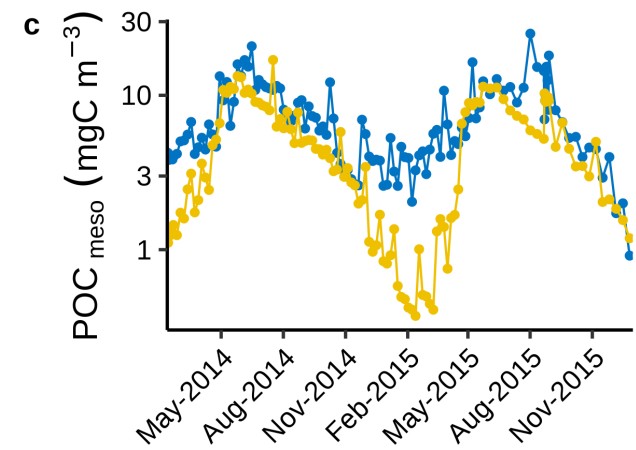

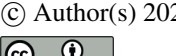



**Figure 6.** Same as Fig. 5 but for **(a)**, sDIC, **(b)**, sPOC, **(c)**, $POC_{meso}$. The float sPOC and
$POC_{meso}$ are calculated from the direct observations of the floats, whereas the float sDIC
result from CANYON-B predictions.





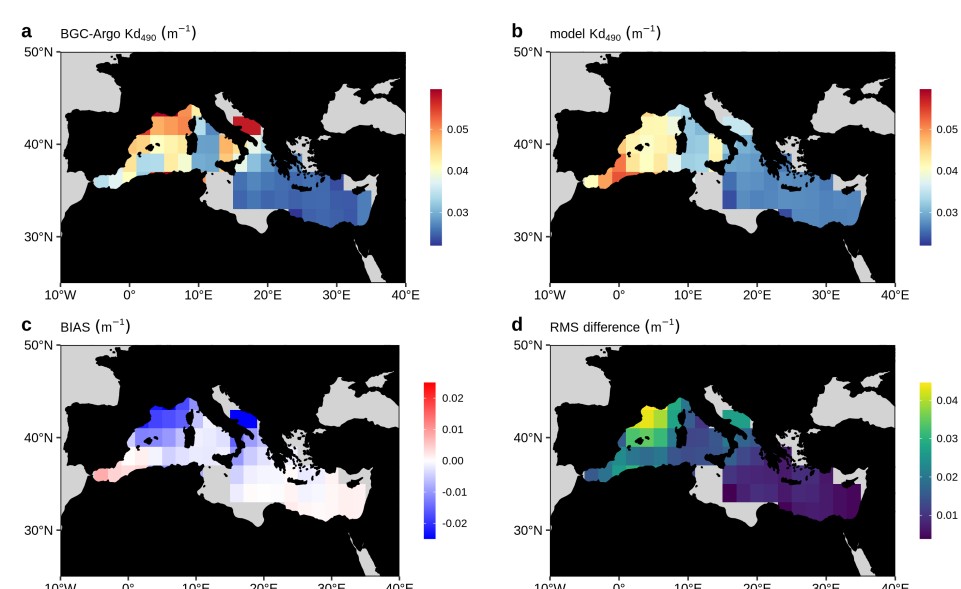

**Figure. 7 .** Spatial distribution maps of BGC-Argo floats' observations $K_d$ at 490 nm **(a),**

modelled $K_d$ at 490 nm from the Mediterranean BGC model **(b)**, the BIAS **(c)** and the RMSD

**(d)**. The data are averaged in 2°x2° bins. Bins containing less than 4 points are excluded.



# 1 Appendix

## 3 A.1 The CMEMS global hydrodynamic-biogeochemical model

The model used in this study features the offline coupled NEMO–PISCES model, with
a 1/4° horizontal resolution 50 vertical levels (with 22 levels in the upper 100 m, the vertical
resolution is 1m near the surface and decreases to 450m resolution near the bottom) and daily
temporal resolution, covering the period from 2009 to 2017.
The biogeochemical model PISCES v2 (Aumont et al., 2015) is a model of
intermediate complexity designed for global ocean applications, and is part of NEMO
modelling platform. It features 24 prognostic variables and includes five nutrients that limit
phytoplankton growth (nitrate, ammonium, phosphate, silicate and iron) and four living
compartments: two phytoplankton size classes (nanophytoplankton and diatoms, resp. small
and large) and two zooplankton size classes (microzooplankton and mesozooplankton, resp.
small and large); the bacterial pool is not explicitly modelled. PISCES distinguishes three
non-living detrital pools for organic carbon, particles of calcium carbonate and biogenic
silicate. Additionally, the model simulates the carbonate system and dissolved oxygen.
PISCES has been successfully used in a variety of biogeochemical studies, both at regional
and global scale (Bopp et al., 2005; Gehlen et al., 2006, 2007; Gutknecht et al., 2019; Lefèvre
et al., 2019; Schneider et al., 2008; Séférian et al., 2013; Steinacher et al., 2010; Tagliabue et
al., 2010).
The dynamical component is the latest Mercator Ocean global 1/12° high-resolution
ocean model system, extensively described and validated in Lellouche et al. (2018, 2013).
This system provides daily and 1/4°-coarsened fields of horizontal and vertical current
velocities, vertical eddy diffusivity, mixed layer depth, sea ice fraction, potential temperature,
salinity, sea surface height, surface wind speed, freshwater fluxes and net surface solar
shortwave irradiance that drive the transport of biogeochemical tracers. This system also
features a reduced-order Kalman filter based on the Singular Evolutive Extended Kalman
filter (SEEK) formulation introduced by Pham et al. (1998), that assimilates, on a 7-day
assimilation cycle, along-track altimeter data, satellite Sea Surface Temperature and Sea-Ice





Concentration from OSTIA, and *in situ* temperature and salinity vertical profiles from the
CORA 4.2 in situ database.

4        In addition, the biogeochemical component of the coupled system also embeds a

reduced order Kalman filter (similar to the above mentioned) that operationally assimilates
daily L4 remotely sensed surface chlorophyll
(https://resources.marine.copernicus.eu/documents/QUID/CMEMS-GLO-QUID-001-
028.pdf). In parallel, a climatological-damping is applied to nitrate, phosphate, oxygen,
silicate - with World Ocean Atlas 2013 - to dissolved inorganic carbon and alkalinity – with
GLODAPv2 climatology (Key et al., 2015) - and to dissolved organic carbon and iron - with a
4000-year PISCES climatological run. This relaxation is set to mitigate the impact of the
physical data assimilation in the offline coupled hydrodynamic-biogeochemical system,
leading significant rises of nutrients in the Equatorial Belt area, and resulting in an unrealistic
drift of various biogeochemical variables e.g. chlorophyll, nitrate, phosphate (Fennel et al.,
2019; Park et al., 2018). The time-scale associated with this climatological damping is set to 1
year and allows a smooth constraint that has been shown to be efficient to reduce the model
drift.
**A.2  The Mediterranean Sea biogeochemical model MedBFM**

21       The Mediterranean Sea biogeochemical model MedBFM, is based on the system

described in Teruzzi et al. (2014) and Salon et al. (2019).

24       The physical forcing fields needed to compute the transport include the 3-d horizontal

and vertical current velocities, vertical eddy diffusivity, potential temperature, and salinity and
2-d data surface data for wind stress. These forcing datasets are simulated by the Mediterranean
Sea Monitoring and Forecasting Centre (MED–MFC) in the Copernicus Marine Environmental
Monitoring Service (CMEMS, http://marine.copernicus.eu). The biogeochemical model is then
offline forced adopting the output computed by the CMEMS MED-MFC. In the present
application, we switched off the biogeochemical assimilation scheme that is currently used in
the operational MED-MFC system.





The light propagation is resolved coupling an atmospheric multispectral radiative
transfer model (Lazzari et al., 2020) with an in-water radiative model (Dutkiewicz et al., 2015)
featuring bands at 25 nm resolution in the UV and visible wavelengths.
The horizontal resolution is approximately 6 km and there are 72 vertical levels with 3
m resolution at surface coarsening at 300 m for the deeper layers. The biogeochemical model
here adopted (Biogeochemical Flux Model -- BFM -- ; (Vichi et al., 2015)) has been already
applied to simulate primary producers biogeochemistry (Lazzari et al., 2012), alkalinity spatial
and temporal variability (Cossarini et al., 2015), and $CO_2$ fluxes (Canu et al., 2015) for the
Mediterranean Sea, and has been corroborated using *in situ* data for the operational purposes
within CMEMS (Salon et al., 2019). The BFM model has been expanded in the present
configuration adding the dynamics of coloured dissolved organic carbon (CDOM) by assuming
a constant CDOM:DOC production ratio (i.e. 2%, as in (Dutkiewicz et al., 2015)). The
absorption of CDOM, is described using reference absorption at 450 nm of 0.015 m2/mgC
(Dutkiewicz et al., 2015) and an exponential slope of 0.017 $nm^{-1}$ (Babin et al., 2003; Organelli
et al., 2014).
**A.3 BGC-Argo $K_d$ estimates**
The data used to compute the $K_d$ metrics are quality checked according to Organelli et
al. (2017). Moreover, for the $K_d$ logarithmic interpolation, the following selection rules were
applied: the profile must have at least 5 BGC Argo float sampling in the first optical depth, the
gap between the two shallower acquisitions must be less than 10 meters, and there must be at
least one measurement deeper than 15 meters.
**A.4 Figures**



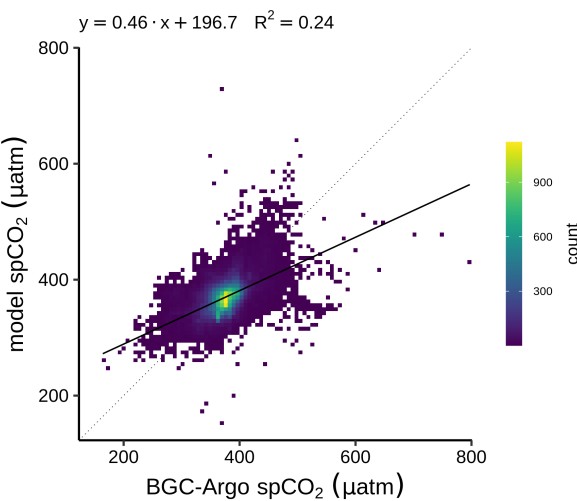

2 **Figure A1.** Same as Figure 3 but for spCO$_2$.

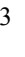

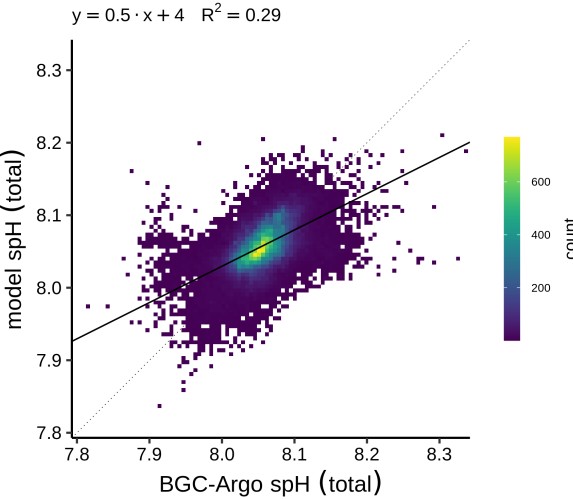

5 **Figure A2.** Same as Figure 3 but for spH. Note that spH is calculated from both the direct

6 observations of the floats and as well as the estimations from CANYON-B.



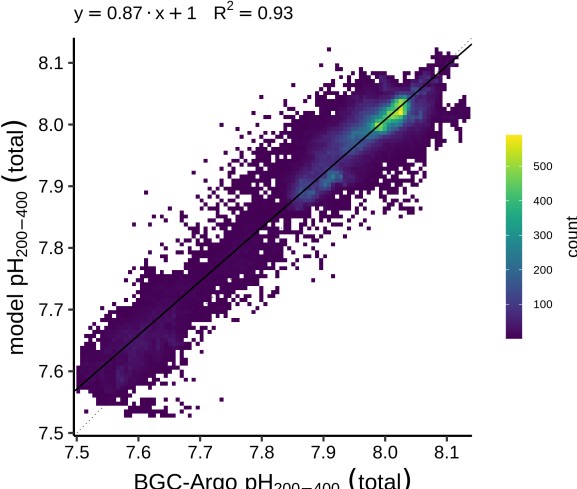

3 **Figure A3.** Same as Figure 3 but for $pH_{200\text{-}400}$. Note that $pH_{200\text{-}400}$ is calculated from both the

4 direct observations of the floats and as well as the estimations from CANYON-B.

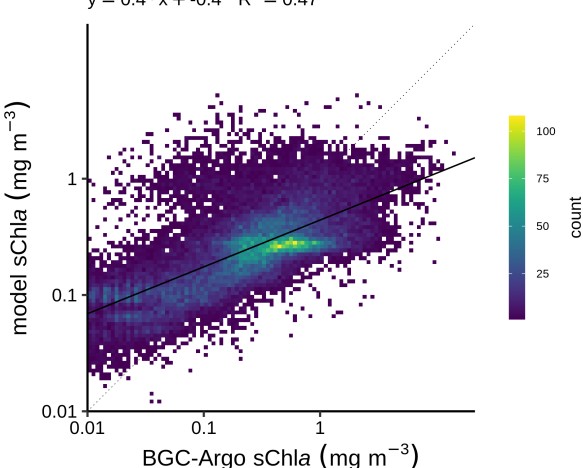



**Figure A4.** Same as Figure 3 but for sChl. Note that the least squares regression is computed
on the $\log_{10}$-transformed data to account that sChl covers several orders of magnitude and it is
lognormally distributed (Campbell, 1995). Data lower than 0.01 mg m$^{-3}$ are not included.

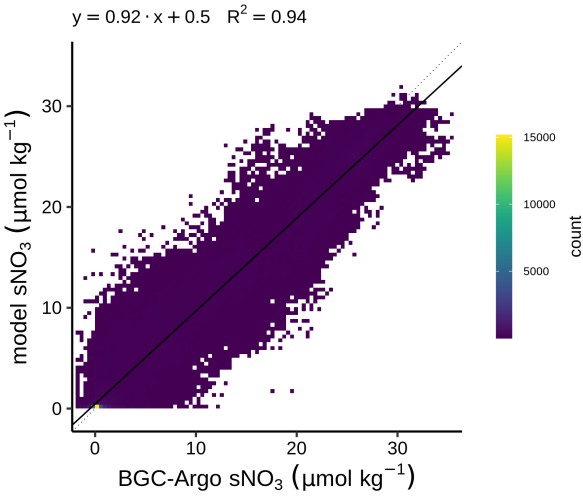

**Figure A5.** Same as Figure 3 but for sNO$_3$. Note that sNO$_3$ is calculated from both the direct
observations of the floats and as well as the estimations from CANYON-B.

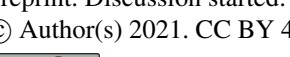


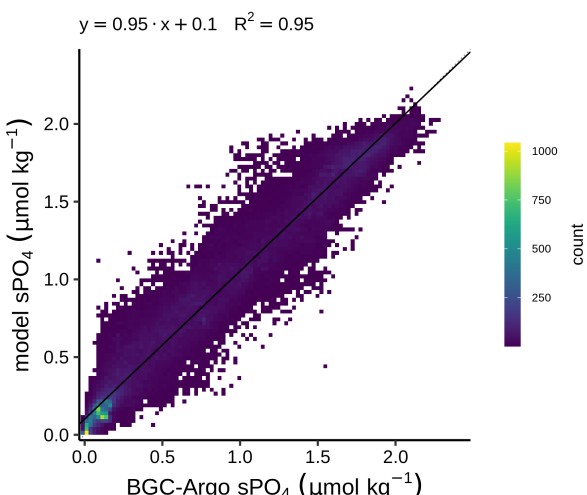

2    **Figure A6.** Same as Figure 3 but for sPO$_4$.

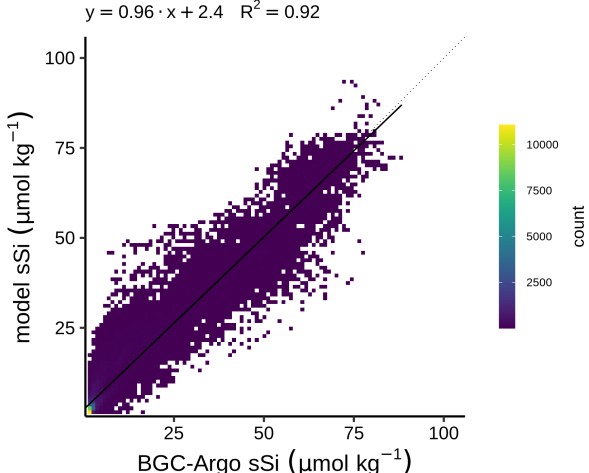

5    **Figure A7.** Same as Figure 3 but for sSi.



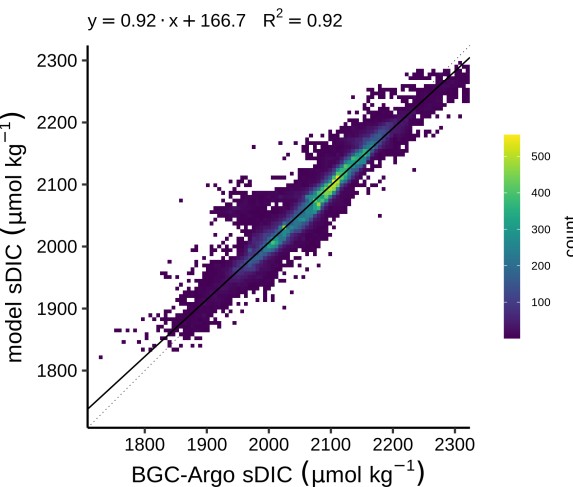

2 **Figure A8.** Same as Figure 3 but for sDIC.

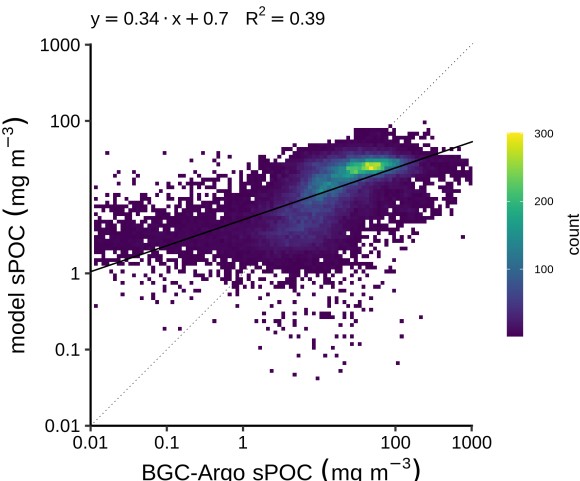

6 **Figure A9.** Same as Figure 3 but for sPOC. Note that the least squares regression is

7 computed on the $\log_{10}$-transformed data to account that sPOC covers several orders of



magnitude and it is lognormally distributed (Campbell, 1995). Data lower than 0.01 mg m$^{-3}$
are not included.



**Figure A10.** Same as Figure 3 but for POC$_{meso}$. Note that the least squares regression is
computed on the log$_{10}$-transformed data to account that POC$_{meso}$ covers several orders of
magnitude and it is lognormally distributed (Campbell, 1995). Data lower than 0.01 mg m$^{-3}$
are not included.



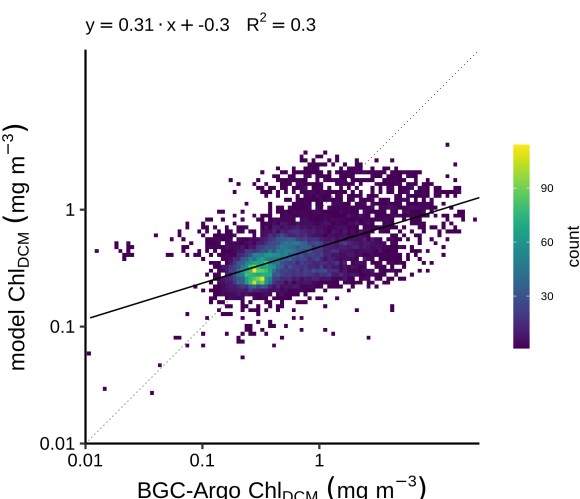

**Figure A11.** Same as Figure 3 but for $Chl_{DCM}$. Note that the least squares regression is
computed on the $\log_{10}$-transformed data to account that $Chl_{DCM}$ covers several orders of
magnitude and it is lognormally distributed (Campbell, 1995). Data lower than 0.01 mg m$^{-3}$
are not included. Observed DCMs deeper than 250 m are not included.





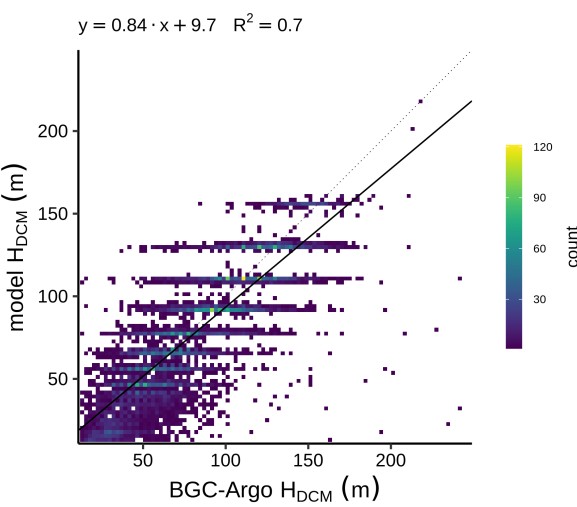

**Figure A12.** Same as Figure 3 but for $H_{DCM}$. Observed DCMs deeper than 250 m are not
included.

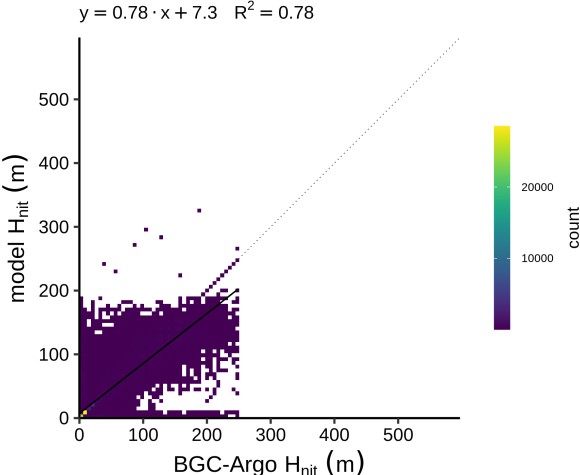

**Figure A13.** Same as Figure 3 but for $H_{nit}$. Observed nitracline deeper than 250 m are not
included.



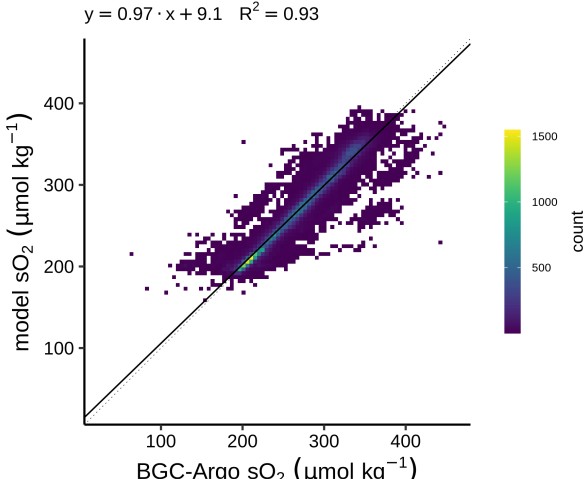

3     **Figure A14.** Same as Figure 3 but for $sO_2$.

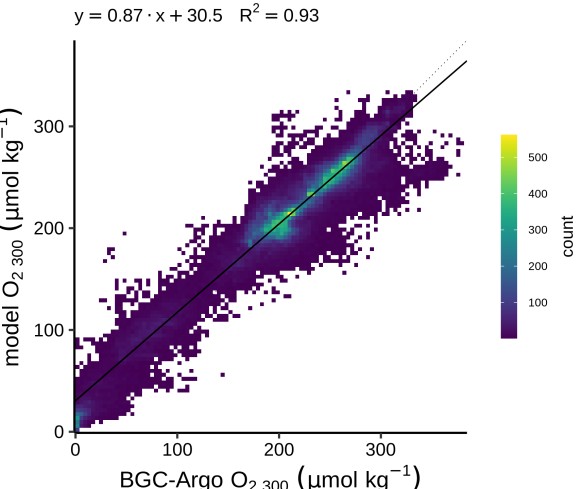

6     **Figure A15.** Same as Figure 3 but for $O_{2\ 300}$.

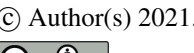



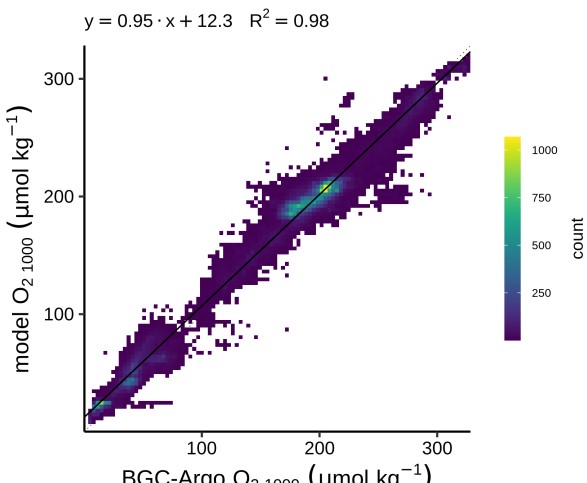

2    **Figure A16.** Same as Figure 3 but for $O_{2\ 1000}$.

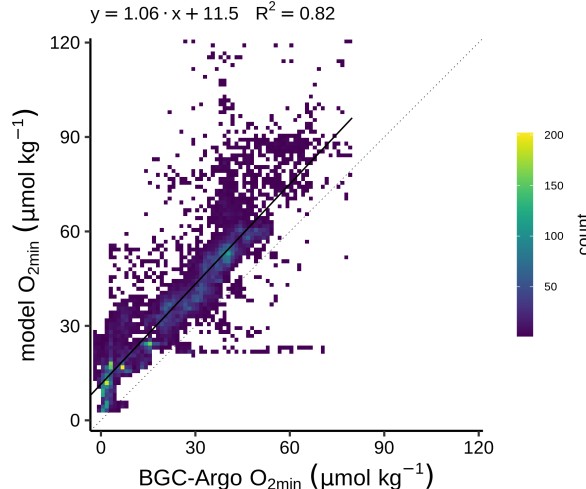

5    **Figure A17.** Same as Figure 3..





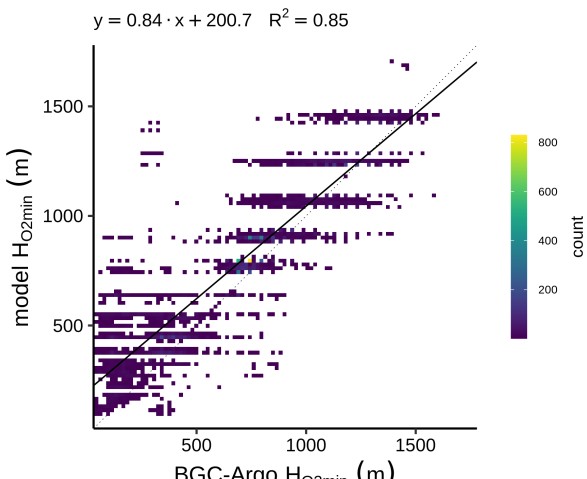

2   **Figure A18.** Same as Figure 3 but for $H_{O2min}$.



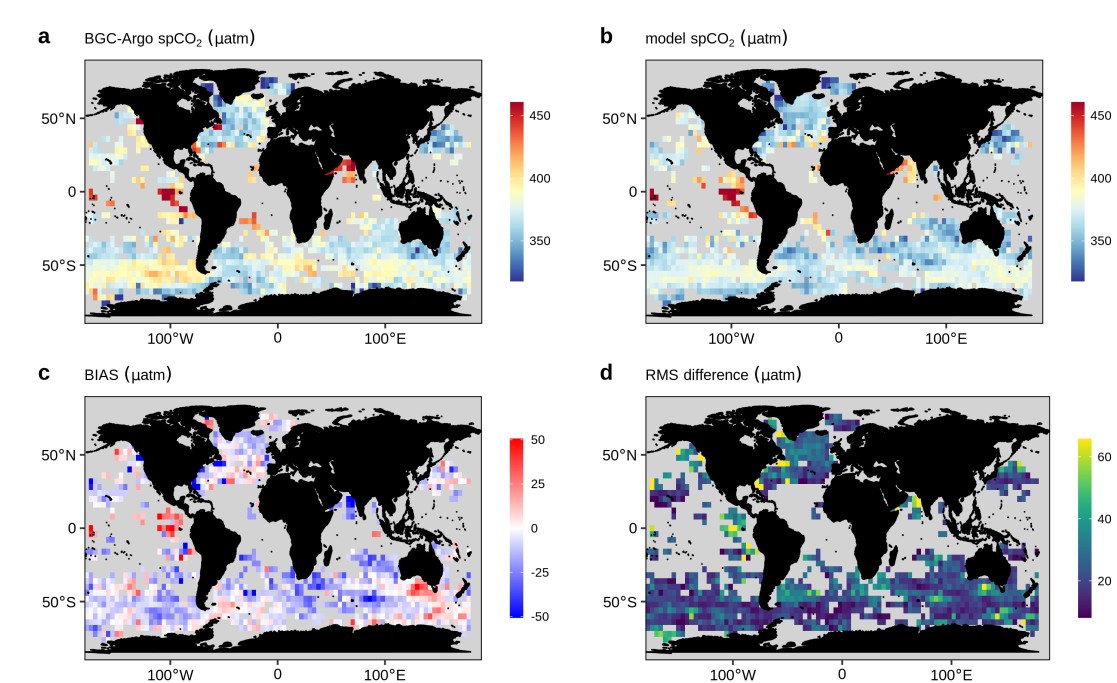

3   **Figure A19.** Same as Figure 4 but for spCO$_2$.





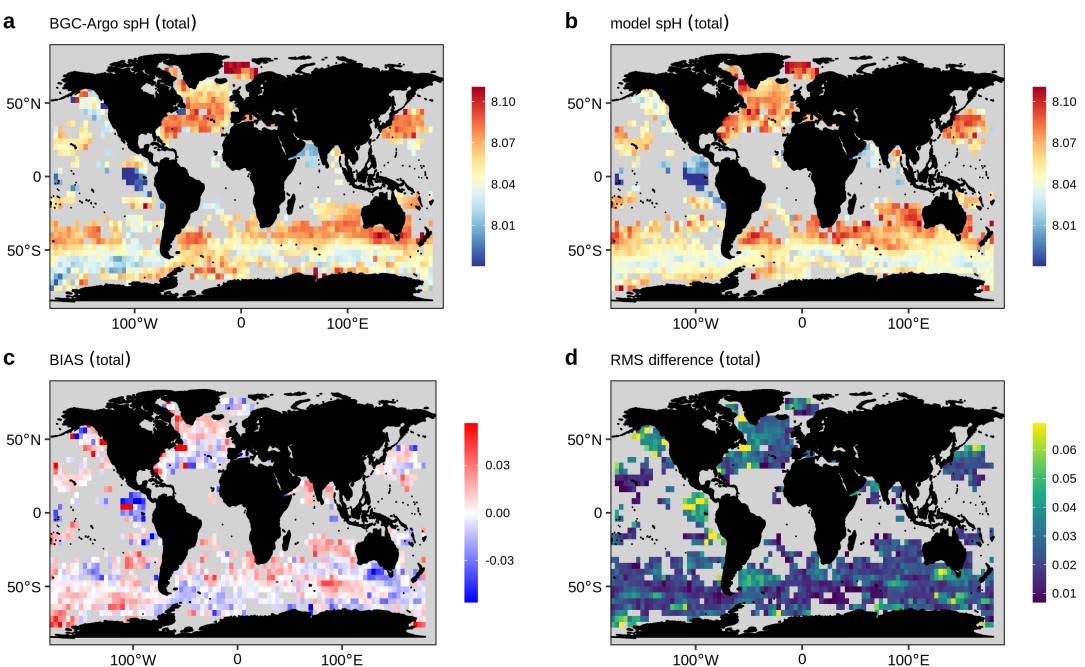

**Figure A20.** Same as Figure 4 but for spH. Note that spH is calculated from both the direct

observations of the floats and as well as the estimations from CANYON-B.

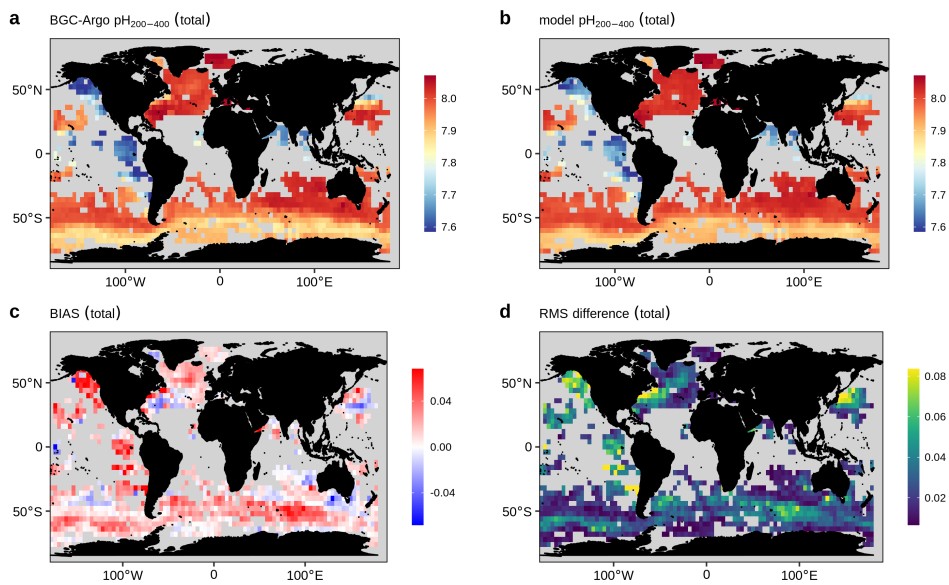

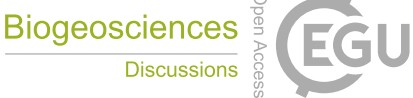

1   **Figure A21.** Same as Figure 4 but for $pH_{200-400}$. Note that $pH_{200-400}$ is calculated from both the

2   direct observations of the floats and as well as the estimations from CANYON-B.

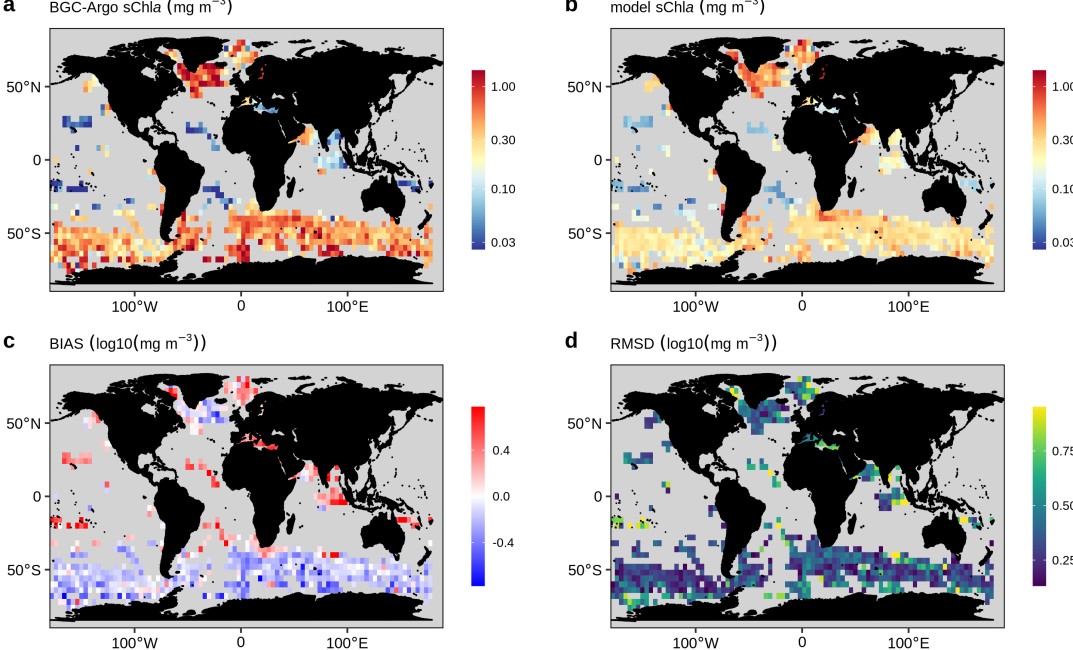

6   **Figure A22.** Same as Figure 4.




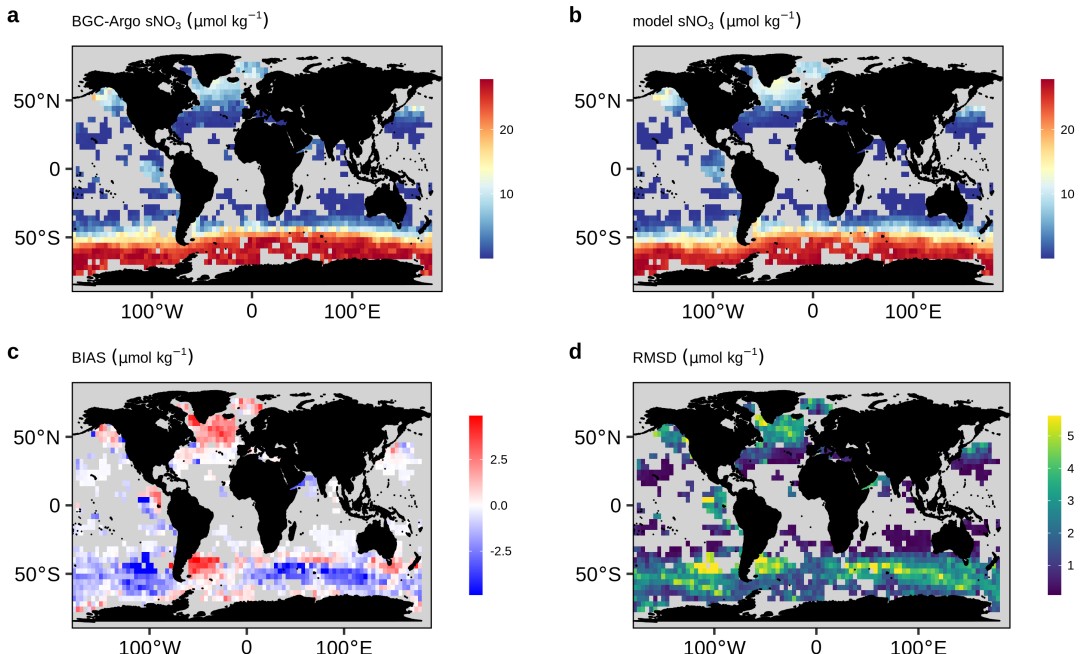

**Figure A23.** Same as Figure 4 but for sNO$_3$. Note that sNO$_3$ is calculated from both the direct observations of the floats and as well as the estimations from CANYON-B.



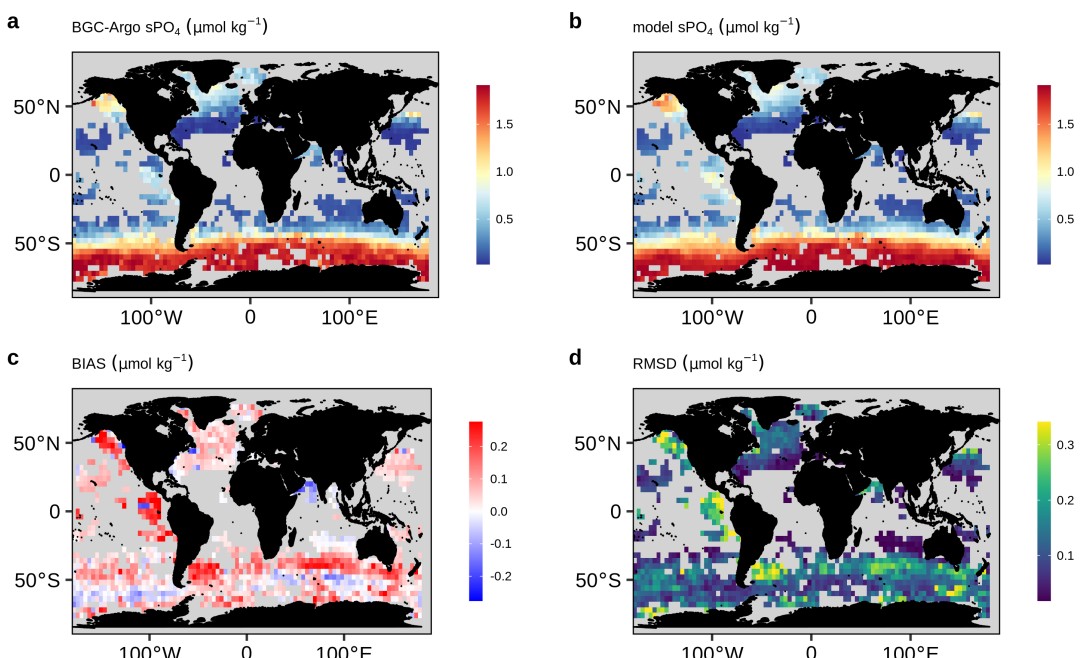

2 **Figure A24.** Same as Figure 4 but for sPO$_4$.



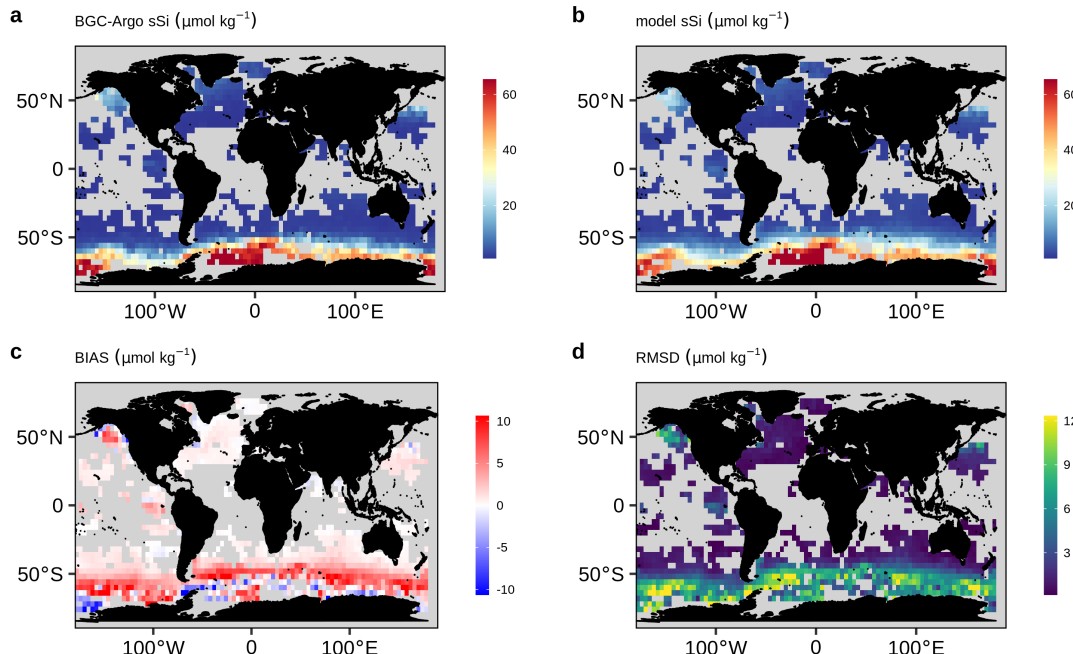

2 **Figure A25.** Same as Figure 4 but for sSi.



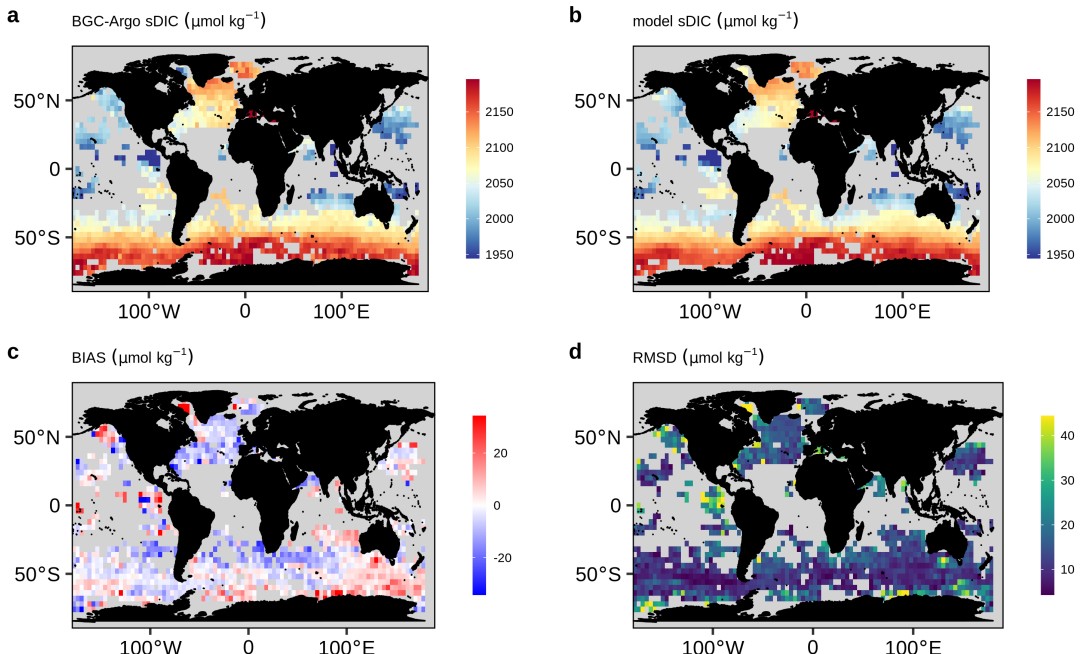

**Figure A26.** Same as Figure 4 but for sDIC.





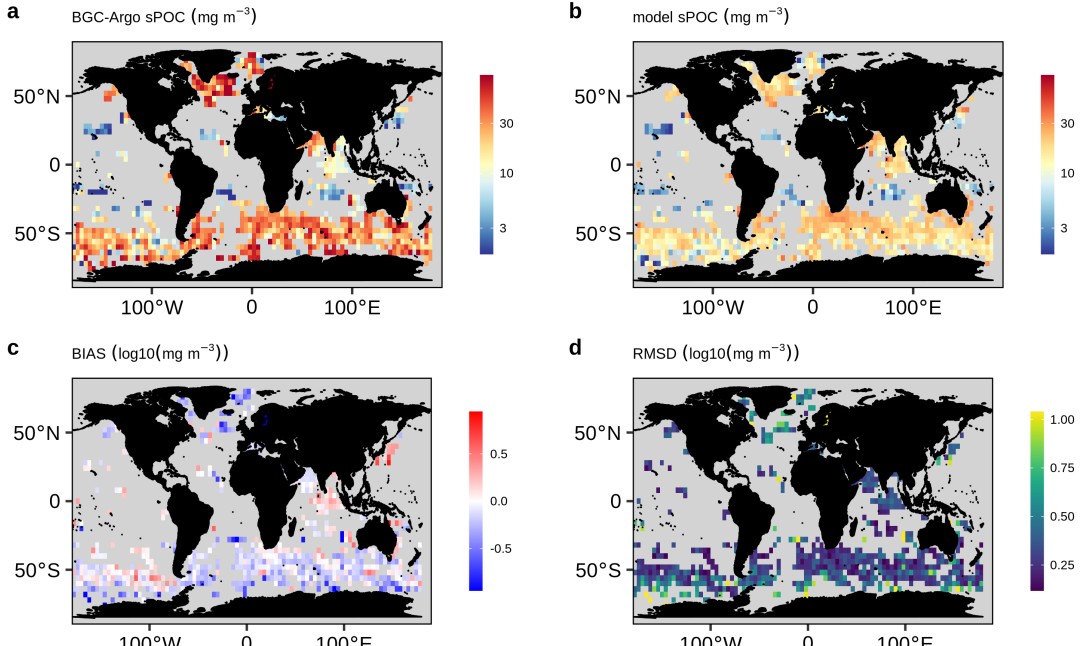

**Figure A27.** Same as Figure 4 but for sPOC. The BIAS and RMSD are computed on the

log$_{10}$-transformed data to account that sPOC covers several orders of magnitude and it is

lognormally distributed (Campbell, 1995)





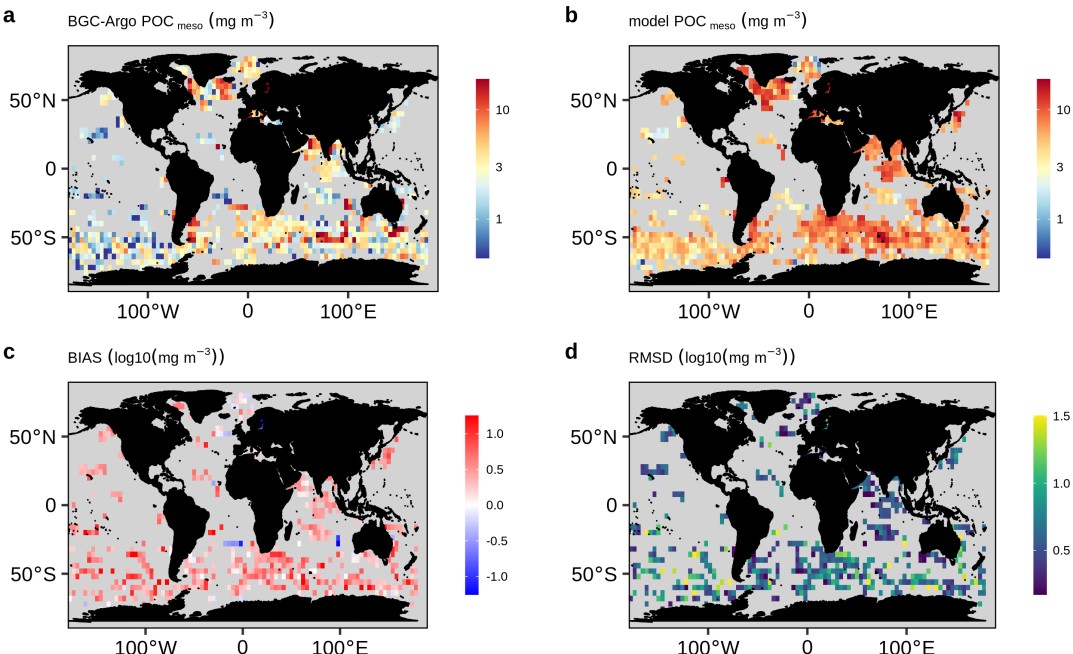

2 **Figure A28.** Same as Figure 4 but for POC$_{meso}$. The BIAS and RMSD are computed on the

3 log$_{10}$-transformed data to account that POC$_{meso}$ covers several orders of magnitude and it is

4 lognormally distributed (Campbell, 1995)


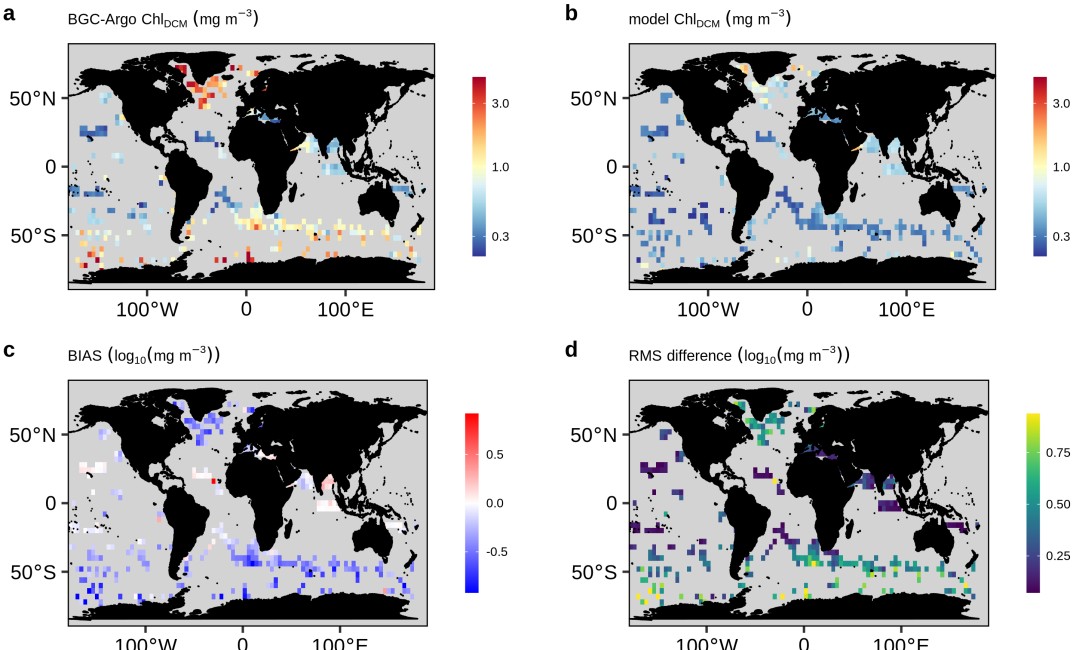

**Figure A29.** Same as Figure 4 but for Chl$_{DCM}$. Note that the BIAS and RMSD are computed on the log$_{10}$-transformed data to account that Chl$_{DCM}$ covers several orders of magnitude and it is lognormally distributed (Campbell, 1995).





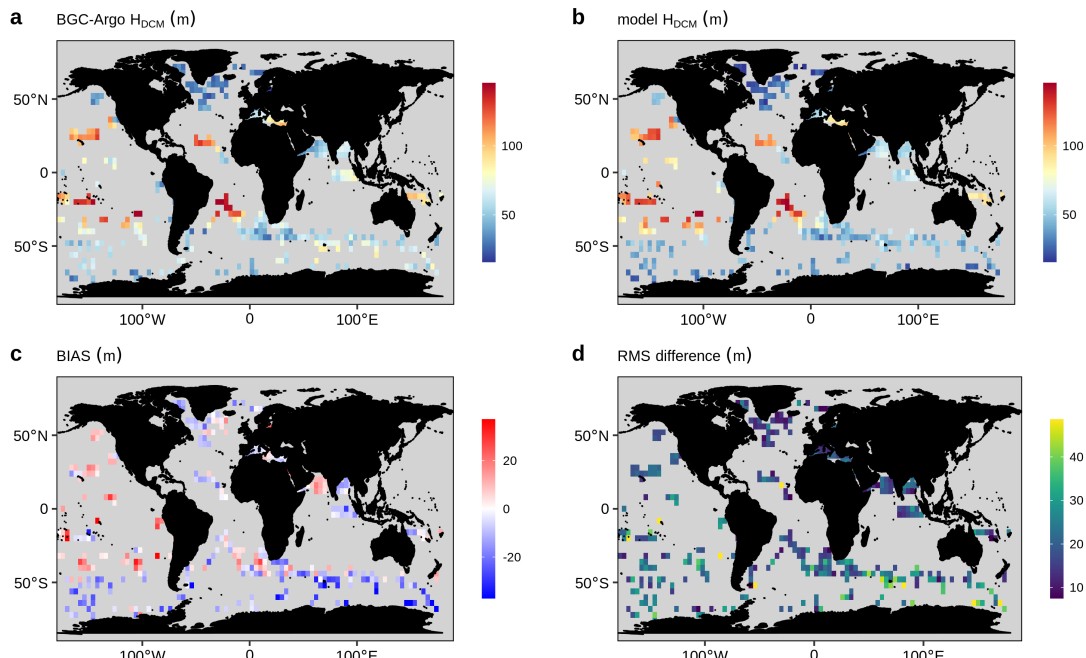

**Figure A30.** Same as Figure 4 but for $H_{DCM}$. Observed DCMs deeper than 250 m are not

included.





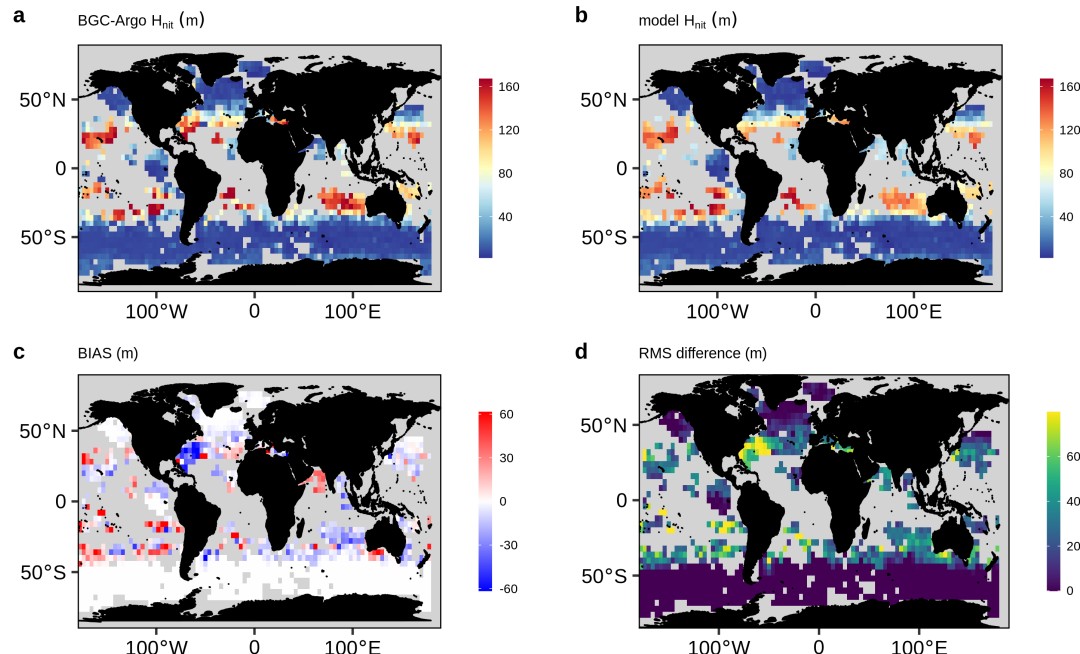

**Figure A31.** Same as Figure 4 but for $H_{nit}$. Observed nitracline deeper than 250 m are not

included.





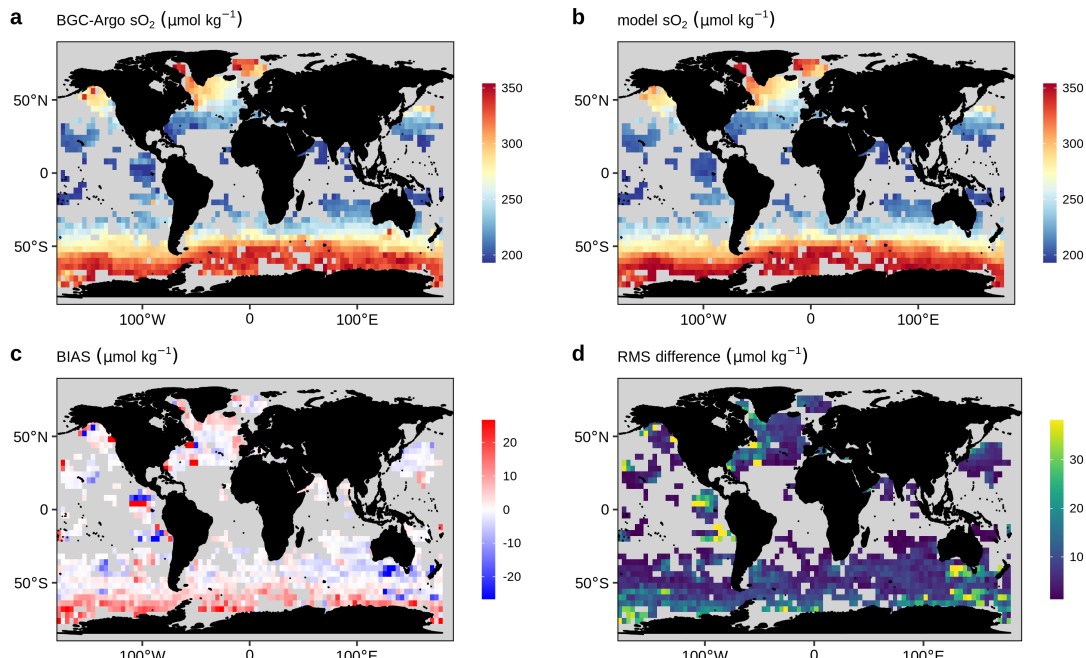

**Figure A32.** Same as Figure 4 but for sO$_2$.





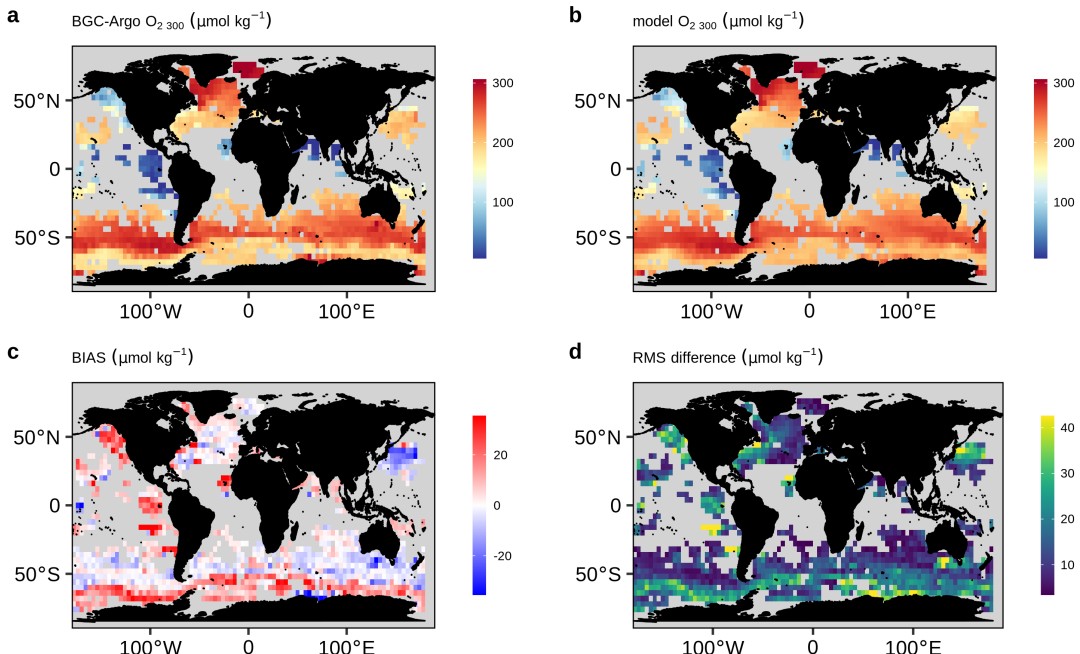

**Figure A33.** Same as Figure 4 but for $O_{2\,300}$.





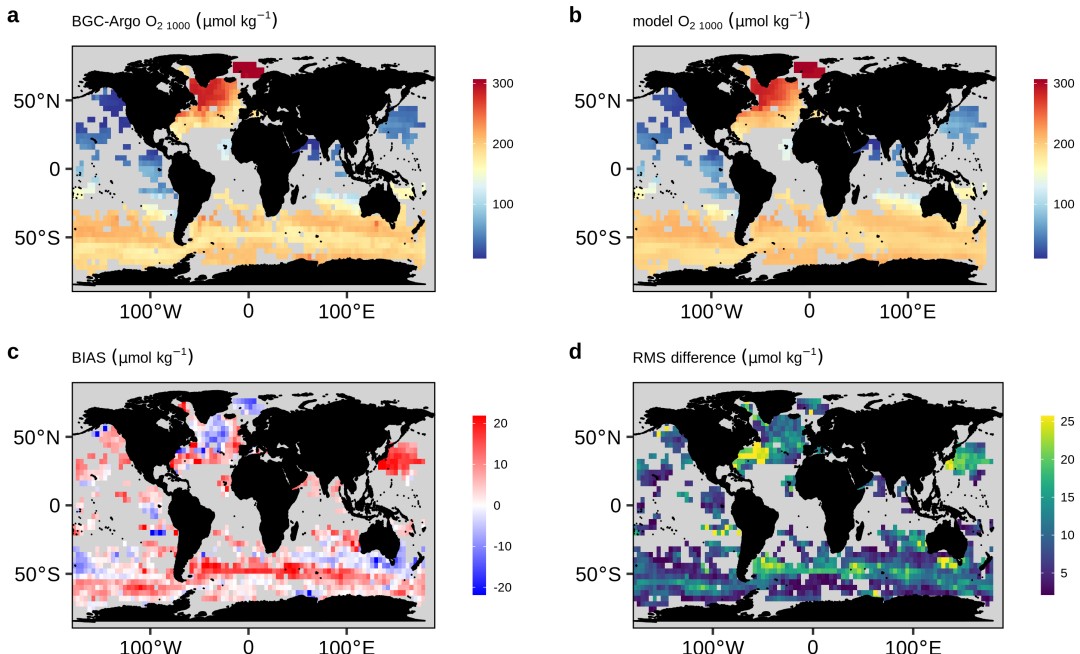

2  **Figure A34.** Same as Figure 4 but for $O_{2\ 1000}$.





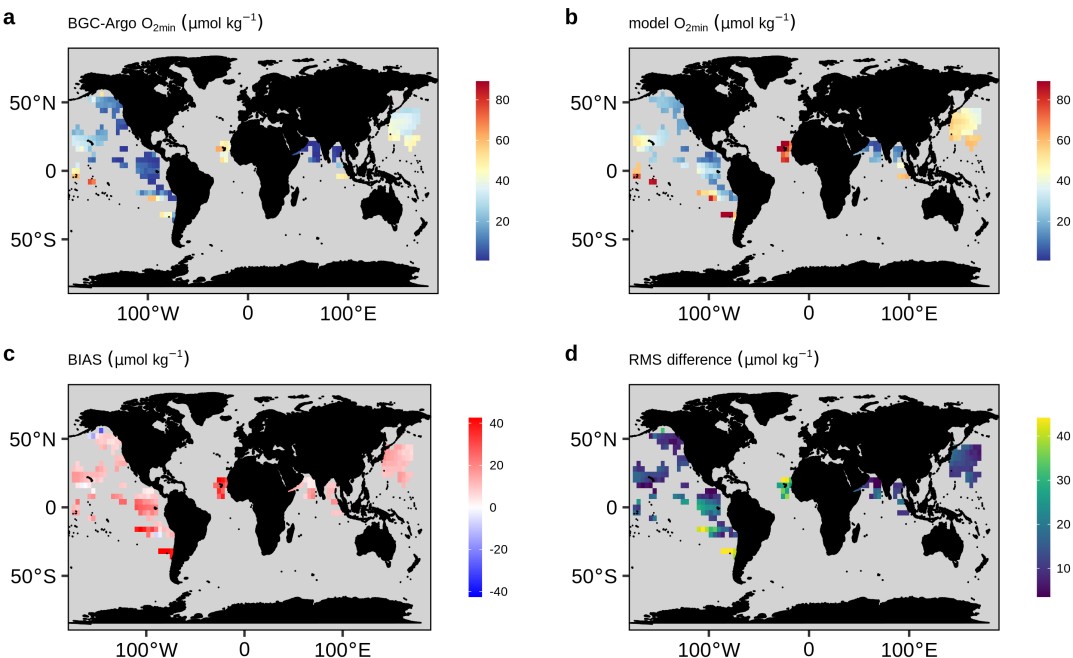

2    **Figure A35.** Same as Figure 4 but for $O_{2min}$.

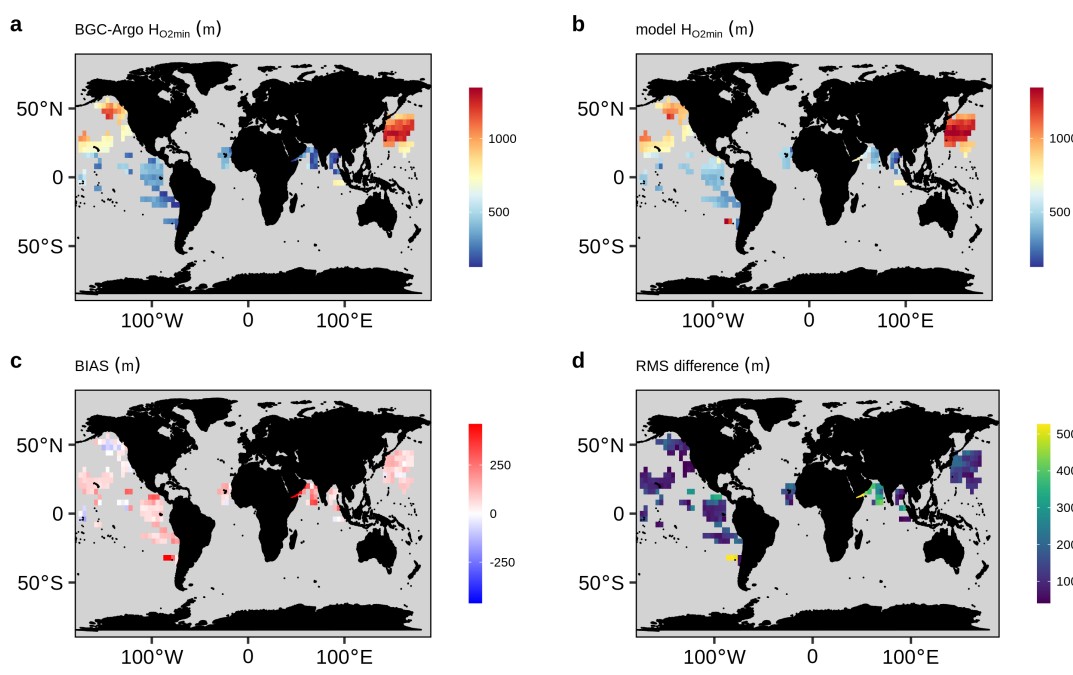

5    **Figure A36.** Same as Figure 4 but for $H_{O2min}$.

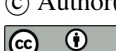



**Data availability**. The BGC model data can be downloaded from the Copernicus Marine
Environmental Monitoring Service
([https://resources.marine.copernicus.eu/?option=com_csw&view=details&product_id=GLOB](https://resources.marine.copernicus.eu/?option=com_csw&view=details&product_id=GLOB)
AL_ANALYSIS_FORECAST_BIO_001_028). The BGC-Argo data were downloaded from
the Argo Global Data Assembly Centre in France (ftp://ftp.ifremer.fr/argo/).
**Authors Contribution**: AM, GC, FD, SS and VT originated the study. AM, HC, FD, RS and
VT designated the study. AM and RS process the BGC-Argo floats data. PL processed the
BGC-Argo float in the Mediterranean Sea and run the Mediterranean BGC model.AM
analysed the data. AM wrote the first draft of the manuscript. HC, GC, FD, EG, PL, CP,
SS,RS,VT and AT contributed to the subsequent drafts. All authors read and approved the
final draft.
**Competing Interests:** The authors declare no competing financial interests.
**Materials and correspondence:** Correspondence and request for material should be
addressed to mignot@mercator-ocean.fr
**Acknowledgements:** This study has been conducted using the Copernicus Marine Service
products (CMEMS). The BGC-Argo data were collected and made freely available by the
International Argo program and the national programs that contribute to it
(https://www.argo.jcommops. org). The Argo program is part of the Global Ocean Observing
System. Part of this work was performed within the framework of the BIOOPTIMOD and
MASSIMILI CMEMS Service Evolution Projects.  This paper represents a contribution to the
following research projects: NAOS (funded by the Agence Nationale de la Recherche in the
framework of the French "Equipement d'avenir" program, grant ANR J11R107-F), remOcean
(funded by the European Research Council, grant 246777), and the French Bio-Argo program
(BGC-Argo France; funded by CNES-TOSCA, LEFE-GMMC).



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
