# Peer review of "Defining BGC-Argo-based metrics of ocean health and biogeochemical functioning for the evaluation of global ocean models"

_Biogeosciences, 2021_

## Referee Comment (RC1)

**General comments:**

Despite the increasing significance of BGC models, the model validation is limited to the comparison with satellite estimates of surface properties, the climatological data, and/or sparse in-situ observations. In recent years, the fast growing BGC-Argo network provides opportunities to evaluate BGC models in an unprecedented spatial and temporal resolutions. Since there is a large number of floats at the global scale, it becomes difficult to evaluate the global model through the point-to-point comparison which has been used in the regional model. This study suggests some BGC-Argo-based metrics to evaluate a global model and provides some diagnostic plots to display these metrics. This manuscript is well structured and easy to follow. I would suggest to publish after minor revision.

**Specific comments:**

P5 Line 18-23: The BGC-Argo-based POC concentrations were obtained from the filtered bbp signals and therefore should be the small, slow-sinking POC (i.e., 0.2-20$\mu$m) (Dall'Olmo and Mork, 2014; Lacour et al., 2019). In the following model description (P6 Line 15-16), the authors mentioned that their POC model had two size classes. Which modelled POC class was compared with the BGC-Argo based one?

Based on Roesler et al. (2017), the BGC-Argo based chlorophyll were suggested to be divided by a factor of 2 due to the systematic error in fluorometers. It seems that the authors did not apply this correction to the chlorophyll. If not, this can partially explain the model underestimation of surface chlorophyll in the high-chlorophyll regions (please see the Figure 4). The authors should include some descriptions on how they process the BGC-Argo based chlorophyll.

P5 Line 25-30: I am concerned with the comparisons between the global model and the estimates from CANYON-B neural network which is also a model. Although it has been validated with some independent observations (e.g. the GO-SHIP cruise data and BGC-Argo floats), differences between the global model and the CANYON-B neural network may come from the CANYON-B neural network's deviation from the observations.

P8 Line 1-2: Since the POC concentrations vary a lot (~ 2 orders of magnitude) within the mesopelagic zone, the averaged POC concentrations will be skewed to the upper layers right below the mixed layer. In addition, the reference is not approporiate here since the upper bound of mesopelagic zone was defined as the base of productive layer (the maximum of mixed layer and the euphotic zone) in Dall'Olmo and Mork (2014).

P8 Line 13: What is the definition of H? I guess it is the mixed layer depth (MLD).

P13 Line 26-29: I don't agree with this sentence "However, *this seems to have a limited effect on the export of POC …*". First, the conclusion here is anti-intuitive because the authors mixed

up the POC concentration with the POC export flux. In the north Atlantic, the POC concentration is largely determined by the small, slow-sinking POC. Although the relative contributions of small, slow-sinking POC has been recently addressed, the POC export flux is dominated by the gravitational sinking flux of large, fast-sinking POC, which is estimated by multiplying the concentration and sinking velocity. Therefore, the large differences in the POC export flux can be hide by the similar POC concentrations. Second, the lower sPOC but the similar levels of $POC_{meso}$ (Figure 6) can be a result of the suboptimal parameter values, e.g. the underestimated remineralization rate. However, this cannot deny the sensitivity of $POC_{meso}$ to the primary productivity. This is very likely that the $POC_{meso}$ will vary a lot if the authors change the modeled primary productivity.

**Reference**

Dall'Olmo, G. and Mork, K. A.: Carbon export by small particles in the Norwegian Sea, Geophysical Research Letters, 41(8), 2921–2927, doi:https://doi.org/10.1002/2014GL059244, 2014.

Lacour, L., Briggs, N., Claustre, H., Ardyna, M. and Dall'Olmo, G.: The Intraseasonal Dynamics of the Mixed Layer Pump in the Subpolar North Atlantic Ocean: A Biogeochemical-Argo Float Approach, Global Biogeochemical Cycles, 33(3), 266–281, doi:10.1029/2018GB005997, 2019.

Roesler, C., Uitz, J., Claustre, H., Boss, E., Xing, X., Organelli, E., Briggs, N., Bricaud, A., Schmechtig, C., Poteau, A., D'Ortenzio, F., Ras, J., Drapeau, S., Haëntjens, N. and Barbieux, M.: Recommendations for obtaining unbiased chlorophyll estimates from in situ chlorophyll fluorometers: A global analysis of WET Labs ECO sensors, Limnology and Oceanography: Methods, 15(6), 572–585, doi:10.1002/lom3.10185, 2017.

---

## Author Comment (AC1)

We thank the reviewer for their thoughtful comments. Here we offer detailed responses to all questions. Reviewer's comments are in black, our replies are in blue.

**Responses to Reviewer #1 comments:**

General comments:

Despite the increasing significance of BGC models, the model validation is limited to the comparison with satellite estimates of surface properties, the climatological data, and/or sparse in-situ observations. In recent years, the fast growing BGC-Argo network provides opportunities to evaluate BGC models in an unprecedented spatial and temporal resolutions. Since there is a large number of floats at the global scale, it becomes difficult to evaluate the global model through the point-to-point comparison which has been used in the regional model. This study suggests some BGC-Argo-based metrics to evaluate a global model and provides some diagnostic plots to display these metrics. This manuscript is well structured and easy to follow. I would suggest to publish after minor revision.

REPLY: Thanks for the positive assessment of our work.

Specific comments:

P5 Line 18-23: The BGC-Argo-based POC concentrations were obtained from the filtered bbp signals and therefore should be the small, slow-sinking POC (i.e., 0.2-20µm) (Dall'Olmo and Mork, 2014; Lacour et al., 2019). In the following model description (P6 Line 15-16), the authors mentioned that their POC model had two size classes. Which modelled POC class was compared with the BGC-Argo based one?

REPLY: In the PISCES model, POC corresponds to the sum of the two size classes of detritus, phytoplankton and zooplankton (Levy et al. 2013). Based on the reviewer's suggestion, we will compare the smallest sizes of simulated detritus and phytoplankton to match the small and slow-sinking POC observed by the BGC-Argo floats.

Based on Roesler et al. (2017), the BGC-Argo based chlorophyll were suggested to be divided by a factor of 2 due to the systematic error in fluorometers. It seems that the authors did not apply this correction to the chlorophyll. If not, this can partially explain the model underestimation of surface chlorophyll in the high-chlorophyll regions (please

see the Figure 4). The authors should include some descriptions on how they process the BGC-Argo based chlorophyll.

REPLY: The gain adjustment of 0.5 is already implemented in the "adjusted" chlorophyll data (Bittig et al., 2019). We have not applied any processing to the BGC-Argo data apart from those already applied at the Data Assembly Center levels as described in the given references. We will clarify this point in the revised version of the manuscript.

P5 Line 25-30: I am concerned with the comparisons between the global model and the estimates from CANYON-B neural network which is also a model. Although it has been validated with some independent observations (e.g. the GO-SHIP cruise data and BGC-Argo floats), differences between the global model and the CANYON-B neural network may come from the CANYON-B neural network's deviation from the observations.

REPLY: We agree with the reviewer that we do not provide justification for mixing together BGC-Argo data with CANYON-B estimates. We propose to add the following paragraph in the Data section to justify our choice.

" *Finally, we complemented the existing BGC-Argo dataset with pseudo-observations of $NO_3$, $PO_4$ , Si, and DIC concentrations as well pH and $pCO_2$ using the CANYON-B neural network  (Bittig et al., 2018). CANYON-B estimates vertical profiles of nutrients as well as the carbonate system variables from concomitant measurements of floats pressure, temperature, salinity and $O_2$ qualified in "Delayed "mode together with the associated geolocalization and date of sampling.  The CANYON-B estimates of $NO_3$ and pH were merged with measured values on the rationale that CANYON-B estimates have RMS errors (NO3 = 0.7 µmol/kg , pH= 0.013) (Bittig et al., 2018) which are of the same order of magnitude than the BGC-Argo observations RMS errors (NO3 = 0.5 µmol/kg , pH= 0.07) (Mignot et al., 2019; Johnson et al., 2017). We also verified that RMS errors of CANYON-B estimates are at least 4 times lower than the RMS difference between the model and CANYON-B estimates, so that the comparison of simulated properties with the neural network estimates leads to an evaluation of the model performance. We believe it is reasonable to draw conclusions on the model uncertainty from CANYON-B estimates as long as the pseudo-observations errors are much lower than the model-pseudo observations RMS difference. However, caution should be considered when errors are comparable.*"

P8 Line 1-2: Since the POC concentrations vary a lot (~ 2 orders of magnitude) within the mesopelagic zone, the averaged POC concentrations will be skewed to the upper layers right below the mixed layer. In addition, the reference is not appropriate here

since the upper bound of mesopelagic zone was defined as the base of productive layer (the maximum of mixed layer and the euphotic zone) in Dall'Olmo and Mork (2014).

REPLY: We agree with the reviewer. We used a log10-transformation to represent the data to account for the skewness in this layer. We will also add in the revised version that our definition of mesopelagic POC differs from the Dall'Olmo and Mork (2014) reference.

P8 Line 13: What is the definition of H? I guess it is the mixed layer depth (MLD).

REPLY: It is an omission on our part. H is the mixed layer depth. We will replace H by MLD.

P13 Line 26-29: I don't agree with this sentence "However, this seems to have a limited effect on the export of POC …". First, the conclusion here is anti-intuitive because the authors mixed up the POC concentration with the POC export flux. In the north Atlantic, the POC concentration is largely determined by the small, slow-sinking POC. Although the relative contributions of small, slow-sinking POC has been recently addressed, the POC export flux is dominated by the gravitational sinking flux of large, fast-sinking POC, which is estimated by multiplying the concentration and sinking velocity. Therefore, the large differences in the POC export flux can be hide by the similar POC concentrations. Second, the lower sPOC but the similar levels of POCmeso (Figure 6) can be a result of the suboptimal parameter values, e.g. the underestimated remineralization rate. However, this cannot deny the sensitivity of POCmeso to the primary productivity. This is very likely that the POCmeso will vary a lot if the authors change the modeled primary productivity.

REPLY: We agree with the reviewer that this paragraph bears lots of assumptions. We will revise this paragraph and we will remove the conclusions about the POC export flux. As pointed out by the reviewer, we don't have sufficient information to assess the skill of the model in simulating the export of POC from sPOC and POCmeso.

**References**

Bittig, H. C., Steinhoff, T., Claustre, H., Fiedler, B., Williams, N. L., Sauzède, R., Körtzinger, A., and Gattuso, J.-P.: An alternative to static climatologies: robust estimation of open ocean CO2 variables and nutrient concentrations from T, S, and O2 data using Bayesian neural networks, Front. Mar. Sci., 5, 328, 2018.
Bittig, H. C., Maurer, T. L., Plant, J. N., Wong, A. P., Schmechtig, C., Claustre, H., Trull, T. W., Udaya Bhaskar, T. V. S., Boss, E., and Dall'Olmo, G.: A BGC-Argo guide: Planning, deployment, data handling and usage, Front. Mar. Sci., 6, 502, 2019.
Dall'Olmo, G. and Mork, K. A.: Carbon export by small particles in the Norwegian Sea, Geophys. Res. Lett., 41, 2921–2927, https://doi.org/10.1002/2014GL059244, 2014.
Johnson, Plant, J. N., Coletti, L. J., Jannasch, H. W., Sakamoto, C. M., Riser, S. C., Swift, D. D.,

Williams, N. L., Boss, E., Haëntjens, N., Talley, L. D., and Sarmiento, J. L.: Biogeochemical sensor performance in the SOCCOM profiling float array: SOCCOM BIOGEOCHEMICAL SENSOR PERFORMANCE, J. Geophys. Res. Oceans, 122, 6416–6436, https://doi.org/10.1002/2017JC012838, 2017.

Mignot, A., D'Ortenzio, F., Taillandier, V., Cossarini, G., and Salon, S.: Quantifying Observational Errors in Biogeochemical‑Argo Oxygen, Nitrate, and Chlorophyll *a* Concentrations, Geophys. Res. Lett., 46, 4330–4337, https://doi.org/10.1029/2018GL080541, 2019.

---

## Author Comment (AC2)

We wish to thank Pr. Marcello Vichi for offering many insightful comments and helping us clarify our results. Here we offer detailed responses to all questions. Reviewer's comments are in black, our replies are in blue.

General Comments:

This manuscript is indeed a valid compendium of diagnostics for assessing global ocean ecosystem models, which has been prepared with the aim to demonstrate the use of the multi-disciplinary dataset made available by the BGC-Argo array. The authors should thus be praised for their intention to bring together the community and follow the steps taken by Russel et al. (2018). However, that paper had different entry points, since it was specifically dedicated to a poorly sampled oceanic region and offered a multi-model analysis. This manuscript is well written and constructed, but only conveys a demonstrative message. I am thus not fully convinced by the scope of this present version of the manuscript, as well as by its effective novelty, since it does not add further knowledge to the existing literature [...]

Hence, I have carefully thought about how to write this review, and realised that the most relevant point of clarity would be to illustrate some cases of how readers could approach it. From a point of view of someone approaching modelling validation as a student or early career researcher, this manuscript offers a limited perspective, and one would gain more theoretical and methodological background in the 2009 JMS special issue (Lynch et al., 2009, and all the other papers in the issue), if not from earlier papers in the ecological modelling literature (Oreskes et a', 1994; Rykiel, 1996). If a reader is interested in the validation of the global version of PISCES, this manuscript is insufficient, because it provides a series of figures with few comments and discussions. It is surely of interest to the PISCES developers who are knowledgeable of the model details and possible deficiencies, but then an internal report would suffice. Finally, for experienced global ocean modellers, this manuscript is an illustration of the minimum set of assessments (which I prefer to the term "validation") that serious modellers have been doing in the last ten years when evaluating their model results. In terms of "metrics", it gives indications to compare the model output against the state variables that can be measured by the array of floats and to add derived state variables from applications of artificial intelligence. Ultimately, the assessment is based on visual comparisons of coarsely gridded spatial maps and time series, or through the use of basic univariate scores (bias and RMSD) and cumulative diagrams that combine the same skill scores (e.g. the Taylor diagram, which also includes linear correlation).

REPLY: Thanks for the careful assessment of our work. The goal of this paper is to demonstrate the use of BGC-Argo floats for the evaluation of BGC models at the global scale, through a concise evaluation of the CMEMS global BGC forecasting system. Our hope is that the methodology employed in this study can be useful and informative for other research teams interested in model assessment with BGC-Argo floats. In particular, the main points we want to highlight are: 1) how do we handle BGC-Argo data (e.g., quality control and flags) for model assessment purposes, and 2) to propose BGC-Argo metrics, which we believe are useful to assess the accuracy of the model state. We have intentionally chosen simple metrics, a minimum set of assessments and basic quantitative techniques (visual inspection, bias and RMSE) to focus the message of the study on the 2 points listed above and not on the evaluation of the model simulation. Therefore, this study is not designed as a review of biogeochemical models validation and it does not represent a thorough assessment of PISCES either.

We agree with the reviewer that the main message conveyed by the manuscript is not clear enough and that it can be confusing for the reader. Based on the reviewer's comments, we will modify the manuscript so that the main message of the study appears more clearly to the reader.

First, we will change the title to "*Using BGC-Argo floats for the assessment of marine biogeochemical models : a case study with CMEMS global forecasting system.* "

In the abstract, P1, L-28, we will change to "*Here, we demonstrate the use of the global array of BGC-Argo floats for the assessment of biogeochemical models through a concise evaluation of the CMEMS global forecasting system. We first detail the handling of the BGC-Argo data set for model assessment purposes, then we present 18 assessment metrics to quantify the success of BGC model simulations. The metrics evaluate either the model state accuracy or the skill of the model in capturing emergent properties, such as the Deep Chlorophyll Maximums (DCMs) or Oxygen Minimum Zones (OMZs). These metrics are associated with the air-sea $CO_2$ flux, the biological carbon pump, oceanic pH and oxygen levels. We also suggest four diagnostic plots for displaying such metrics.*"

In the introduction, the paragraph starting P. 4, L2 , will change to " *We aim to demonstrate the use of the BGC-Argo global array for the assessment of BGC models at the global scale. To that end, we performed a concise evaluation of CMEMS global BGC forecasting system using the global fleet of BGC-Argo floats. We expect that the methodology employed here (from the data handling to the use of assessment metrics) would be useful and informative for other research teams interested in model evaluation with BGC-Argo floats.* "

The BGC-Argo data are certainly invaluable, and this is the reason why the community has strived to develop the technology and the financial support to deploy them. The authors did not however succeed in showing their enhanced value for model assessment, beyond the obvious consideration that this increases the number of data, which would be much more evident if this same assessment was done by comparing datasets with and without the contribution of the BGC-Argo.

REPLY: The reviewer brings up an interesting point. It is true that BGC-Argo dramatically increases the availability of data collected by traditional oceanographic cruises. It would indeed be informative to repeat the same assessment by comparing datasets with and without the contribution of the BGC-Argo, such as for example the World Ocean Atlas. While we are very interested in this question, we do not think it belongs to this paper whose main focus is to show the use of BGC-Argo floats for model assessment rather than showing the impact of increasing the number of observations on skill scores.

In summary I have found two major issues with this manuscript that the authors have not considered to a satisfactory extent:
The loose definition of metrics and the absence of uncertainties' treatment. The authors use the term metrics in a rather ambiguous way. They also do not differentiate between measured data and artificially generated data. This implies that the evaluation process does not necessarily lead to an improvement of the model(s).

REPLY: We agree with the reviewer that our definition of metrics was somewhat ambiguous. In the introduction, we will change our definition of metrics based on the recent review of Hipsey et al. (2020):

"*In this study, the BGC-Argo dataset is used in conjunction with the model evaluation framework developed by Hipsey et al. (2020). In particular, they propose three levels of assessment metrics to evaluate the skill of a model simulation: state variables validation (e.g., Chla, nitrate, oxygen, etc…), mass fluxes and process rates validation (e.g.,primary production or division rates), and emergent properties validation (e.g., Deep Chlorophyll maximum, or Oxygen Minimum zones). In this study we present 18 metrics for the assessment of a model simulation with BGC-Argo data. Most of them evaluate the model state accuracy through the comparison of simulated state variables with BGC-Argo observations in the mixed layer or at fixed depth. In addition, some of the metrics assess the skill of the model in capturing emergent properties. These metrics are associated with the air-sea $CO_2$ flux, the biological carbon pump, oceanic pH, oxygen levels and Oxygen Minimum Zones (OMZs). Recent works demonstrated*

*the feasibility of calculation at basin scale, from BGC-Argo observations, of mass fluxes and process rates, such as primary production, phytoplankton division and accumulation rates (Yang et al., 2021; Mignot et al., 2018), net community production (Plant et al., 2016), or carbon export (Dall'Olmo et al., 2016). However, it would be arduous to achieve such estimations on the global BGC-Argo dataset as it requires ad hoc calibration that cannot be easily defined.  As a consequence, the evaluation of simulated process rates with BGC-Argo data is not addressed in this study.*"

In reply to the second comment, as we explain above, the object of the paper is not a thorough analysis of the model performance. Nevertheless, the proposed concise evaluation of the model (e.g., maps of rmsd) can be further exploited (e.g., by analysing the spatial and temporal distribution of the rmsd maps or multivariate relationships of the errors) to study the model uncertainty sources.

Last , we agree with the reviewer that we do not provide justification for mixing together measured data with artificially-generated data. We will add a paragraph in the Data section that justify our choice.

" *Finally, we complemented the existing BGC-Argo dataset with pseudo-observations of $NO_3$, $PO_4$ , Si, and DIC concentrations as well pH and $pCO_2$ using the CANYON-B neural network (Bittig et al., 2018). CANYON-B estimates vertical profiles of nutrients as well as the carbonate system variables from concomitant measurements of floats pressure, temperature, salinity and $O_2$ qualified in "Delayed "mode together with the associated geolocalization and date of sampling.  The CANYON-B estimates of $NO_3$ and pH were merged with measured values on the rationale that CANYON-B estimates have RMS errors (NO3 = 0.7 µmol/kg , pH= 0.013) (Bittig et al., 2018) which are of the same order of magnitude than the BGC-Argo observations errors (NO3 = 0.5 µmol/kg , pH= 0.07) (Mignot et al., 2019; Johnson et al., 2017). We also verified that RMS errors of CANYON-B estimates are at least 4 times lower than the RMS difference between the model and CANYON-B estimates, so that the comparison of simulated properties with the neural network estimates leads to an evaluation of the model performance. We believe it is reasonable to draw conclusions on the model uncertainty from CANYON-B estimates as long as the pseudo-observations errors are much lower than the model-pseudo observations RMS difference. However, caution should be considered when errors are comparable.*"

The unconvincing enhancement of the effective role of BGC-Argo data in model assessment. Basically, the question I have is: why BGC-Argo are good enough and should be used separately and not as part of a global compilation of data such as the World Ocean Atlas? (which incidentally includes or will include the BGC-ARgo data).

Since BGC-Argos are ultimately increasing the availability of data that are usually collected by means of traditional oceanographic cruises, what is indeed their value in model validation?

REPLY: We thank the reviewer for bringing this to our attention. When we wrote the first version of the manuscript, we did not know that the BGC-Argo data were available from the World Ocean Database (WOD). We have examined the documentation that deals with the data processing in the WOD (https://www.ncei.noaa.gov/sites/default/files/2020-04/wod_intro_0.pdf) but we haven't found sufficient information concerning the data mode used in the WOD. As we detail in the manuscript, the "Delayed-mode" represents the highest quality of data but for some variables, only a limited fraction of data is accessible in "Delayed-Mode". Consequently, for each variable, we selected the highest quality of data (i.e., "Adjusted" or "Delayed mode") that did not compromise too much the number of observations available. We are not sure whether such data selection is possible with the World Ocean Database, so we prefer to use the BGC-Argo data directly downloaded from Argo Coriolis Global Data Assembly Centre and not as part of a global compilation of data.

Furthermore, one of the issues of large databases such as WOD, is the accessibility and the interoperability of the data that compose it, which, ultimately, affects their overall accuracy. Using the BGC-Argo dataset separately is a way to ensure consistent accuracy. The GLODAP V2 data set (on which CANYON B is developed) is an illustration of an interoperable homogenous data set (with very strict data QC procedure) used for model assessment and not used as part of a global compilation of data.

Finally, in reply to the last question, the BGC-Argo floats provide observations at high vertical and temporal resolutions and for long periods of time allowing to compute time-series of vertical characteristics of the variables. This is not possible with discrete vertical samplings provided by cruise cast *in situ* measurements..

We will comment on these points in the revised version of the manuscript.

For clarity, I would like to elaborate more on the first concept above, while the second point is mostly derived from the specific comments detailed in the next section. Russel et al (2018) also use the concept of metrics in a wider sense, although they define metrics as "any quantity or quantifiable pattern that summarizes a particular process or the response in a model to known forcings". The strength of the ACC transport at Drake Passage or the latitude of the maximum zonal mean winds over the Southern Ocean

are "metrics" in this context. They are combinations of state variables, or values of state variables at specific locations.

In this context, all the surface state variables listed in Table 2, are indeed components of the biological carbon pump, but they are not metrics. They are simply state variables. Only when considered together to evidence emergent patterns they may give indications of proper process functionality (e.g. the ratio of particulate organic carbon to total chlorophyll, de Mora et al, 2016). I agree that the DCM and the "nutricline" (which would deserve a more appropriate definition, see specific points below) are "metrics", as well as the depth of the hypoxic layer. Mixing together indicators of processes with state variables is confusing, unless a rigorous link between a single state variable and the process is established.

REPLY: As we explain above, we have changed our definition of metrics. We now use the framework proposed by Hipsey et al. (2020). They propose three levels of assessment metrics to evaluate the skill of a model simulation: state variables validation (e.g., Chla, nitrate, oxygen, etc…), mass fluxes and process rates validation (e.g.,primary production or division rates), and emergent properties validation (e.g., Deep Chlorophyll maximum, or Oxygen Minimum zones). We will indicate in Table 2, which level a proposed metric is referring to. We will also make a rigorous link between the state variable and the associate process.

This manuscript increases the risk of misinterpretation by mixing together "metrics" and skill scores. Neither Russel et al (2018) and this manuscript expand on the concept of metrics performance and objective assessment (performance indicators, skill scores, cost functions, are all synonyms that depend on the specific discipline), which was instead done by Allen et al. (2007), Friedrichs et al. (2009), Vichi and Masina (2009) and others in the JMS special issue. For ease of simplicity, I will use the term skill score, which is the one used in the more mature field of weather forecasting. State variables can be assessed using univariate skill scores, and this is a necessary exercise for any modeller to ensure the model has some grip with reality. Figure 3 and the other density plots in the Appendix give a visual indication of the skill score, but they do not quantify it (e.g. Smith and Rose, 1995; Rose and Smith, 1998). I also have another question linked to my Point 2 (and further detailed in the specific comments): why should this exercise be done only with the BGC-Argo and not also including the other existing data? Since BGC-Argo are evaluated against cruise cast benchmarks, then those data are usually considered always superior, and should be used. Again, the real value of the BGC-Argo would have been shown if the score had been substantially modified with the inclusion of the Argo data.

REPLY: We will add a Table that quantifies the skill scores for each metrics as done in Vichi and Masina (2009) or Doney et al. (2009). As we explain above, we believe it is

more reasonable to use the BGC-Argo data as a separate dataset rather than as part of a global compilation of data.

Specific comments:

P2L1 - Earlier work has specifically addressed the impact of assimilation on the carbonate system (Visinelli et al., 2017)

REPLY: We will add the reference in the revised version of the manuscript.

P2L26-29 - This sentence is mixing together sensor accuracy, which has been assessed by Johnson et al and Mignot et al, in two specific regions of the world ocean) and temporal/vertical resolutions, which have not been assessed as far as I am aware. This is misleading. 10 days may not be sufficient for all variables, as well as the vertical binning that is done. The comparisons have assessed the equivalence between rosette casts and the floats, but they say nothing about the temporal and vertical resolution. For certain processes, such as carbon exchange and phytoplankton biomass through chlorophyll and backscattering proxies, a resolution of 10 days would lead to sampling aliases either of the mean or of the variability (Monteiro et al., 2015, Little et al., 2018). These are examples from the Southern Ocean, where there is the highest density of buoys.

REPLY: We will revise the sentence and we will remove the part about the temporal and vertical resolutions.

P2L32-34 - The authors should be more specific. Other datasets, such as for instance remote sensing, are less limited in terms of temporal and spatial resolutions. This is connected to the concerns expressed in Point 1 above.

REPLY: We will revise the sentence, and we will be more specific about the temporal and spatial resolutions.

P4L3-5 This sentence seems to imply that one can only perform point-by-point comparisons when there are few floats, which is odd. Again linked to my main Point 1 above. The authors should explain why given the current computing capability, they only suggest to perform diagnostics for few selected tracks and not for the overall dataset (Section 5.d).

REPLY: We have changed this paragraph based on point 1 and point 2 (see above). This sentence will be removed in the revised version of the manuscript.

P4L12-16 The connection between the variables and the ocean health/ecosystem functioning is not made explicit in the text. Taking as an example the ocean health index (http://www.oceanhealthindex.org/), establishing ocean health is obtained as a multivariate analysis of several data layers, forming a selected set of drivers and their associated thresholds. The authors should be more explicit about their intent here.

REPLY: We have changed our definition of metrics. We will no longer refer to ocean health and ecosystem functioning in the revised version of the manuscript.

P5L12-13 This is not an objective criterion. What is an acceptable level of compromise?

REPLY: We have added an objective criterion to the paragraph: *" However, for some variables, only a limited fraction of data is accessible in "Delayed-Mode". Consequently, for each variable, we selected the data modes, where at least 80 % of the data are available (see Table 1). Note that this criterion does not apply to $O_2$, where only delayed mode data were selected in order to generate the pseudo-observations from CANYON-B neural network (see after). "*

P5L22 There are many other relationships, and they have been shown to give different results (e.g. Thomalla et a., 2017l). The authors should explain why they are recommending this one.

REPLY: In the revised version of the manuscript, we will use a POC vs $b_{bp}$ relationship developed for the global ocean (https://catalogue.marine.copernicus.eu/documents/QUID/CMEMS-MOB-QUID-015-010.pdf) based on a global database of in situ POC and satellite $b_{bp}$ (Evers-King et al., 2017). This relationship, developed for global application, has been shown to outperform regional relationships, such as Cetinic et al. (2012), at global scales.

P6L12-15 It appears that this method of linear resampling would artificially increase the number of data, and hence bias the statistical results, especially in conditions where there are not enough data.

REPLY: This is a good comment. We will add that the method of linear resampling can possibly bias our statistical results.

P7L10-12 The authors do not discuss what would happen if the MLD is different between the observations and the model.

REPLY: In this study, the dynamical component has been extensively validated (Lellouche et al., 2013, 2018), and correctly represented variables that are constrained by observations (e. g., temperature and salinity), including Argo profiles. We verified that the MLD, which is calculated on a density criterion basis, is indeed correctly represented in the model. The global bias between the model and the BGC-Argo observations is 0.3 m. We will add a sentence that specifies that we verified that the MLD is well simulated by the model.

P7L29-30 Related to my point 1 above. The relationship between the state variables and the ecosystem functions is not made explicit. The term "useful" should be motivated.

REPLY: We will revise this section, and we will make the relationship between the state variables and ecosystem function more explicit. Note that, we will add new metrics in the mesopelagic layer as explained below.

"*The biological carbon pump is the transformation of nutrients and dissolved inorganic carbon into organic carbon in the upper part of the ocean through phytoplankton photosynthesis and the subsequent transfer of this organic material into the deep ocean. The functioning of this pump relies on key pools of nutrients and carbon as well as a number of processes that control mass fluxes between the pools.*

*The first level of assessment of a biological carbon pump simulated by a model consists in evaluating the different pools (or state variables) of the pump (Hipsey et al. 2020). In particular, the comparison of simulated surface nutrients ($NO_3$, $PO_4$, and Si), DIC, Chla and POC with BGC-Argo observations gives an indirect evaluation to demonstrate the model is capturing key processes of the biological carbon pump in the upper layer of the ocean, such as primary production, respiration, grazing. A second-level , and more indicative, assessment would be to directly compare these key processes with measured mass fluxes, but this is not addressed in this study. The surface nutrients , DIC, Chla and POC (hereinafter denoted $sNO_3$, $sPO_4$, sSi, sDIC, sChl and sPOC) correspond to the average concentrations in the mixed layer.*

*Similarly, the evaluation of the mesopelagic nutrients, DIC and POC concentration (hereinafter indicated with the subscript $_{meso}$) provides an indirect evaluation of the key processes in the mesopelagic layer, such as export production, respiration,etc. The mesopelagic concentrations correspond to the depth-averaged concentrations between the base of the mixed layer down to 1000 m.* "

P8L7-8 Same as above, the value of DCM as an indicator should be contextualized. Why are BGC-Argo data providing a better estimate of this metric than other data?

REPLY: We will revise the paragraph and we will contextualize the use of the DCM as an indicator. "*At the base of the euphotic layer of stratified systems, a Chla maximum (hereinafter denoted Deep Chlorophyll Maximum, DCM) develops that generally escapes detection by remote sensing (Barbieux et al., 2019; Cullen, 2015; Letelier et al., 2004; Mignot et al., 2011, 2014). It has been suggested that the DCM plays an important role in the synthesis of organic carbon by phytoplankton (Macías et al., 2014). DCMs are therefore important features to be assessed in BGC models with respect to biological carbon pump processes such as the primary production. Furthermore, DCMs are also an emergent feature that develops in response to complex physical and biogeochemical interactions (Cullen, 2015). Thus, their evaluation provides critical information regarding the accuracy of the model in capturing complex patterns of key ecosystem processes.*"

As we explain above, the BGC-Argo data provide consistent profiles at high vertical and temporal resolution allowing to derive time-series of DCM depths. In comparison, discrete vertical samplings provided by cruise cast *in situ* measurements have a vertical resolution much lower (10 samples taken over a 100 m layer ), with no repetitive sampling.

P8L13 Please explain what H is.

REPLY: It is an omission on our part. H is the mixed layer depth. We will replace H by MLD.

P8L14-16 This may be confusing for some readers, since it's not technically a gradient. The cited paper uses and justifies this definition. I'd suggest the authors to be more precise and give their definition and how this is an effective metric of the carbon pump. Also, there is a difference in sampling between argo and the layers of discrete models. How is this taken into account?

REPLY: We will be more precise about the definition of the nitracline depth and describe how this is an effective metric of the carbon pump.

"*The vertical supply of $NO_3$ to the surface layers is a critical process of the biological carbon pump as $NO_3$ is often depleted in the surface layers and is a limiting factor for phytoplankton photosynthesis in most oceanic regions. This flux depends, among other things, on the vertical gradient of $NO_3$ (the nitracline), and, in particular, its depth (the*

*nitracline depth)* (Cermeno et al., 2008; Omand and Mahadevan, 2015). *Therefore, the comparison of the simulated nitracline depth with BGC-Argo observations allows for an indirect assessment of the model quality in reproducing vertical fluxes of $NO_3$. Following previous studies* (Cermeno et al., 2008; Lavigne et al., 2013; Richardson and Bendtsen, 2019)*, the depth of the nitracline corresponds to the first depth where $NO_3$ is detected. The threshold value is set to 1 µmol/kg, which corresponds to an upper estimate of BGC-Argo $NO_3$ data accuracy* (Johnson et al., 2017; Mignot et al., 2019). "

Finally, there is indeed a difference in sampling between the BGC-Argo and the layers of discrete models. This is clearly visible in the scatterplot for the nitracline , the DCM and the OMZ depths. We will comment on this point in the revised version of the manuscript.

P8l28-30 At P4L11 it is reported "depth of the OMZ". This the depth of the oxygen minimum. It should be explained how and why this is a good indicator, and why the BGCArgo data are superior in its identification.

REPLY: We will explain in the revised version of the manuscript, why the depth of the oxygen minimum is a good indicator. "*Oxygens levels in the global and coastal waters have declined over the whole water column over the past decades (Schmidtko et al., 2017)and OMZs are expanding (Stramma et al., 2008). Assessing how models correctly represent ocean oxygen levels as well as the OMZs is therefore critical to monitor their changes over time. Similarly to DCMs, the assessment of OMZs is also informative on how the model simulates emergent dynamics as OMZs originate from intricate physical and biogeochemical interactions (Paulmier and Ruiz-Pino, 2009).* "

We detail in a previous reply, why the BGC-Argo are particularly fit in the identification of vertical characteristics of BGC variables.

P9L26 This statement about non-linearity is odd in the context of model goodness-of-fit (Smith and Rose, 1995; Pineiro et al, 2008; Vichi and Masina, 2009). If it's non-linear, then the assessment is failed.

REPLY: We will remove this sentence.

P10-8-12 The choice of the binning interval should be discussed. What is the advantage of losing the variability measured by the floats? Why not using the standard deviation as an indicator of the model skill to reproduce the proper scales? These are enhanced features that only the BGC-Argo data would allow to compute.

REPLY: We will discuss the choice of the binning interval in the revised version of the manuscript. "...*To do so, the metrics from 2009 to 2017 are averaged in 4°x4° bins, bins with less than 4 points being not included. The 4° distance in an upper estimate of the autocorrelation length scales for $O_2$, nutrients, and $pCO_2$ (comprised between 300 and 400 km) between 20° and 40° of latitude in both hemispheres* (Biogeochemical-Argo Planning Group, 2016)."

We will also add in section 4.c that standard deviation can also be displayed on spatial maps as an indicator of the model skill to reproduce the proper scales. However, we won't show it in the manuscript as we prefer to not overload Figure 4 and the associated supplementary figures with additional panels.

P10L22-24 Allen et al (2007) warned against the visual comparison of time series. This sentence is generic and should be explained in the context of the augmented data provided by the BGC-Argo.

REPLY: We agree with the reviewer that visual inspection relies on the subjective appreciation of the evaluator. Consequently, we will add time-series of normalized skill scores to Figures 5 and 6.  We will add this sentence at the end of the section 4c. " *In addition to the time series of metrics, we also displayed time series of normalized skill scores such as percent BIAS and RMSD to avoid relying only on subjective visual inspection*. "

P11L11-14 The results are not presented according to the concept of the biological carbon pump "metric". It is evident that the nutrients are correlated while all carbon flux variables are not performing. Which ultimately questions the use of surface nutrients as indicators of carbon cycling.

REPLY: The fact that nutrients are well represented in the model suggests that the model captures the combination of process rates that drive nutrients dynamics.  Some of these process rates drive both the nutrients and carbon dynamics, but there are also rates that are specific to each state variable. This probably explains why the carbon variables are not performing while the nutrients are well simulated. However, it must be recognised that without a direct assessment of the individual rates, we cannot verify this hypothesis. We will clarify this point in the revised version of the manuscript.

P11L31 I cannot see the data "around" the line. I rather see an overestimation. (it is either Cape Verde or Cap Vert)

REPLY: We will improve the clarity of the figure in the revised version of the manuscript.

P12-L2-17 Linked to Point 2 above. The authors seem to imply that BGC-Argo data are more suitable than ocean colour for model assessment. I acknowledge that this is not explicitly written, but there is no clear rationale. This kind of map would certainly be superior in terms of spatial and temporal resolution when using that product as Benchmark.

REPLY: We do not imply that BGC-Argo data are more suitable than ocean colour for model assessment.

P12-section-d This is the section that mostly led to the inclusion of Point 2 above. The shown time series is 2 years long, which is an invaluable source of data from a region that has been influential in shaping our understanding of the spring bloom. I am missing the point why the authors are writing the term spring bloom in quotes. The advantage of time series from floats that remained in a given province of the global ocean is of huge potential in model validation. The offered description is quite generic, which could have been done even using monthly climatological time series obtained from the WOA, or from the existing long-term observational ocean sites (BATS, PAPA, HOT). The BGC-Argo floats are an unprecedented source of multiple opportunities to do validation in several regions of the world ocean (with some limitations), but this present form of the manuscript does not offer any specific recommendation of what numerical modellers should do to unleash this potential. I would be very interested in seeing an exploitation of the multivariate nature of BGC-Argo, while I only see multi-panel plots.

REPLY: Based on this comment, we will revise this section. We will remove the unnecessary description of the spring bloom. We will also highlight the invaluable opportunities of such time series for the assessment of models by showing other time series in regions where in situ data are scarce. Concerning the evaluation of the multivariate nature of BGC-Argo, we agree that it is an interesting point to pursue. We are very interested in applying the multivariate approach proposed by Allen et al. (2007) to the BGC-Argo data set. However, we prefer to focus this manuscript on the presentation of the metrics and to exploit the multivariate approach in another study.

P13L4-5 The authors should do more than simply say "correctly represented". This is a subjective statement, which is based on a visual comparison, exactly what the community challenged in the last 10-15 years. The advantage is that now we can use a frequency of 10 days, when initially phenology analysis was based on monthly data. Again, the authors are missing an opportunity to demonstrate the intrinsic value of this new data set.

REPLY: As explained above, we will include time series of skill scores to avoid relying only on subjective visual inspection. We agree that the frequency of 10 days is a significant progress over previous data sets. However, as explained in the conclusion, we do not address phenology metrics in this study because the number of observations per month and per bins is still too low to perform a global analysis.

P13-L13-20 This is a more detailed analysis of this specific model, which indeed brings in some of the advantages of a multivariate data set. However, there is a combination of measured and derived variables, which are treated as if they were equivalent. Quite a few questions come to mind: Is there a possibility that there is artificial correlation in the derivation of the phosphate and silicate concentration? What is the error associated with the CANYON-B method? Which is the effective (measured) variable mostly responsible for the response of the other estimated nutrients? The reduced consumption occurs during the spring period, and is continued during summertime. Hence, there is a factor at play during the late spring period, which is less likely to be reduced uptake from smaller phytoplankton during summer as suggested. It may thus be a delayed onset of the phytoplankton succession, or maybe a faster remineralization occurring in the upper layers, which retain more inorganic nutrients closer to the surface. This may indeed be beyond the scope of the manuscript, but it has been the authors' decision to propose some mechanistic explanations of this discrepancy. Showing a complete example of how the use of multivariate data allows modellers to investigate model deficiencies would offer guidelines to other modellers.

REPLY: As explained above, we will include a paragraph in the Data section that discusses the error associated with the CANYON-B method. In reply to the second comment, we will also discuss the hypothesis proposed by the reviewer in the revised version of the manuscript.

P13-L22-23 This sentence bears lots of assumptions. This is really where BGC-Argo can make a difference. The related uncertainties should however be highlighted, together with recommendations to other modellers on how to best approach the assessment of the carbon cycle metrics.

REPLY: Based on the reviewer's comment, we will revise this paragraph. We will also provide recommendations on how to best approach the assessment of the carbon cycle metrics.

P13L26-29 This argument is flawed. If the occurrence of the peak is matched in the

mesopelagic layer rather than at the surface, it is a clear indication of vertical mismatches in the export. I would thus argue that POC concentration is a proper metric for the export component of the carbon cycle. I would again encourage the authors to replace the use of subjective terms such as "consistent" with objective indicators (see Allen et al., 2007). For instance the comparison of the skill score computed in two consecutive years would give indication if there is some variability or if the model tends to repeat the same pattern.

REPLY: We will revise this paragraph in the revised version of the manuscript and we will compute time-series of skill scores.

P14L16-19 I would recommend more clarity on this statement. Are these sensors not available on the global ocean floats? It is not clear why this example is presented for Mediterranean floats, and not introduced earlier as one major advantage of the BGC-Argo floats.

REPLY: We will clarify this statement. We will also add that the sensors are available on the global ocean. However, the global model used in the study does not resolve the spectral and directional properties of the underwater light field. That's why we didn't use the global model but a model of the Mediterranean Sea equipped with a multispectral light module. We will also clarify this point.

P14L26-28 This sentence is similar to the statements done in the earlier sections. This is not technically a perspective statement.

REPLY: We will add a perspective statement in the revised version of the manuscript.

P15L1-6 The question is whether these data should be used "on their own" or in conjunction with the other existing datasets. The authors should clearly explain in the conclusion why this dataset should be exploited as a separate unit.

REPLY: Based on our previous replies to this comment, we will explain in the conclusion why this dataset should be exploited as a separate unit.

P15L32-P16L3 I would thus recommend the authors to thoroughly address the issue of how the uncertainties should be treated. This is particularly important in the case of mixing measured and derived variables. If BGC-Argo are capable, within their limits, to reduce uncertainties in model assessment exercise, this should be adequately

argumented. The fact that there are more data available is undoubtedly of relevance, but I wonder if it does help to reduce uncertainties in model states.

REPLY: We verified that the RMS errors of BGC-Argo data are always lower than the RMS difference between the model and BGC-Argo observations, so that the comparison of simulated properties with the observations leads to an evaluation of the model performance. We will detail this point in the conclusion.

P16L15-18 Please highlight in which part of the results this is shown.

REPLY: We will highlight in which part of the results this is shown.

P17L2 Please add in the caption the meaning of the codes (or a link to where they are explained more in detail). Also, in the heading of the 3rd column, correct Date with Data. Figure 2 Taylor diagrams are based on geometric properties of the circle. Hence they should be presented using equal axes.

REPLY: We will add the meaning of the codes, change Date with Data and present the Taylor diagram using equal axes.

---

## Author Response (AR1)

**REVIEWER #1**

We thank the reviewer for their thoughtful comments. Here we offer detailed responses to all questions. Reviewer's comments are in black, our replies are in blue.

General comments:

Despite the increasing significance of BGC models, the model validation is limited to the comparison with satellite estimates of surface properties, the climatological data, and/or sparse in-situ observations. In recent years, the fast growing BGC-Argo network provides opportunities to evaluate BGC models in an unprecedented spatial and temporal resolutions. Since there is a large number of floats at the global scale, it becomes difficult to evaluate the global model through the point-to-point comparison which has been used in the regional model. This study suggests some BGC-Argo-based metrics to evaluate a global model and provides some diagnostic plots to display these metrics. This manuscript is well structured and easy to follow. I would suggest to publish after minor revision.

REPLY: Thanks for the positive assessment of our work.

Specific comments:

P5 Line 18-23: The BGC-Argo-based POC concentrations were obtained from the filtered bbp signals and therefore should be the small, slow-sinking POC (i.e., 0.2-20µm) (Dall'Olmo and Mork, 2014; Lacour et al., 2019). In the following model description (P6 Line 15-16), the authors mentioned that their POC model had two size classes. Which modelled POC class was compared with the BGC-Argo based one?

REPLY: Following the approach of Gali et al. (2021) and based on the reviewer's suggestion, in the new version of the manuscript we compare the two sizes classes of phytoplankton, the small detrital particles and microzooplankton modelled by PISCES to match the small and slow-sinking POC observed by the BGC-Argo floats.

Based on Roesler et al. (2017), the BGC-Argo based chlorophyll were suggested to be divided by a factor of 2 due to the systematic error in fluorometers. It seems that the authors did not apply this correction to the chlorophyll. If not, this can partially explain the model underestimation of surface chlorophyll in the high-chlorophyll regions (please see the Figure 4). The authors should include some descriptions on how they process the BGC-Argo based chlorophyll.

REPLY: The gain adjustment of 0.5 is already implemented in the "adjusted" chlorophyll data (Bittig et al., 2019). We have not applied any processing to the BGC-Argo data apart from those already applied at the Data Assembly Center levels as described in the given references. We have clarified this point in the revised version of the manuscript.

P5 Line 25-30: I am concerned with the comparisons between the global model and the estimates from CANYON-B neural network which is also a model. Although it has been validated with some independent observations (e.g. the GO-SHIP cruise data and BGC-Argo floats), differences between the global model and the CANYON-B neural network may come from the CANYON-B neural network's deviation from the observations.

REPLY: We agree with the reviewer that we do not provide justification for mixing together BGC-Argo data with CANYON-B estimates. We added the following paragraph in the Data section to justify our choice.

"Finally, we complemented the existing BGC-Argo dataset with pseudo-observations of $NO_3$, PO4 , Si, and DIC concentrations as well as pH and pCO2 using the CANYON-B neural network (Bittig et al., 2018). CANYON-B estimates vertical profiles of nutrients as well as the carbonate system variables from concomitant measurements of floats pressure, temperature, salinity and O2 qualified in "Delayed" mode together with the associated geolocalization and date of sampling. The CANYON-B estimates of $NO_3$ and pH were merged with measured values on the rationale that CANYON-B estimates have RMS errors ( $NO_3$ = 0.7 µmol $kg^{-1}$ , pH = 0.013) (Bittig et al., 2018) that are of the same order of magnitude as those of the BGC-Argo observations errors ( $NO_3$ = 0.5 µmol $kg^{-1}$, pH = 0.07) (Mignot et al., 2019; Johnson et al., 2017).

Finally, we verified that the RMS errors of BGC-Argo data (both measured and from CANYON-B estimates) are lower than the RMS difference between the model and BGC-Argo data, so that the comparison of simulated properties with the BGC-Argo data leads to a meaningful evaluation of the model performance. We believe it is reasonable to draw conclusions on the model uncertainty from BGC-Argo data as long as the BGC-Argo errors are much lower than the model-observations RMS difference."

P8 Line 1-2: Since the POC concentrations vary a lot (~ 2 orders of magnitude) within the mesopelagic zone, the averaged POC concentrations will be skewed to the upper layers right below the mixed layer. In addition, the reference is not appropriate here since the upper bound of mesopelagic zone was defined as the base of productive layer (the maximum of mixed layer and the euphotic zone) in Dall'Olmo and Mork (2014).

REPLY: We agree with the reviewer. We used a log10-transformation to represent the data to account for the skewness in this layer. In the revised version of the manuscript, we have removed the Dall'Olmo and Mork (2014) reference.

P8 Line 13: What is the definition of H? I guess it is the mixed layer depth (MLD).

REPLY: It is an omission on our part. H is the mixed layer depth. We have replaced H by MLD.

P13 Line 26-29: I don't agree with this sentence "However, this seems to have a limited effect on the export of POC …". First, the conclusion here is anti-intuitive because the authors mixed up the POC concentration with the POC export flux. In the north Atlantic, the POC concentration is largely determined by the small, slow-sinking POC. Although the relative contributions of small, slow-sinking POC has been recently addressed, the POC export flux is dominated by the gravitational sinking flux of large, fast-sinking POC, which is estimated by multiplying the concentration and sinking velocity. Therefore, the large differences in the POC export flux can be hide by the similar POC concentrations. Second, the lower sPOC but the similar levels of POCmeso (Figure 6) can be a result of the suboptimal parameter values, e.g. the underestimated remineralization rate. However, this cannot deny the sensitivity of POCmeso to the primary productivity. This is very likely that the POCmeso will vary a lot if the authors change the modeled primary productivity.

REPLY: We agree with the reviewer that this paragraph bears lots of assumptions. We have revised this paragraph and we have removed the conclusions about the POC export flux. As pointed out by the reviewer, we do not have sufficient information to assess the skill of the model in simulating the export of POC from sPOC and POCmeso.

**References**

Bittig, H. C., Steinhoff, T., Claustre, H., Fiedler, B., Williams, N. L., Sauzède, R., Körtzinger, A., and Gattuso, J.-P.: An alternative to static climatologies: robust estimation of open ocean CO2 variables and nutrient concentrations from T, S, and O2 data using Bayesian neural networks, Front. Mar. Sci., 5, 328, 2018.

Bittig, H. C., Maurer, T. L., Plant, J. N., Wong, A. P., Schmechtig, C., Claustre, H., Trull, T. W., Udaya Bhaskar, T. V. S., Boss, E., and Dall'Olmo, G.: A BGC-Argo guide: Planning, deployment, data handling and usage, Front. Mar. Sci., 6, 502, 2019.

Dall'Olmo, G. and Mork, K. A.: Carbon export by small particles in the Norwegian Sea, Geophys. Res. Lett., 41, 2921–2927, https://doi.org/10.1002/2014GL059244, 2014.

Galí, M., Falls, M., Claustre, H., Aumont, O., and Bernardello, R.: Bridging the gaps between particulate backscattering measurements and modeled particulate organic carbon in the ocean, Biogeochemistry: Open Ocean, https://doi.org/10.5194/bg-2021-201, 2021.

Johnson, Plant, J. N., Coletti, L. J., Jannasch, H. W., Sakamoto, C. M., Riser, S. C., Swift, D. D., Williams, N. L., Boss, E., Haëntjens, N., Talley, L. D., and Sarmiento, J. L.: Biogeochemical sensor performance in the SOCCOM profiling float array: SOCCOM BIOGEOCHEMICAL

SENSOR PERFORMANCE, J. Geophys. Res. Oceans, 122, 6416–6436, https://doi.org/10.1002/2017JC012838, 2017.

Mignot, A., D'Ortenzio, F., Taillandier, V., Cossarini, G., and Salon, S.: Quantifying Observational Errors in Biogeochemical-Argo Oxygen, Nitrate, and Chlorophyll *a* Concentrations, Geophys. Res. Lett., 46, 4330–4337, https://doi.org/10.1029/2018GL080541, 2019.

**REVIEWER #2**

We wish to thank Pr. Marcello Vichi for offering many insightful comments and helping us to clarify our results. Here we offer detailed responses to all questions. Reviewer's comments are in black, our replies are in blue.

General Comments:

This manuscript is indeed a valid compendium of diagnostics for assessing global ocean ecosystem models, which has been prepared with the aim to demonstrate the use of the multi-disciplinary dataset made available by the BGC-Argo array. The authors should thus be praised for their intention to bring together the community and follow the steps taken by Russel et al. (2018). However, that paper had different entry points, since it was specifically dedicated to a poorly sampled oceanic region and offered a multi-model analysis. This manuscript is well written and constructed, but only conveys a demonstrative message. I am thus not fully convinced by the scope of this present version of the manuscript, as well as by its effective novelty, since it does not add further knowledge to the existing literature [...]

Hence, I have carefully thought about how to write this review, and realised that the most relevant point of clarity would be to illustrate some cases of how readers could approach it. From a point of view of someone approaching modelling validation as a student or early career researcher, this manuscript offers a limited perspective, and one would gain more theoretical and methodological background in the 2009 JMS special issue (Lynch et al., 2009, and all the other papers in the issue), if not from earlier papers in the ecological modelling literature (Oreskes et a', 1994; Rykiel, 1996). If a reader is interested in the validation of the global version of PISCES, this manuscript is insufficient, because it provides a series of figures with few comments and discussions. It is surely of interest to the PISCES developers who are knowledgeable of the model details and possible deficiencies, but then an internal report would suffice. Finally, for experienced global ocean modellers, this manuscript is an illustration of the minimum

set of assessments (which I prefer to the term "validation") that serious modellers have been doing in the last ten years when evaluating their model results. In terms of "metrics", it gives indications to compare the model output against the state variables that can be measured by the array of floats and to add derived state variables from applications of artificial intelligence. Ultimately, the assessment is based on visual comparisons of coarsely gridded spatial maps and time series, or through the use of basic univariate scores (bias and RMSD) and cumulative diagrams that combine the same skill scores (e.g. the Taylor diagram, which also includes linear correlation).

REPLY: Thanks for the careful assessment of our work. The goal of this paper is to demonstrate the use of BGC-Argo floats for the evaluation of BGC models at the global scale, through a concise evaluation of the CMEMS global BGC forecasting system. Our hope is that the methodology employed in this study can be useful and informative for other research teams interested in model assessment with BGC-Argo floats. In particular, the main points we want to highlight are: 1) how do we handle BGC-Argo data (e.g., quality control and flags) for model assessment purposes, and 2) to propose BGC-Argo metrics, which we believe are useful to assess the accuracy of model states. We have intentionally chosen simple metrics, a minimum set of assessments and basic quantitative techniques (visual inspection, bias and RMSE) to focus the message of the study on the 2 points listed above and not on the evaluation of the model simulation. Therefore, this study is not designed as a review of biogeochemical modelling validation and it does not represent a thorough assessment of PISCES either.

We agree with the reviewer that the main message conveyed by the manuscript is not clear enough and that it can be confusing for the reader. Based on the reviewer's comments, we have modified the manuscript so that the main message of the study appears more clearly to the reader.

First, we changed the title to "*Using BGC-Argo floats for the assessment of marine biogeochemical models: a case study with CMEMS global forecasting system.* "

In the abstract, P1, L-28, we changed to "*Here, we demonstrate the use of the global array of BGC-Argo floats for the assessment of biogeochemical models through a concise evaluation of the CMEMS global forecasting system. We first detail the handling of the BGC-Argo data set for model assessment purposes, then we present 22 assessment metrics to quantify the consistency of BGC model simulations with respect to BGC-Argo data. The metrics evaluate either the model state accuracy or the skill of the model in capturing emergent properties, such as the Deep Chlorophyll Maximums (DCMs) or Oxygen Minimum Zones (OMZs). These metrics are associated with the air-*

*sea CO$_2$ flux, the biological carbon pump, and the oceanic pH and oxygen levels. Moreover, we suggest four diagnostic plots for displaying such metrics.*"

In the introduction, the paragraph starting P. 4, L2, changed to " *We aim to demonstrate the use of the BGC-Argo global array for the assessment of BGC models at the global scale. To that end, we performed a concise evaluation of the CMEMS global BGC forecasting system using the global fleet of BGC-Argo floats. We expect that the methodology employed here (from the data handling to the use of assessment metrics) would be useful and informative for other research teams interested in model evaluation with BGC-Argo floats.* "

The BGC-Argo data are certainly invaluable, and this is the reason why the community has strived to develop the technology and the financial support to deploy them. The authors did not however succeed in showing their enhanced value for model assessment, beyond the obvious consideration that this increases the number of data, which would be much more evident if this same assessment was done by comparing datasets with and without the contribution of the BGC-Argo.

REPLY: The reviewer brings up an interesting point. It is true that BGC-Argo dramatically increases the availability of data collected by traditional oceanographic cruises. It would indeed be informative to repeat the same assessment by comparing datasets with and without the contribution of the BGC-Argo, such as for example the World Ocean Atlas. While we are very interested in this question, we do not think it belongs to this paper, whose main focus is to show the use of BGC-Argo floats for model assessment rather than showing the impact of increasing the number of observations on skill scores.

In summary I have found two major issues with this manuscript that the authors have not considered to a satisfactory extent:
The loose definition of metrics and the absence of uncertainties' treatment. The authors use the term metrics in a rather ambiguous way. They also do not differentiate between measured data and artificially generated data. This implies that the evaluation process does not necessarily lead to an improvement of the model(s).

REPLY: We agree with the reviewer that our definition of metrics was somewhat ambiguous. In the introduction, we have changed our definition of metrics based on the recent review of Hipsey et al. (2020):

"*In this study, the BGC-Argo dataset is used in conjunction with the model evaluation framework developed by Hipsey et al. (2020). In particular, they propose three levels of*

*assessment metrics to evaluate the skill of a model simulation: state variables validation (e.g., Chla, nitrate, oxygen), mass fluxes and process rates validation (e.g., primary production, division rates), and emergent properties validation (e.g., deep chlorophyll maximum, oxygen minimum zones). In this study we present 22 metrics for the assessment of a model simulation with BGC-Argo data. Most of them evaluate the model state accuracy through the comparison of simulated state variables with BGC-Argo observations in the mixed layer or at fixed depth. In addition, some of the metrics assess the skill of the model in capturing emergent properties. These metrics are associated with the air-sea $CO_2$ flux, the biological carbon pump, the oceanic pH, and oxygen levels and Oxygen Minimum Zones (OMZs). Further, our validation framework could, in principle, include the second level of assessment metrics (i.e., flux and process). Indeed recent works demonstrated the feasibility of calculation at basin scale, from BGC-Argo observations, of mass fluxes and process rates, such as primary production, phytoplankton division and accumulation rates (Yang et al., 2021; Mignot et al., 2018), net community production (Plant et al., 2016), and carbon export (Dall'Olmo et al., 2016). However, it would be arduous to achieve such estimations on the global BGC-Argo dataset as it requires ad hoc calibration that cannot be easily defined.  As a consequence, the evaluation of simulated process rates with BGC-Argo data is not addressed in this study.*"

Concerning the reviewer's second comment, as we explain above, the object of the paper is not a thorough analysis of the model performance. Nevertheless, the proposed concise evaluation of the model (e.g., maps of RMSD) can be further exploited (e.g., by analyzing the spatial and temporal distribution of the RMSD maps or multivariate relationships of the errors) to investigate the model uncertainty sources.

Finally, we agree with the reviewer that we do not provide justification for mixing together measured data with artificially-generated data. We have added a paragraph in the Data section that justifies our choice.

"*Finally, we complemented the existing BGC-Argo dataset with pseudo-observations of $NO_3$, $PO_4$ , Si, and DIC concentrations as well as pH and $pCO_2$ using the CANYON-B neural network (Bittig et al., 2018). CANYON-B estimates vertical profiles of nutrients as well as carbonate system variables from concomitant measurements of floats pressure, temperature, salinity and $O_2$ qualified in "Delayed" mode together with the associated geolocalization and date of sampling.  The CANYON-B estimates of $NO_3$ and pH were merged with measured values on the rationale that CANYON-B estimates have RMS errors ($NO_3$ = 0.7 µmol $kg^{-1}$, pH= 0.013) (Bittig et al., 2018) that are of the same order of magnitude as those of the BGC-Argo ($NO_3$ = 0.5 µmol $kg^{-1}$, pH= 0.07) (Mignot et al., 2019; Johnson et al., 2017).*

*"Finally, we verified that the RMS errors of BGC-Argo data (both measured and from CANYON-B estimates) are lower than the RMS difference between the model and BGC-Argo data, so that the comparison of simulated properties with the BGC-Argo data leads to a meaningful evaluation of the model performance. We believe it is reasonable to draw conclusions on the model uncertainty from BGC-Argo data as long as the BGC-Argo errors are much lower than the model-observations RMS difference."*

The unconvincing enhancement of the effective role of BGC-Argo data in model assessment. Basically, the question I have is: why BGC-Argo are good enough and should be used separately and not as part of a global compilation of data such as the World Ocean Atlas? (which incidentally includes or will include the BGC-ARgo data). Since BGC-Argos are ultimately increasing the availability of data that are usually collected by means of traditional oceanographic cruises, what is indeed their value in model validation?

REPLY: We thank the reviewer for bringing this to our attention. We have carefully examined the documentation that deals with the BGC-Argo data processing in the WOD (https://www.ncei.noaa.gov/sites/default/files/2020-04/wod_intro_0.pdf) but we have not found sufficient information concerning the data mode used in the WOD. As we detail in the manuscript, the "Delayed-mode" represents the highest quality of data but for some variables, only a limited fraction of data is accessible in Delayed-Mode. Consequently, for each variable, we selected the highest quality of data (i.e., "Adjusted" or "Delayed-mode") that did not compromise too much the number of available observations. We are not sure whether such data selection is possible with the WOD, so we prefer to use the BGC-Argo data directly downloaded from Argo Coriolis Global Data Assembly Centre and not as part of a global compilation of data.

Furthermore, one of the issues of large databases such as WOD, is the interoperability of the data that compose them, and which, ultimately, affect their overall accuracy. Using the BGC-Argo dataset separately is a way to ensure consistent accuracy. The GLODAP V2 data set (on which CANYON B is developed) is an illustration of an interoperable homogenous data set (with very strict data QC procedure) used for model assessment and not used as part of a global compilation of data.

Finally, concerning the last of the above reviewer's questions, the BGC-Argo floats provide observations at high vertical and temporal resolutions and for long periods of time providing nearly continuous time series of the vertical distribution of a number of biogeochemical variables. This is not possible with discrete vertical samplings provided by cruise cast *in situ* measurements.

We have commented on the two last points discussed above in the 5ᵗʰ paragraph of revised Introduction.

For clarity, I would like to elaborate more on the first concept above, while the second point is mostly derived from the specific comments detailed in the next section. Russel et al (2018) also use the concept of metrics in a wider sense, although they define metrics as "any quantity or quantifiable pattern that summarizes a particular process or the response in a model to known forcings". The strength of the ACC transport at Drake Passage or the latitude of the maximum zonal mean winds over the Southern Ocean are "metrics" in this context. They are combinations of state variables, or values of state variables at specific locations.

In this context, all the surface state variables listed in Table 2, are indeed components of the biological carbon pump, but they are not metrics. They are simply state variables. Only when considered together to evidence emergent patterns they may give indications of proper process functionality (e.g. the ratio of particulate organic carbon to total chlorophyll, de Mora et al, 2016). I agree that the DCM and the "nutricline" (which would deserve a more appropriate definition, see specific points below) are "metrics", as well as the depth of the hypoxic layer. Mixing together indicators of processes with state variables is confusing, unless a rigorous link between a single state variable and the process is established.

REPLY: As we explained above, we have changed our definition of metrics, and in the new version of the manuscript we use the framework proposed by Hipsey et al. (2020). They propose three levels of assessment metrics to evaluate the skill of a model simulation: state variables validation (e.g., chlorophyll, nitrate, oxygen ), mass fluxes and process rates validation (e.g., primary production, division rates), and emergent properties validation (e.g., deep chlorophyll maximum, or oxygen minimum zones). We have inserted a new column in Table 2 to inform about the level each proposed metric is referring to. In Section 3, we have made a rigorous link between the state variable and the associate process in the section that defines the assessment metrics.

This manuscript increases the risk of misinterpretation by mixing together "metrics" and skill scores. Neither Russel et al (2018) and this manuscript expand on the concept of metrics performance and objective assessment (performance indicators, skill scores, cost functions, are all synonyms that depend on the specific discipline), which was instead done by Allen et al. (2007), Friedrichs et al. (2009), Vichi and Masina (2009) and others in the JMS special issue. For ease of simplicity, I will use the term skill score, which is the one used in the more mature field of weather forecasting. State variables can be assessed using univariate skill scores, and this is a necessary exercise for any

modeller to ensure the model has some grip with reality. Figure 3 and the other density plots in the Appendix give a visual indication of the skill score, but they do not quantify it (e.g. Smith and Rose, 1995; Rose and Smith, 1998). I also have another question linked to my Point 2 (and further detailed in the specific comments): why should this exercise be done only with the BGC-Argo and not also including the other existing data? Since BGC-Argo are evaluated against cruise cast benchmarks, then those data are usually considered always superior, and should be used. Again, the real value of the BGC-Argo would have been shown if the score had been substantially modified with the inclusion of the Argo data.

REPLY: In the revised manuscript, we have inserted Table 3 that quantifies the skill scores for each metrics as done in Vichi and Masina (2009) or Doney et al. (2009). As we explained above, we believe it is more reasonable to use the BGC-Argo data as a separate dataset rather than as part of a global compilation of data.

Specific comments:

P2L1 - Earlier work has specifically addressed the impact of assimilation on the carbonate system (Visinelli et al., 2017)

REPLY: Thanks for suggesting this reference. This study showed that the assimilation of physical data improves the simulation of alkalinity, DIC and $pCO_2$. However, a number of recent studies have shown that the assimilation of physical observations tends to degrade the simulation of BGC state (Fennel et al., 2019; Park et al., 2018; Gasparin et al., 2021).

P2L26-29 - This sentence is mixing together sensor accuracy, which has been assessed by Johnson et al and Mignot et al, in two specific regions of the world ocean) and temporal/vertical resolutions, which have not been assessed as far as I am aware. This is misleading. 10 days may not be sufficient for all variables, as well as the vertical binning that is done. The comparisons have assessed the equivalence between rosette casts and the floats, but they say nothing about the temporal and vertical resolution. For certain processes, such as carbon exchange and phytoplankton biomass through chlorophyll and backscattering proxies, a resolution of 10 days would lead to sampling aliases either of the mean or of the variability (Monteiro et al., 2015, Little et al., 2018). These are examples from the Southern Ocean, where there is the highest density of buoys.

REPLY: We have revised the sentence removing the part about the temporal and vertical resolutions.

P2L32-34 - The authors should be more specific. Other datasets, such as for instance remote sensing, are less limited in terms of temporal and spatial resolutions. This is connected to the concerns expressed in Point 1 above.

REPLY: We have revised the sentence, being more specific about the BGC-Argo resolutions (P3L33-P4L1).

P4L3-5 This sentence seems to imply that one can only perform point-by-point comparisons when there are few floats, which is odd. Again linked to my main Point 1 above. The authors should explain why given the current computing capability, they only suggest to perform diagnostics for few selected tracks and not for the overall dataset (Section 5.d).

REPLY: We have changed this paragraph based on point 1 and point 2 (see above), consequently this sentence was removed in the revised version of the manuscript.

P4L12-16 The connection between the variables and the ocean health/ecosystem functioning is not made explicit in the text. Taking as an example the ocean health index (http://www.oceanhealthindex.org/), establishing ocean health is obtained as a multivariate analysis of several data layers, forming a selected set of drivers and their associated thresholds. The authors should be more explicit about their intent here.

REPLY: Since we have changed our definition of metrics, we no longer refer to ocean health and ecosystem functioning in the revised version of the manuscript.

P5L12-13 This is not an objective criterion. What is an acceptable level of compromise?

REPLY: We have added an objective criterion in the revised manuscript:

*" However, for some variables, only a limited fraction of data is accessible in "Delayed-Mode". Consequently, for each variable, we selected the highest data modes, where at least 80 % of the data are available (see Table 1). Note that this criterion does not apply to $O_2$, where only delayed mode data were selected in order to generate the pseudo-observations from CANYON-B neural network (see after). "*

P5L22 There are many other relationships, and they have been shown to give different results (e.g. Thomalla et a., 2017l). The authors should explain why they are recommending this one.

REPLY: In the revised version of the manuscript, we now use a POC vs $b_{bp}$ relationship developed for the global ocean (https://catalogue.marine.copernicus.eu/documents/QUID/CMEMS-MOB-QUID-015-010.pdf) based on a global database of in situ POC and satellite $b_{bp}$ (Evers-King et al., 2017). This relationship, developed for global application, has been shown to outperform regional relationships, such as Cetinic et al. (2012), at global scales (P6L9-18).

P6L12-15 It appears that this method of linear resampling would artificially increase the number of data, and hence bias the statistical results, especially in conditions where there are not enough data.

REPLY: We thank the reviewer for raising this interesting issue. In the revised manuscript, we have commented on the possible bias introduced by the linear resampling method on our statistical results (P7L20-22).

P7L10-12 The authors do not discuss what would happen if the MLD is different between the observations and the model.

REPLY: The dynamical component, used in this study, has been extensively validated (Lellouche et al., 2013, 2018), and demonstrate to correctly represent variables that are constrained by observations (e. g., temperature and salinity), including Argo profiles. We verified that the MLD, which is calculated on a density criterion basis, is indeed correctly represented in the model. The global bias between the model and the BGC-Argo observations is 0.3 m. In the revised manuscript, we added a sentence that specifies that we verified the model skill in simulating the MLD (P8L22).

P7L29-30 Related to my point 1 above. The relationship between the state variables and the ecosystem functions is not made explicit. The term "useful" should be motivated.

REPLY: This section has been revised making the relationship between the state variables and ecosystem function more explicit. In addition, we have added new metrics for the mesopelagic layer as explained below.

"*The biological carbon pump is the transformation of nutrients and dissolved inorganic carbon into organic carbon in the upper part of the ocean through phytoplankton photosynthesis and the subsequent transfer of this organic material into the deep ocean. The functioning of this pump relies on key pools of nutrients and carbon as well as a number of processes that control mass fluxes between the pools.*

*The first level of assessment of a biological carbon pump simulated by a model consists in evaluating the different pools (or state variables) of the pump (Hipsey et al. 2020). In particular, the comparison of simulated surface nutrients ($NO_3$, $PO_4$, and Si), DIC, Chla and POC with BGC-Argo observations gives an indirect evaluation of the model capability to capture key processes of the biological carbon pump in the ocean upper layer,  such as primary production, respiration, and grazing.  A second-level, , assessment would be to directly compare these key processes with measured mass fluxes, but this assessment level is not addressed in this study. The surface  nutrients , DIC, Chla and POC (hereinafter denoted $sNO_3$, $sPO_4$, sSi, sDIC, sChl and sPOC) are calculated as the average concentrations in the mixed layer.*

*Similarly, the assessment of the mesopelagic nutrients, DIC and POC concentration (hereinafter indicated with the subscript $_{meso}$) provides an indirect evaluation of the key mesopelagic layer processes , such as export production, respiration, etc. The mesopelagic concentrations correspond to the depth-averaged concentrations between the base of the mixed layer down to 1000 m. "*

P8L7-8 Same as above, the value of DCM as an indicator should be contextualized. Why are BGC-Argo data providing a better estimate of this metric than other data?

REPLY: In the revised manuscript the use of the DCM as an indicator is better contextualized:  "*In stratified systems, a Chla maximum (hereinafter denoted Deep Chlorophyll Maximum, DCM) is formed at the base of the euphotic layer (Barbieux et al., 2019; Cullen, 2015; Letelier et al., 2004; Mignot et al., 2011, 2014). It has been suggested that the DCM plays an important role in the synthesis of organic carbon by phytoplankton (Macías et al., 2014). DCMs are therefore important features to be assessed in BGC models with respect to processes involved in the biological carbon pump processes such as the primary production, however the DCM layer generally escapes detection by remote sensing. Furthermore, DCMs are also an emergent feature that develops in response to complex physical and biogeochemical interactions (Cullen, 2015). Thus, their evaluation provides critical information regarding the accuracy of the model in capturing complex patterns of key ecosystem processes.*"

As we explain above, the BGC-Argo data provide consistent profiles at high vertical and temporal resolution allowing to derive time-series of DCM depths. In comparison, discrete vertical samplings provided by cruise cast *in situ* measurements have a vertical resolution much lower (10 samples taken over a 100 m layer ), without repetitive sampling.

P8L13 Please explain what H is.

REPLY: It is an omission on our part. H is the mixed layer depth. We have replaced H by MLD.

P8L14-16 This may be confusing for some readers, since it's not technically a gradient. The cited paper uses and justifies this definition. I'd suggest the authors to be more precise and give their definition and how this is an effective metric of the carbon pump. Also, there is a difference in sampling between argo and the layers of discrete models. How is this taken into account?

REPLY: We have provided a more precise definition of the nitracline depth in the revised manuscript, and we described how this is an effective metric of the carbon pump:

"*The vertical supply of $NO_3$ to the surface layers is a critical process of the biological carbon pump as $NO_3$ is often depleted in the surface layers and is a limiting factor for phytoplankton growth in most oceanic regions. The $NO_3$ vertical supply depends, among other factors, on the vertical gradient of $NO_3$ (the nitracline), and, in particular, on its depth (the nitracline depth)* (Cermeno et al., 2008; Omand and Mahadevan, 2015). *Therefore, the comparison of the simulated nitracline depth with BGC-Argo observations allows for an indirect assessment of the model quality in reproducing vertical fluxes of $NO_3$. Following previous studies* (Cermeno et al., 2008; Lavigne et al., 2013; Richardson and Bendtsen, 2019)*, the depth of the nitracline corresponds to the first depth where $NO_3$ is detected. The detection threshold was set to 1 µmol/kg, which corresponds to an upper estimate of BGC-Argo $NO_3$ data accuracy* (Johnson et al., 2017; Mignot et al., 2019). "

Finally, there is indeed a difference in sampling between the BGC-Argo and the layers of discrete models. This is clearly visible in the scatterplot for the nitracline, the DCM and the OMZ depths.

P8l28-30 At P4L11 it is reported "depth of the OMZ". This the depth of the oxygen minimum. It should be explained how and why this is a good indicator, and why the BGCArgo data are superior in its identification.

REPLY: In the revised version of the manuscript, we explain why the depth of the oxygen minimum is a good indicator. "*Oxygens levels in the global and coastal waters have declined over the whole water column over the past decades (Schmidtko et al., 2017) and OMZs are expanding (Stramma et al., 2008). Assessing how models*

*correctly represent ocean oxygen levels as well as the OMZs is therefore critical to monitor their changes over time. Similarly to DCMs, the assessment of OMZs is also informative on how the model simulates emergent dynamics as OMZs originate from complex physical and biogeochemical interactions (Paulmier and Ruiz-Pino, 2009).* ""

As we detailed in a previous reply, BGC-Argo floats are particularly fit in the identification of vertical characteristics of BGC variables.

P9L26 This statement about non-linearity is odd in the context of model goodness-of-fit (Smith and Rose, 1995; Pineiro et al, 2008; Vichi and Masina, 2009). If it's non-linear, then the assessment is failed.

REPLY: We have removed the sentence.

P10-8-12 The choice of the binning interval should be discussed. What is the advantage of losing the variability measured by the floats? Why not using the standard deviation as an indicator of the model skill to reproduce the proper scales? These are enhanced features that only the BGC-Argo data would allow to compute.

REPLY: We discuss the choice of the binning interval in the revised version of the manuscript. "...*To do so, the metrics from 2009 to 2017 are averaged in 4°x4° bins, excluding those with less than 4 points. The 4° distance is an upper estimate of the autocorrelation length scales for $O_2$, nutrients, and $pCO_2$ (comprised between 300 and 400 km) between 20° and 40° of latitude in both hemispheres* (Biogeochemical-Argo Planning Group, 2016)*.*"

Moreover, in section 4.c we have commented about using standard deviation maps as an indicator of the model skill in properly reproducing variability scales. However, we won't show standard deviation in the manuscript as we prefer to not overload Figure 4 and the associated supplementary figures with additional panels.

P10L22-24 Allen et al (2007) warned against the visual comparison of time series. This sentence is generic and should be explained in the context of the augmented data provided by the BGC-Argo.

REPLY: We agree with the reviewer that visual inspection relies on the subjective appreciation of the evaluator. Consequently, we have added normalized skill scores to Figures 5 and 6.  Moreover, we have added the following sentence at the end of the section 4c. "*In addition to the time series of metrics, we also display normalized skill*

*scores such as percent BIAS and RMSD as a function of season in order to combine quantitative metrics with visual comparison.*"

P11L11-14 The results are not presented according to the concept of the biological carbon pump "metric". It is evident that the nutrients are correlated while all carbon flux variables are not performing. Which ultimately questions the use of surface nutrients as indicators of carbon cycling.

REPLY: The fact that nutrients are well represented in the model suggests that the model captures the combination of process rates that drive nutrients dynamics. Some of these process rates drive both the nutrients and carbon dynamics, but there are also rates that are specific to each state variable. This probably explains why the carbon variables are not performing while the nutrients are well simulated. However, it must be recognized that without a direct assessment of the individual rates, we cannot verify this hypothesis. We have clarified this point in the revised version of the manuscript in Section 5a (P13L27-32-P14L1-14).

P11L31 I cannot see the data "around" the line. I rather see an overestimation. (it is either Cape Verde or Cap Vert)

REPLY: We have improved the clarity of the figure in the revised version of the manuscript.

P12-L2-17 Linked to Point 2 above. The authors seem to imply that BGC-Argo data are more suitable than ocean colour for model assessment. I acknowledge that this is not explicitly written, but there is no clear rationale. This kind of map would certainly be superior in terms of spatial and temporal resolution when using that product as Benchmark.

REPLY: Indeed, such a map would be superior in term of spatial and temporal resolution. However, the BGC-Argo data allows to assess the skill of the model in estimating Chla concentration, when ocean color data are not available, i.e., during cloudy days and during winters at high latitudes.

P12-section-d This is the section that mostly led to the inclusion of Point 2 above. The shown time series is 2 years long, which is an invaluable source of data from a region that has been influential in shaping our understanding of the spring bloom. I am missing the point why the authors are writing the term spring bloom in quotes. The advantage of time series from floats that remained in a given province of the global ocean is of huge potential in model validation. The offered description is quite generic, which could have

been done even using monthly climatological time series obtained from the WOA, or from the existing long-term observational ocean sites (BATS, PAPA, HOT). The BGC-Argo floats are an unprecedented source of multiple opportunities to do validation in several regions of the world ocean (with some limitations), but this present form of the manuscript does not offer any specific recommendation of what numerical modellers should do to unleash this potential. I would be very interested in seeing an exploitation of the multivariate nature of BGC-Argo, while I only see multi-panel plots.

REPLY: Based on this comment, we have revised this section. We have removed the unnecessary description of the spring bloom, while highlighting the invaluable opportunities of such time series for the assessment of models by showing other time series in an oligotrophic region where in situ data are scarce. Concerning the evaluation of the multivariate nature of BGC-Argo, we agree that it is an interesting point to pursue. We are very interested in applying the multivariate approach proposed by Allen et al. (2007) to the BGC-Argo data set. However, we prefer to focus this manuscript on the presentation of the metrics and to exploit the multivariate approach in another study.

P13L4-5 The authors should do more than simply say "correctly represented". This is a subjective statement, which is based on a visual comparison, exactly what the community challenged in the last 10-15 years. The advantage is that now we can use a frequency of 10 days, when initially phenology analysis was based on monthly data. Again, the authors are missing an opportunity to demonstrate the intrinsic value of this new data set.

REPLY: As suggested by the reviewer, we have included normalized skill scores to avoid relying only on subjective visual inspection. We agree that the frequency of 10 days is a significant progress over previous data sets. However, as explained in the conclusion, we do not address phenology metrics in this study because the number of observations per month and per bins is still too low to perform a global analysis.

P13-L13-20 This is a more detailed analysis of this specific model, which indeed brings in some of the advantages of a multivariate data set. However, there is a combination of measured and derived variables, which are treated as if they were equivalent. Quite a few questions come to mind: Is there a possibility that there is artificial correlation in the derivation of the phosphate and silicate concentration? What is the error associated with the CANYON-B method? Which is the effective (measured) variable mostly responsible for the response of the other estimated nutrients? The reduced consumption occurs during the spring period, and is continued during summertime. Hence, there is a factor at play during the late spring period, which is less likely to be reduced uptake from

smaller phytoplankton during summer as suggested. It may thus be a delayed onset of the phytoplankton succession, or maybe a faster remineralization occurring in the upper layers, which retain more inorganic nutrients closer to the surface. This may indeed be beyond the scope of the manuscript, but it has been the authors' decision to propose some mechanistic explanations of this discrepancy. Showing a complete example of how the use of multivariate data allows modellers to investigate model deficiencies would offer guidelines to other modellers.

REPLY: As explained above, we have included a paragraph in the Data section that discusses the error associated with the CANYON-B method. Concerning the second comment, we have removed the mechanistic explanation of this discrepancy. As suggested by the reviewer, we agree that this is beyond the scope of the manuscript.

P13-L22-23 This sentence bears lots of assumptions. This is really where BGC-Argo can make a difference. The related uncertainties should however be highlighted, together with recommendations to other modellers on how to best approach the assessment of the carbon cycle metrics.

REPLY: Please, see next REPLY.

P13L26-29 This argument is flawed. If the occurrence of the peak is matched in the mesopelagic layer rather than at the surface, it is a clear indication of vertical mismatches in the export. I would thus argue that POC concentration is a proper metric for the export component of the carbon cycle. I would again encourage the authors to replace the use of subjective terms such as "consistent" with objective indicators (see Allen et al., 2007). For instance the comparison of the skill score computed in two consecutive years would give indication if there is some variability or if the model tends to repeat the same pattern.

REPLY: We have removed the conclusions about the oceanic carbon cycle and POC export flux, and we have removed the time series of SDIC, sPOC and $POC_{meso}$ in Figure 5. As pointed out by the reviewers #1 and #2, this paragraph bears a lot of assumption and we don't have sufficient information to assess the model skill in simulating the process rates that drive sPOC, and $POC_{meso}$.

P14L16-19 I would recommend more clarity on this statement. Are these sensors not available on the global ocean floats? It is not clear why this example is presented for Mediterranean floats, and not introduced earlier as one major advantage of the BGC-Argo floats.

REPLY: We have clarified this statement adding that the sensors are available on the global ocean. However, the global model used in the study does not resolve the spectral and directional properties of the underwater light field. That's why we didn't use the global model but a model of the Mediterranean Sea equipped with a multispectral light module, as clarified in the new manuscript version.

P14L26-28 This sentence is similar to the statements done in the earlier sections. This is not technically a perspective statement.

REPLY: We agree with the reviewer that this section does not provide a perspective statement, thus we have added a perspective statement at the end of this section( P17L1-2).

P15L1-6 The question is whether these data should be used "on their own" or in conjunction with the other existing datasets. The authors should clearly explain in the conclusion why this dataset should be exploited as a separate unit.

REPLY: As explained above, in the introduction we have added motivation about using this dataset as a separate unit.

P15L32-P16L3 I would thus recommend the authors to thoroughly address the issue of how the uncertainties should be treated. This is particularly important in the case of mixing measured and derived variables. If BGC-Argo are capable, within their limits, to reduce uncertainties in model assessment exercise, this should be adequately argumented. The fact that there are more data available is undoubtedly of relevance, but I wonder if it does help to reduce uncertainties in model states.

REPLY: As explained previously, we have added a paragraph in the Data Section that provides justification for mixing together measured data with artificially-generated data. We also removed the paragraph about fluorescence quenching as it can be misleading for the reader. As discussed above, we have verified that the RMS difference between model and BGC-Argo Chla is always lower than the BGC-Argo Chla RMS error, so that the comparison of simulated Chla with the BGC-Argo Chla leads to an evaluation of the skill of the model in simulating Chla concentrations.

P16L15-18 Please highlight in which part of the results this is shown.

REPLY: We have highlighted in which part of the results this is shown (P18L21).

P17L2 Please add in the caption the meaning of the codes (or a link to where they are

explained more in detail). Also, in the heading of the 3rd column, correct Date with Data.
Figure 2 Taylor diagrams are based on geometric properties of the circle. Hence they
should be presented using equal axes.

REPLY: In the revised manuscript, we have added the meaning of the codes, changed
Date with Data and presented the Taylor diagram using equal axes.

---

## Author Response (AR2)

We wish to thank Pr. Marcello Vichi for offering many insightful comments and helping us to improve the manuscript. Here we offer detailed responses to all questions. Reviewer's comments are in black, our replies are in blue.

General Comments:

There is some need to further strengthen the concept of why the BGC-Argo data should be considered the most appropriate reference dataset for global model assessment, and how they relate to the other existing datasets (especially satellites, which are going to be superior for evaluating surface chlorophyll than BGC-Argo; see my comment 4 below). There is little doubt that the BGC-Argo program will become a reference climate data record in the longer term. Maybe the authors should provide some clearer recommendations to the readers in their final section. As it stands, the conclusion section appears truncated, with a series of comments that one would mostly expect in a report rather than in a journal article (see in this regard my comment 3 below).

REPLY: We have strengthened the concept of why we use the BGC-Argo float as the unique reference dataset in our study, in the abstract, and introduction.

In the abstract:
 *"The use of BGC-Argo observations as the single evaluation data set ensure the accuracy of the data as it is an homogenous data set with strict sampling methodologies and data quality control procedures"* .

In the introduction:

*"The BGC-Argo data set represents a significant improvement for the assessment of models comparing to large databases such as the World Ocean Database (WOD) (Boyer et al., 2013) or the Copernicus Marine Service in situ dataset (European Union-Copernicus Marine Service, 2015). Large databases are composed of data collected from various instrument types with heterogenous data sampling methodologies. Therefore, for a given variable, the accuracy numbers are not the same and change depending on the instrument type (European Union-Copernicus Marine Service, 2019). Consequently, this affects the overall accuracy over time due to the changing proportion of instrument types over the years. On the other hand, the BGC-Argo data set is an homogenous data set with strict and uniform sampling methodologies and data Quality-Control (QC) procedures. As a result, the BGC-Argo data set have a satisfactory level of accuracy, which remains stable over time (Johnson et al., 2017; Mignot et al., 2019).*

*Moreover, the number of quality-controlled observations collected every year by the BGC-Argo fleet is now greater than any other data set (Claustre et al., 2020). Using the BGC-Argo dataset as the single evaluation data set is therefore a way to ensure consistent accuracy. "*

We have also provided  recommendations in the manuscript and in the conclusion how to relate the BGC-Argo and satellite Chl*a*.

*"While the assimilation decreases the model-BGC-argo data misfit for $Chl_{mixed}$ comparing to a simulation without assimilation (not shown), the model errors for the three metrics associated with Chla remains systematically larger than the BGC-Argo variability. Yet, it has been shown that, when comparing to the satellite Chla product assimilated (European Union-Copernicus Marine Service, 2022), the model-satellite misfit was lower than the variability of the satellite data (European Union-Copernicus Marine Service, 2019). This suggest that the model-BGC-Argo data misfit could originate, in part, from discrepancies between the satellite Chla product assimilated and the BGC-Argo data. We propose that studies should check the consistency between ocean colour products and BGC-Argo Chla products at the global scale as these two products are expected to be assimilated together in future operational BGC systems (Ford, 2021)."*

We have also rewritten the conclusion entirely.

Section 3 is still confusing. I apologise with the authors if this is due to my own limitation, but I feel there could be other readers raising questions like mine. Somehow, the previous version of the manuscript was clearer, although I realize that this may be a consequence of all the other changes in this revision. I would suggest the users be clear with their definitions. They now indicate that 22 metrics can be extracted from the BGC-Argo datasets, but they do not explain clearly that these metrics have been grouped according to key components/processes of marine ecosystem functioning (i.e. the 4 sub-sections presented in Sec. 3). This grouping is evident in Table 2, but the text is unsatisfactory. The confusion is further augmented by naming one of the key processes "Oceanic pH" (one of the metrics) instead of "Ocean acidification". The authors say: "The metrics are described below", but actually they first describe the processes, and then how the metrics derived from the BGC-Argo data can be used to quantify these processes.

REPLY: We agree with reviewer. We have rewritten the section that defines the assessment metrics and made the grouping more evident in the text.

They should also explain why certain metrics are included in one grouping rather than another. For instance, the surface partial pressure of CO2, which is essential for estimating the air-sea flux, can be computed from pH and DIC, which have been included in two different groups. It is true that inorganic carbon is linked to both the physical solubility pump and the biological carbon pump, and this ambiguity should be recognized.

REPLY: We agree with the reviewer that DIC is linked to both the physical solubility pump and the biological carbon pump. However, it is now included in the carbonate chemistry metric considering that the classical variables for the study of carbonate chemistry are DIC, Alk, pH and $pCO_2$ (Williams and Follows, 2011).

I am (now) aware of the main intent of this manuscript. However, more effort should be put into demonstrating that this exercise is a contribution to the literature on global biogeochemical models and their assessment, rather than a report that could have been produced by CMEMS as part of their operational endeavour. For this reason, I would recommend the authors to improve their description of results, which is often written as a dry reporting of the model discrepancies. This is instead well done in Sec. 6, which is now very clear and combines the demonstrative aims with the provision of some directions for future research and/or analyses. I have given some more specific comments in the next section.

REPLY: We thank the reviewer for bringing this to our attention. The main text of the manuscript has been largely revised, and we have improve the descriptions of the results. We now provide directions for future research and/or analyses that are summarized in the conclusion:

"

*Overall, the model surpasses the BGC-Argo climatology in predicting pH, DIC, Alk and $O_2$ in the mesopelagic and the mixed layers, as well as $NO_3$, Si and $PO_4$ in the mesopelagic layer. Concerning the other metrics, whose model predictions are outperformed by the BGC-Argo climatology, we provide suggestions to reduce the model-data misfit and thus to increase the model efficiency. For, $PO_4$, Si, and $NO_3$, we propose to test if the uncertain model error covariances during the assimilation of satellite Chla could lead to a degradation in predicting nutrients in the mixed layer. For Chla-related metrics, we recommend to check the consistency between ocean colour products and BGC-Argo Chla products at the global scale as it may explain part of the*

*misfit between the model, that assimilates satellite Chla, and BGC-Argo observations. The discrepancies between modelled and observed POC and OMZs have been already investigated in previous studies. It has been suggested that improving the BGC-Argo POC-$b_{bp}$ conversion factor, tuning the model parameters and implementing missing processes in the model structure could decrease the model-data inconsistencies associated with POC dynamics. Similarly, the improvement of the ocean circulation in physical models should improve the accuracy of OMZs model predictions. Finally, $pH_{mixed}$ and $pCO_{2\ mixed}$ should be better modelled if the uncertainties associated with DIC, Alk, temperature and salinity in the mixed layer are reduced.*

*The method proposed here is also beneficial to inform about the BGC-Argo network design. In particular, the regions where BGC-Argo observations should be enhanced to reduce the model-data misfit through the assimilation of BGC-Argo data or process-oriented assessment studies. We strongly recommend to enhance the Arctic region, which is critically under sampled and is constantly outperformed by the BGC-Argo climatology. Likewise, BGC-Argo observations should be enriched in the Equatorial region and in the Southern Oceans, two regions where the model error barely exceed the BGC-Argo observations variability."*

The authors rightly claim the unicity of this data set as well as its multivariate nature. However, this is not always put into practice in a demonstrative sense. I am particularly critical with Section 5.c, in which surface Chl is presented as an example of the maps. Why using sChl as the demonstrative metrics? This field is far better represented in terms of temporal frequency and spatial coverage by the satellite record and I'm sure the authors would recommend modellers to use this product for their validation. I am also sure satellite Chl has been used thoroughly before making the CMEMS model publicly available as a shared product.

REPLY: We thank the reviewer for bringing this to our attention. We have revised the methodology of the study and we have followed the approach of Allen et al. (2007) to put into practice the multivariate nature f the BGC-Argo data. The new methodology is summarized in the abstract:

*" Here, we propose a new method to inform about the model predictive skill in a concise way. The method is based on the conjoint use of a K-means clustering technique -- an unsupervised learning algorithm, assessment metrics and BGC-Argo observations. The K-mean algorithm and the assessment metrics reduce the number of model data points to be evaluated. The metrics evaluate either the model state accuracy or the skill of the model in capturing emergent properties, such as the Deep Chlorophyll Maximums or*

*Oxygen Minimum Zones. The use of BGC-Argo observations as the single evaluation data set ensure the accuracy of the data as it is an interoperable homogenous multivariate data set with strict data quality-controlled procedures. The method is applied to the Copernicus Marine Service global forecasting system. The model performance is evaluated using the model efficiency statistical score that compare the model-observations misfit with the variability of the observations, and thus objectively quantifies whether the model outperforms the BGC-Argo climatology."*

We cannot use the satellite Chl*a* for the validation of the model, as this product is already assimilated in the model.

I question the decision to not include in the main text one of the other variables that would not be available without the BGC-Argo dataset and CANYON-B. They are in the Appendix, and to me far more informative than sChl. As a modeller, my main question when reading this section is not how relevant the BGC-Argo dataset is to assess model performances, but rather how surface Chl from that dataset compares with the satellite record. This issue also applies to the results presented for the Atlantic time series in Sec 4.d. Why choose variables that have previously been used to assess models (nutrients and chlorophyll), instead of selecting new variables such as pH, DIC and POC, which would definitely give information on the processes of interest. In this case, these figures are not provided in the Appendix, which is a missed opportunity to demonstrate one of the main aims of this paper. If this is done because these results are not very good, then it is even more worrying.

REPLY: We agree with the reviewer. We have now included in the main text all variables derived from the BGC-Argo dataset and CANYON-B.

Specific comments
P1 L20-22 This is a generic statement for an abstract. The same can be said of BGC-Argo data, since rates are also not directly measured

REPLY: We have removed this sentence.

P2 L7 Has taken

REPLY: Thank you, we have made the correction.

P3 L7 All datasets are incomplete and have limitations, including the BGC-Argo

REPLY: We agree but we did not imply that the BGC-Argo dataset has no limitations.

P3 L30 Please explain why these AI methods cannot be applied to the other datasets

REPLY: These AI methods can also be applied to datasets that include temperature, salinity and oxygen measurements.

P4 L4 The dataset represents

REPLY: Thank you, we have made the correction.

P5 L2-L5 This sentence is unclear. What does it mean to be arduous? Is it a problem with the data set? Should the readers abstain from attempting it because it would not be possible? This sentence would be understandable if further discussed in the conclusions. As it stands, it seems the authors are justifying themselves for not having done it.

REPLY: We have removed this sentence.

P5 L19-20 There is a need to clarify from the beginning which are the variables directly measured with the on-board sensors of the BGC-Argo devices (primary variables?) and which ones are further derived (secondary or derived variables?). This would help in understanding the author's definition of metrics. This is further complicated in the reminder because some variables are a combination of derived and measured products (pH, NO3), and it is not always clear what is the percentage (for instance, in Sec. 5.d).

REPLY: We agree with the reviewer that some variables are primary while others are secondary. However, for simplicity, the variables derived directly from BGC-Argo or CANYON-B are mixed together. This is justified in the data section: "

"Finally, we complemented the existing BGC-Argo dataset with pseudo-observations of $NO_3$, PO4 , Si, and DIC concentrations as well as pH and pCO2 using the CANYON-B neural network (Bittig et al., 2018). CANYON-B estimates vertical profiles of nutrients as well as the carbonate system variables from concomitant measurements of floats pressure, temperature, salinity and O2 qualified in "Delayed" mode together with the associated geolocalization and date of sampling. The CANYON-B estimates of $NO_3$ and pH were merged with measured values on the rationale that CANYON-B estimates have RMS errors ( $NO_3$ = 0.7 µmol $kg^{-1}$ , pH = 0.013) (Bittig et al., 2018) that are of the same order of magnitude as those of the BGC-Argo observations errors ( $NO_3$ = 0.5 µmol $kg^{-1}$, pH = 0.07) (Mignot et al., 2019; Johnson et al., 2017) .

*Finally, we verified that the RMS errors of BGC-Argo data (both measured and from CANYON-B estimates) are lower than the RMS difference between the model and BGC-Argo data, so that the comparison of simulated properties with the BGC-Argo data leads to a meaningful evaluation of the model performance. We believe it is reasonable to draw conclusions on the model uncertainty from BGC-Argo data as long as the BGC-Argo errors are much lower than the model-observations RMS difference."*

P7 L1-2 This is a very relevant addition. However, I do not see this concept further used in the presentation of the results. It is for instance not discussed when showing the RMSD in Fig. 5 and 6.

REPLY: We now use the model efficiency statistical score to assess the model performance.

*"The model efficiency tests whether the model outperforms the BGC-Argo climatology ($0 < m_e < 1$ ,Fennel et al., 2022), or stated differently, if the model-data mean square difference is lower than the observation variance, i.e., $\sum_{i=1}^{N}(m_i - o_i)^2 < \sum_{i=1}^{N}(o_i - \bar{o})^2$ ."*

P7 L20-22 This sentence is not connected with the following. It is customary to use the lower time frequency or coarser spatial resolution when comparing data and models (as done with the spatial maps in the results section). Why did the authors decide not to use weekly averages of the Argo data?

REPLY: We have connected the sentence with the following. We did not use the weekly averages because all model variables except POC are available as daily values.

P7 L25 to match

REPLY: Thank you, we have made the correction.

P7 L30 Was this done using the daily interpolation?

REPLY: Yes, it was done using the daily interpolation.

P12 L13 I would suggest using sparseness rather than scarcity. Argo data are still scarce.

REPLY: This sentence was removed during the rewriting of the main text.

P12 L21-22 Unclear sentence. Does it mean that showing this would confound the reader?

REPLY: This sentence was removed during the rewriting of the main text.

P13 L14 This section is presented as a technical report. I would recommend adding a few more sentences that point at the relationship between the metrics and the processes in section 3. For instance, when referring to the oxygen levels, make an explicit connection with sec. 3.a, and the same with the other variables. I think the value of the message would be further enhanced if there is a more direct connection between Sec. 3 and Sec. 5. The demonstrative aim of the manuscript is clear, but because there is no discussion section it would help to have some additional comments. Many questions arise, for instance, why Chl performs badly while nutrients don't, while DIC is also good and spCO2 and spH are similarly worse? I am not asking the authors to offer full explanations since this would be beyond the scope of the work, but the indication that the BGC-Argo data help to highlight these discrepancies, which would not be possible with other datasets.

REPLY: This section was removed during the rewriting of the main text. In the new version of the manuscript we have made a direct connection between the variables when assessing the model.

P13 L20 close to the

REPLY: This sentence was removed during the rewriting of the main text.

P14 L2-4 This is another sentence that would be improved through references and linkages to Sec 3.

REPLY: This sentence was removed during the rewriting of the main text.

P14 L12 as well as

REPLY: This sentence was removed during the rewriting of the main text.

P14 L16 There is also a lack of sensitivity in the model for very low oxygen regions close to 0 umol kg-1. The model can have any number between 0 and 30 umol kg-1

when observed values are close to 0 umol kg-1. The feature reported in the text is relevant but the number of data is not very high. While the discrepancy around zero has a higher data density.

REPLY: This section was removed during the rewriting of the main text.

P14 L17 Cape Verde (https://en.wikipedia.org/wiki/Cape_Verde)

REPLY: This section was removed during the rewriting of the main text.

P14 L29 Figure 1 shows data counts, not Chl patterns. Please clarify.
REPLY: This sentence was removed during the rewriting of the main text.

P14 L31 Please explain the meaning of coherent. This should not be the first time this model is assessed against surface chl from satellites.

REPLY: This section was removed during the rewriting of the main text.

P14 L34 This is another comment that I would expect in a report. My understanding is that the aim of the work is to highlight what can be learned from the use of BGC-Argo data that is not possible with other datasets (e.g. satellite data).

REPLY: We have clarified the aim of the work in the introduction:

*"The objectives of the present study are twofold. Our first aim is to propose a methodology that uses the BGC-Argo data set, an unsupervised learning algorithm and assessment metrics to simplify marine BGC model-data comparisons, and thus inform, in a concise way, about model performance. The second objective is to use this methodology to also identify ocean regions where the model-observations misfit is larger than the variability of the BGC-Argo data and thus inform the BGC-Argo observing system of regions that should be better sampled."*

P15 L6 Is there a reason for using quotation marks for the spring bloom?

REPLY: This sentence was removed during the rewriting of the main text.

P15 L18-22 This is another dry sentence used for a major misestimation, which would require some more context or a brief discussion. The percentages are extremely high. I am not questioning the model quality, rather the value of offering interpretations based on the assessment exercise.

REPLY: This sentence was removed during the rewriting of the main text.

P15 L28 Please indicate if these percentages are satisfactory with respect to the reference uncertainties indicated in the methods (P7 L1-2 and previous lines). This comment also applies to the previous point.

REPLY: This sentence was removed during the rewriting of the main text.

P15 L30-32 It would be helpful if the authors could add some comments on how the multivariate data from BGC-Argo allow to constrain models in a way that was sparse and more difficult 15 years ago. Consider for instance Vichi, Masina and Navarra (2007), in which all possible existing data were used to assess a global ocean BGC model. There is no need to add this reference, it's just one of the examples of how model assessment has been done in the literature.

REPLY: We have commented on the multivariate aspect of the BGC-Argo data in the introduction:

*"The BGC-Argo floats also provide multivariate observations at high vertical and temporal resolutions and for long periods of time providing nearly continuous time series of the vertical distribution of several biogeochemical variables. This is not possible with discrete, univariate vertical samplings provided by cruise cast in situ measurements or from climatological values derived from the WOA. All these specificities overcome the limitations of the previous datasets, especially with respect to their univariate nature, as well as their limited vertical and temporal resolution. This opens new perspectives for the evaluation of BGC models(Gutknecht et al., 2019; Salon et al., 2019; Terzić et al., 2019)."*

P17 L10 I suggest to use "limited" instead of lack

REPLY: This sentence was removed during the rewriting of the main text.

P17 L11 Increased number is not the only advantage. They are coherent, consolidated and sustainable. They could become equivalent to the concept of climate data records used for satellite data.

REPLY: This sentence was removed during the rewriting of the main text.

P17 L20-21 I would suggest to refer to the processes presented in Sec. 3

REPLY: This sentence was removed during the rewriting of the main text.

P21 Table 2 Please clarify in the section text if the definition used here is the same for both the model and the data

REPLY: We have clarified in the text that the definition of the metrics is the same for the model and the BGC-Argo data.

Fig. 4 and all the maps. It would be very helpful to add the maximum and minimum values of the range in the colorbar, to better understand the spread of data values

REPLY: This maps were removed during the rewriting of the main text.

################################################################################
################################################################################

We thank the reviewer #3 for their thoughtful comments. Here we offer detailed responses to all questions. Reviewer's comments are in black, our replies are in blue.

The manuscript "Using BGC-Argo floats for the assessment of marine biogeochemical models: a case study with CMEMS global forecast system" by Mignot et al. proposes 22 metrics for the assessment of biogeochemical models and applies them to a single model. As such, the analysis is a very welcomely comprehensive application of ocean BGC Argo observations, but is done in a vacuum without reference to previous or alternative modeling efforts. While this approach is fine from a technical report documentation perspective, it does not fit the standard of a scientific research paper. As such, it would seem more appropriate for "Geoscientific Model Development" than "Biogeosciences" in its present form.

REPLY: We have rewritten the main text of the manuscript in order to transform the manuscript into a scientific research article. In the introduction, the problematic is stated more clearly:

[revised manuscript text omitted]

The null hypothesis for establishing that the model is "good" should be defined.

REPLY: We now use the model efficiency statistical score to assess the performance of the model. The null hypothesis for establishing that the model is "good" is tested against the BGC-Argo climatology:

*"The model efficiency tests whether the model outperforms the BGC-Argo climatology ($0 < m_e < 1$ ,Fennel et al., 2022), or stated differently, if the model-data mean square difference is lower than the observation variance, i.e., $\sum_{i=1}^{N}(m_i - o_i)^2 < \sum_{i=1}^{N}(o_i - \bar{o})^2$ . "*

Also, there are some really interesting of the value and needs for BGC Argo observations in the conclusions that are completely unsupported by the body of the manuscript… if the authors want to bring some of this Appendix material into the manuscript body so as to support these conclusions, (and leave the focus on just the

current model) that would also be an appropriate means of turning the paper from a technical report on diagnostics into a scientific research paper.

REPLY: We thank the reviewer for this suggestion. All the Appendix material are now integrated into the manuscript body. We have made the design of the BGC-Argo observing system one the main objective of the study as summarized in the conclusion:

"

*The method proposed here is also beneficial to inform about the BGC-Argo network design. In particular, the regions where BGC-Argo observations should be enhanced to reduce the model-data misfit through the assimilation of BGC-Argo data or process-oriented assessment studies. We strongly recommend to enhance the Arctic region, which is critically under sampled and is constantly outperformed by the BGC-Argo climatology. Likewise, BGC-Argo observations should be enriched in the Equatorial region and in the Southern Oceans, two regions where the model error barely exceed the BGC-Argo observations variability.* "

Finally, the paper includes a multitude of language mistakes which I have tried to rectify in my technical comments.

REPLY: Thank you very much.

Technical comments:
1-16 – "a major tool" should be "major tools"

REPLY: Thank you, we have made the correction.

2-1 – "or" should be "and". Also, is there a difference between "These metrics" and "The metrics in the sentence before? If not, ". These metrics" should be "and"

REPLY: Thank you, we have made the correction.

2-3 – "suggest" seems an odd word here given that nearly all scientific papers display plots. Perhaps instead of "suggest" should be "recommend as a community standard"

REPLY: This sentence was removed during the rewriting of the main text.

2-7 – "had" should be "has"

REPLY: Thank you, we have made the correction.

2-14 – No, numerical simulations are not necessary "to monitor these ongoing changes". Instead, the authors could say, "to contextualize monitoring of ongoing changes"

REPLY: Thank you, we have made the correction.

2-23 – remove "being". Also, the attribution here with "mostly" is overconfidently placed on lack of BGC understanding. In many instances it is lack of understanding of the physics and lack of characterization of the forcing that are the bigger issues than the BGC parameterization.

REPLY: Thank you, we have made the correction.

2-25 – add comma before "and" .

REPLY: Thank you, we have made the correction.

2-30 – add "a" before "few"

REPLY: Thank you, we have made the correction.

3-3 – The list should reflect back to the same part of the sentence, not three different parts. If reflecting back to "to test their", then it should be "to test their predictive skills, ability to reproduce BGC processes, and confidence intervals on model predictions" or if these are separate statements reflecting back to "their" and "to", then "to test their predictive skills and ability to reproduce BGC processes and estimate confidence intervals on model predictions"

REPLY: Thank you, we have made the correction.

3-11 – "All these datasets neither have a" Should be "These datasets have neither"

REPLY: Thank you, we have made the correction.

3-12 – remove "can"

REPLY: Thank you, we have made the correction.

3-23 – "so far essentially sampled" should be "well sampled only"

REPLY: Thank you, we have made the correction.

3-24 – Add "the" before "regional"

REPLY: Thank you, we have made the correction.

3-24 – remove comma before "large"

REPLY: Thank you, we have made the correction.

3-25 – remove comma before "like", and replace "or" with 'and"

REPLY: Thank you, we have made the correction.

4-4 – "represent" should be "represents".

REPLY: Thank you, we have made the correction.

 Also, while this statement may be true in terms of quantity of data for a few parameters, it is not true in terms of either accuracy or comprehensiveness. Just because you can derive an estimate of $SiO_4$ from an $O_2$ sensor does not mean the dataset is better than actually measuring $O_2$ from a Winkler titration, much less using that value to extrapolate $SiO_4$.

4-6 – I do not know what "interoperability" means in this context. Is it something about the inherent environmental variability, or the measurement uncertainty?

4-8 – I don't know what "separately" is being used for here. Are the authors saying that they are initializing the model with "WOA/WOD" and then evaluating performance separately with BGC-Argo? Or that the initialization of the model is done independently from WOA/WOD and then both WOA/WOD and BGC-Argo are used for independent evaluation?

REPLY: We agree that this paragraph was not clear. We have revised it. It now reads:

"

*The BGC-Argo data set represents a significant improvement for the assessment of models comparing to large databases such as the World Ocean Database (WOD) (Boyer et al., 2013) or the Copernicus Marine Service in situ dataset (European Union-Copernicus Marine Service, 2015). Large databases are composed of data collected from various instrument types with heterogenous data sampling methodologies. Therefore, for a given variable, the accuracy numbers are not the same and change depending on the instrument type (European Union-Copernicus Marine Service, 2019). Consequently, this affects the overall accuracy over time due to the changing proportion of instrument types over the years. On the other hand, the BGC-Argo data set is an homogenous data set with strict and uniform sampling methodologies and data Quality-Control (QC) procedures. As a result, the BGC-Argo data set have a satisfactory level of accuracy, which remains stable over time (Johnson et al., 2017; Mignot et al., 2019). Moreover, the number of quality-controlled observations collected every year by the BGC-Argo fleet is now greater than any other data set (Claustre et al., 2020). Using the BGC-Argo dataset as the single evaluation data set is therefore a way to ensure consistent accuracy.* "

4-18 – The sentence "We expect that the methodology employed here (from the data handling to the use of assessment metrics) would be useful and informative for other research teams interested in model evaluation with BGC-Argo floats." Belongs in the discussion/conclusions, not in the introduction.

REPLY: This sentence was removed during the rewriting of the main text.

4-27 – "them" should be "these metrics"

REPLY: Thank you, we have made the correction.

4-29 – ". These metrics" should be "and"

REPLY: This sentence was removed during the rewriting of the main text.

4-31 to 5-5 – Again, the sentences beginning "Further, our validation framework could..." to "… is not addressed in this study" Belongs in the discussion/conclusions, not in the introduction.

REPLY: This sentence was removed during the rewriting of the main text.

4-33 – This sentence needs a lot of work. The authors could try adding "have" before "demonstrated", "flux" before "calculation" and "the" before "basin", remove the commas and remove "of mass fluxes and process rates" and see if it makes sense.

REPLY: This sentence was removed during the rewriting of the main text.

5-3 "use of the word "arduous" seems odd here. Whether something is hard to do is not necessarily relevant. More relevant is whether the effort is warranted… would it be too uncertain so as not to be robust?

REPLY: This sentence was removed during the rewriting of the main text.

5-7 – "follow: s" Should be "follows. S"

REPLY: Thank you, we have made the correction.

5-26 – "variable" should be "variables,"

REPLY: Thank you, we have made the correction.

5-32 – It would be helpful to site the WCRP standard here for essential climate

variables, e.g Bojinski et al, 2014, "The concept of essential climate variables in support of climate research, applications, and policy", BAMS

REPLY: We have cited the WCRP standard, as proposed by the reviewer.

6-1 – Unclear what is intended for "highest" here. Is it "highest quality" or "highest density" or something else?

REPLY: We agree it was not clear. "highest" is intended for "highest level of data modes". We have changed it in the manuscript.

6-11,12 – "points" should be "point"

REPLY: Thank you, we have made the correction.

6-18 – So this means that the low values are biased high as the chance of a low positive value includes the possibility of the value being zero. How big is this problem? What fraction of the data had to be adjusted to zero?

REPLY: We now use an improved version of POC/bbp relationship. Consequently, there are no longer negative values.

6-23 – "floats" should be "float"

REPLY: Thank you, we have made the correction.

6-24 – add comma after "salinity"

REPLY: Thank you, we have made the correction.

6-25 – there should be a statement here on the carbon system data source that is used for the training of the algorithm… eventually the skill has to be traced back to the GLODAP or other data source.

REPLY: Ok, we have added a statement on the carbonate system data source.

6-34:7-2 – The authors should note that whether or not it is "reasonable" to draw these conclusions is also entirely reliant on both the BGC Argo data and the model capturing the underlying environmental variability.

REPLY: Ok, we agree.

7-8 remove comma

REPLY: Thank you, we have made the correction.

7-10 – remove "it" after "and"

REPLY: Thank you, we have made the correction.

7-19 – what is the advantage, if there is one, of saving only weekly and then recreating the daily values with interpolation? Is this to speed the model or otherwise reduce data size?

REPLY: The advantage is to reduce the data size.

7-26 – remove "values". Again, is there an advantage of calculating output offline? Are CO2 fluxes calculated online and saved out? Perhaps it would be better to move this to the next section where the CO2 flux calculation is discussed.

REPLY: This sentence was removed during the rewriting of the main text.

7-32 – and "space to" between "and" and "the"

REPLY: Thank you, we have made the correction.

8-3 – The bias in MLD is provided, but what is the average MLD that would allow me to know the % bias?

REPLY: The BIAS is now indicated in %.

*"The overall mean square difference between the model and the data is equal to ~30% of the overall variance of the observations"*.

8-28 – This is a strange phrasing. It sounds from this that acidification does not impact the subsurface down to 200 m, on the "surface" and the 200-400 m range… Why not just say that acidification is expected to have its largest impact in the upper 400 m and then separately that the present analysis chooses the 200-400 m range of Kwiatkowski? Presumably the surface and 200-400 m ranges are shown to highlight different signals rather than to suggest the area in between is unimportant. This should b clarified.

REPLY: This sentence was removed during the rewriting of the main text.

9-12 – I would replace "first level" with "most simple but indirect level", 9-13 – replace "of" with "associated with" and 9-17 – replace "second level" with "more process level" since the "second level" isn't being pursued.

REPLY: Thank you, we have made the corrections.

9-22:9-26 – A brief statement and reference on the motivation for providing these mesopelagic estimates is warranted. Also, is there a reference or other rationale for this choice of varying depth range? This MLD-1000 m variable depth definition would seem to include the part of the euphotic zone below the mixed layer as "mesopelagic", at least during the growing season. I would have thought the area below the mixed layer within the euphotic zone to look more like the surface than the mesopelagic, or "twilight zone", a constant 200-1000 m range would have been easier to interpret, particularly against the 200-400 definition for pH.

REPLY: Ok , we have added a statement and reference on the motivation for providing these mesopelagic estimates:

"This two-layer comparison between model and BGC-Argo data provides an indirect evaluation of the key mesopelagic processes and fluxes associated with the carbonate chemistry, biological carbon pump and oxygen levels in the mixed, and mesopelagic layers."

We have also added a rationale for the choice of this varying depth range:
"

*The mesopelagic layer is defined as the layer between the MLD and 1000m. For simplicity, we use a simplified definition of the mesopelagic layer proposed by Dall' Olmo and Mork (2014). In their study, this layer is comprised between the deepest of the euphotic layer depth and the MLD, and 1000 m"*

9-33 – remove second "processes" and end sentence after "production"

REPLY: Thank you, we have made the corrections.

10-10 – This sentence is very misleading. The vertical supply of NO3 to the surface is accompanied with remineralized DIC which is the reverse of the biological carbon pump. This sentence should be reworded.

REPLY: OK, we have reworded the sentence.

20-32 - Why define a biased average for O2 300? Shouldn't the average oxygen between 250-300 be referred to O2 275? Why not use the same 200-400 definition as pH? Or 250-350?

11-2 – Similarly, why define O2 1000 as O2 950-1000? Should this be o2 975, or alternatively, defined as 950-1050… do the floats only go down to 1000m? This would seem a reasonable justification if it were the case since gradients at this depth tend to be weak, but still wouldn't explain the odd 250-300 definition.

REPLY: This section was removed during the rewriting of the main text.

12-12 – "on a climatological level" should be "as a climatology"

REPLY: This section was removed during the rewriting of the main text.

12-13 – what is the purpose of "etc.."? "imposes" should be "requires"

12-14 – "in a climatological way" should be "as a climatology"
12-19 – why is "Biogeochemical-Argo Planning Group, 2016" in parenthesis here. Was this means of gridding a recommendation from this group? If so, please be explicit.
12-21 – "clarity" should be "clarity in visualization" or "simplicity in visualization"

12-26 – Add "While" before Taylor" and replace "but" with a comma in the next sentence. That would make It more clear that you are introducing a new topic rather than simply revisiting how great are the first three presentation methods.
12-33 – "for" should be "in"

13-7 – The sentence "Examples of the diagnostic plots described in section 4 in combination with the metrics defined in Section 3 are shown." Seems redundant with the orientation statement in the introduction section and should be removed.

REPLY: These sections were removed during the rewriting of the main text.

13-16:13-25 – The null hypothesis that the reader should use to define "well represented" are not clear. Isn't much or all of this fidelity due to the initial condition derived through the assimilation? I am not sure what to take from this. Is there an "unassimilated" version of the model with which the assimilation should be compared? Or a previous generation model? Or other unassimilative models such as CMIP6? Or is the objective just to show the broad contrast in pattern agreement between model and observations across variables? Why is pH so poorly predicted?

REPLY:  As explained before, we now use the model efficiency statistical score to assess the performance of the model. The null hypothesis for establishing that the model is "good" is tested against the BGC-Argo climatology:

*"The model efficiency tests whether the model outperforms the BGC-Argo climatology ($0 < m_e < 1$ ,Fennel et al., 2022), or stated differently, if the model-data mean square difference is lower than the observation variance, i.e., $\sum_{i=1}^{N}(m_i - o_i)^2 < \sum_{i=1}^{N}(o_i - \bar{o})^2$ . "*

We also provide some explanation as to why pH is so poorly predicted.

14-1:14-4 – This discussion of the value of Taylor diagrams is very superficial and somewhat misleading. The presentation here certainly shows what patterns and variability in different variables are relatively well reproduced, but whether this should inform future model development priorities entirely depends on the intended use of the model and associated requirements. Further, the most common scientific use of Taylor

diagrams is the comparison of the same metric across models so that one can quantify the improvements.

REPLY: We agree. This section was removed during the rewriting of the main text.

14-25 – Without a frame of reference, it is not at all clear whether the model is good or bad. Like in the case of the Taylor diagrams, it seems like the analysis is being done in a vacuum without any awareness of other modeling efforts. There is also the lack of appreciation of the satellite derived estimate for this metric.

REPLY: We agree. This section was removed during the rewriting of the main text.

18-18 – The conclusions "Here, we showed that the spatial maps of model-observations comparison are also informative a posteriori, with respect to the network design, as they highlight sensitive areas where BGC-Argo observations are critical and where sustained BGC-Argo observations are required to better constrain the model. These maps correspond to the regions where the model uncertainty (see RMSD spatial maps in Figs. A22-A44) is the highest, i.e., the Equatorial belt with respect to the carbonate system variables, the Southern Ocean with respect to the nutrients and the DCM variables, and the western boundary currents and OMZs with respect to oxygen." Are very interesting scientific research conclusions but are not at all discussed in the body of the manuscript. This is totally unacceptable. The paper cannot bring in unsupported information at the conclusion stage referencing Appendix material. The authors need to show this or restate these conclusions as hypotheses for future work.

REPLY: We agree, as explained above. We have made the design of the BGC-Argo observing system one the main objective of the study as summarized in the conclusion:

[revised manuscript text omitted]

---

## Author Response (AR3)

We would like to thank Tina Treude for her insightful feedback. We have responded to the Associate Editor's suggestions, with the original comments in black and our responses in blue.

Dear Alexandre Mignot and Co-Workers,

thank you for submitting your thoroughly revised manuscript to Biogeoscience. The reviewer and I agree that the study strongly benefitted from the revision and has now matured to a full scientific manuscript.

REPLY: Thank you for your message regarding our revised manuscript submission to Biogeoscience. We are pleased to hear that the reviewer and you found the revisions to be thorough and that the study has now matured into a comprehensive scientific manuscript.

The reviewer has a couple of minor technical comments that I would like you to take into consideration.

REPLY: We appreciate the reviewer's time and effort in providing feedback on our manuscript. We would like to confirm that we have considered all of the reviewer's technical comments and have addressed them accordingly in our revised manuscript.

I further would like to ask you to revise the following:

1. Title
I find the new title a little hard to read. It is quite long and strangely structured. Try to break it up, for example:

Assessment metrics of BGC-Argo floats: Biogeochemical model performance and observing system design evaluation using an unsupervised machine learning algorithm

REPLY: Thank you for your message and for providing your feedback on the revised title of our manuscript. We have carefully considered your suggestion and have revised the title accordingly to make it more clear and concise. The title now read: "Using machine learning and BGC Argo floats to assess biogeochemical models and optimize observing system design".

2. Red-Green in Figures

Please correct the color scales in Fig 1 and 2 to accommodate red-green color weaknesses (see our figure preparation guideline)

REPLY: Thank you for bringing this to our attention. We have revised all of the figures in our manuscript to ensure that the color scales account for the red-green color deficiency, as per your figure preparation guidelines.

##########################################

We thank the reviewer #3 for their thoughtful comments. Here we offer detailed responses to all questions. Reviewer's comments are in black, our replies are in blue.

I am satisfied with the author revisions to address my concern and recommend publication after addressing a few minor language issues I identified in the first two pages:

REPLY: We thank the reviewer for the positive assessment of our work and for recommending publication. The languages issues has been addressed in the revised version of the manuscript.

1-19 – The attribution of "However, the assessment of biogeochemical models is becoming increasingly challenging due to the continuous improvement in model structure and spatial resolution." Does not make sense to me… why does "improvement in models" make their assessment "increasingly challenging"? Rather than relating to the same metric, e.g. average surface PO4, I think the authors are trying to say that the metrics themselves are becoming more complex. Perhaps the authors mean instead something like "However, with the continuous improvement in model structure and spatial resolution, incorporation of these additional degrees of freedom into fidelity assessment has become increasingly challenging."

REPLY: Thank you, we have made the correction.

1-28 – "ensure" should be "ensures"

REPLY: Thank you, we have made the correction.

1-32 – "compare" should be "compares"

REPLY: Thank you, we have made the correction.

2-6 – remove "about"

REPLY: Thank you, we have made the correction.

2-7 – "network. In particular…" should be "network, in particular…" as this is not a complete sentence.

REPLY: Thank you, we have made the correction.